# Fine-grained Expressivity of Graph Neural Networks

**Jan Böker**
RWTH Aachen University

**Ron Levie**
Technion - Israel Institute of Technology

**Ningyuan Huang**
Johns Hopkins University

**Soledad Villar**
Johns Hopkins University

**Christopher Morris**
RWTH Aachen University

## Abstract

Numerous recent works have analyzed the expressive power of message-passing graph neural networks (MPNNs), primarily utilizing combinatorial techniques such as the 1-dimensional Weisfeiler–Leman test (1-WL) for the graph isomorphism problem. However, the graph isomorphism objective is inherently binary, not giving insights into the degree of similarity between two given graphs. This work resolves this issue by considering continuous extensions of both 1-WL and MPNNs to graphons. Concretely, we show that the continuous variant of 1-WL delivers an accurate topological characterization of the expressive power of MPNNs on graphons, revealing which graphs these networks can distinguish and the level of difficulty in separating them. We identify the finest topology where MPNNs separate points and prove a universal approximation theorem. Consequently, we provide a theoretical framework for graph and graphon similarity combining various topological variants of classical characterizations of the 1-WL. In particular, we characterize the expressive power of MPNNs in terms of the tree distance, which is a graph distance based on the concept of fractional isomorphisms, and substructure counts via tree homomorphisms, showing that these concepts have the same expressive power as the 1-WL and MPNNs on graphons. Empirically, we validate our theoretical findings by showing that randomly initialized MPNNs, without training, exhibit competitive performance compared to their trained counterparts. Moreover, we evaluate different MPNN architectures based on their ability to preserve graph distances, highlighting the significance of our continuous 1-WL test in understanding MPNNs' expressivity.

## 1   Introduction

Graph-structured data is widespread across several application domains, including chemo- and bioinformatics [11, 101], image analysis [107], and social-network analysis [38], explaining the recent growth in developing and analyzing machine learning methods tailored to graphs. In recent years, *message-passing graph neural networks* (MPNNs) [25, 50, 92] emerged as the dominant paradigm, and alongside the growing prominence of MPNNs, numerous works [89, 115] analyzed MPNNs' expressivity. The analysis, typically based on combinatorial techniques such as the 1-*dimensional Weisfeiler–Leman test* (1-WL) for the graph isomorphism problem [113, 114], provides explanations of MPNNs' limitations (see [92] for a thorough survey). However, since the graph isomorphism problem only concerns whether the graphs are exactly the same, it only gives insights into MPNNs' ability to distinguish graphs. Hence, such approaches cannot quantify the graphs' degree of similarity. Nonetheless, understanding the similarities induced by MPNNs is crucial for precisely quantifying their generalization abilities [82, 94], stability [44], or robustness properties [60].

**Present work.** To address these shortcomings, we show how to integrate MPNNs into the theory of *iterated degree measures* first developed by Grebík and Rocha [52], which generalizes the 1-WL and its characterizations to graphons [78]. Integrating MPNNs into this theory allows us to identify the finest

37th Conference on Neural Information Processing Systems (NeurIPS 2023).

topology in which MPNNs separate points, allowing us to prove a universal approximation theorem for graphons. Inspired by the Weisfeiler–Leman distance [26], we show that metrics on measures also integrate beautifully into the theory of iterated degree measures. Concretely, we define metrics $\delta_{\mathsf{P}}$ via the *Prokhorov metric* [99] and $\delta_{\mathsf{W}}$ via an *unbalanced Wasserstein metric* [97] that metrize the compact topology of iterated degree measures. *By leveraging this theory, we show that two graphons are close in these metrics if, and only if, the output of all possible MPNNs, up to a specific Lipschitz constant and number of layers, is close as well.* This refines the result of Chen et al. [26], which shows only one direction of this equivalence, i.e., graphs similar in their metric produce similar MPNN outputs.

We focus on graphons without node feature information to focus purely on MPNNs' ability to distinguish their structure. Our main result offers a topological generalization of classical characterizations of the 1-WL, showing that the above metrics represent the optimal approach for defining a metric variant of the 1-WL. Informally, the main result states the equivalence of our metrics $\delta_{\mathsf{P}}$ and $\delta_{\mathsf{W}}$, the tree distance $\delta_{\square}^{\mathcal{T}}$ of Böker [18], the Euclidean distance of MPNNs' output, and tree homomorphism densities. These metrics arise from the topological variants of the 1-WL test, fractional isomorphisms, MPNNs, and tree homomorphism counts.

**Theorem 1** (informal)**.** The following are equivalent for all graphons $U$ and $W$:

1. $U$ and $W$ are close in $\delta_{\mathsf{P}}$ (or alternatively $\delta_{\mathsf{W}}$).
2. $U$ and $W$ are close in $\delta_{\square}^{\mathcal{T}}$.
3. MPNN outputs on $U$ and $W$ are close for all MPNNs with Lipschitz constant $C$ and $L$ layers.
4. Homomorphism densities in $U$ and $W$ are close for all trees up to order $k$.

Up to now, except for the connection between the tree distance and tree homomorphism densities by Böker [18], these equivalences were only known to hold on a discrete level where graphs are either exactly isomorphic or not. The "closeness" statements in the above theorem are epsilon-delta statements, i.e., for every $\varepsilon > 0$, there is a $\delta > 0$ such that, if graphons are $\delta$-close in one distance measure, they are $\varepsilon$-close in the other distance measures, where the constants are independent of the actual graphons. In particular, for graphs, these constants are independent of their number of vertices. Theorem 1 is formally stated and proved in Appendix C.4. A key point in the proof is to consider compact operators (graphons) as limits of graphs. Empirically, we verify our findings by demonstrating that untrained MPNNs yield competitive predictive performance on established graph-level prediction tasks. Further, we evaluate the usefulness of our derived metrics for studying different MPNN architectures. Our theoretical and empirical results also provide an efficient lower bound of the graph distances in Böker [18], Chen et al. [26] by using the Euclidean distance of MPNN outputs.

In summary, we quantify which distance MPNNs induce, leading to a more fine-grained understanding of their expressivity and separation capabilities. Our results provide a deeper understanding of MPNNs' capacity to capture graph structure, precisely determining when they can and when they cannot assign similar and dissimilar vectorial representations to graphs. *Our work establishes the first rigorous connection between the similarity of graphs and their learned vectorial presentations, paving the way for a more detailed understanding of MPNNs' expressivity and their connection to graph structure.*

## 1.1 Related work and motivation

In the following, we discuss relevant related work and provide additional background and motivation.

**MPNNs.** Following Gilmer et al. [50], Scarselli et al. [105], MPNNs learn a vectorial representation, i.e., a $d$-dimensional real-valued vector, representing each vertex in a graph by iteratively aggregating information from neighboring vertices. Subsequently, MPNNs compute a single vectorial representation of a given graph by aggregating these vectorial vertex representations. Notable instances of this architecture include, e.g., Duvenaud et al. [36], Hamilton et al. [61], and Velickovic et al. [111], which can be subsumed under the message-passing framework introduced in Gilmer et al. [50]. In parallel, approaches based on spectral information were introduced in, e.g., Bruna et al. [23], Defferrard et al. [33], Gama et al. [43], Kipf and Welling [73], Levie et al. [75], and Monti et al. [87]—all of which descend from early work in Baskin et al. [14], Goller and Küchler [51], Kireev [74], Merkwirth and Lengauer [84], Micheli [85], Micheli and Sestito [86], Scarselli et al. [105], and Sperduti and Starita [108].

**Expressivity and limitations of MPNNs.** The *expressivity* of an MPNN is the architecture's ability to express or approximate different functions over a domain, e.g., graphs. High expressivity means the neural network can represent many functions over this domain. In the literature, the expressivity of MPNNs is modeled mathematically based on two main approaches, algorithmic alignment with graph

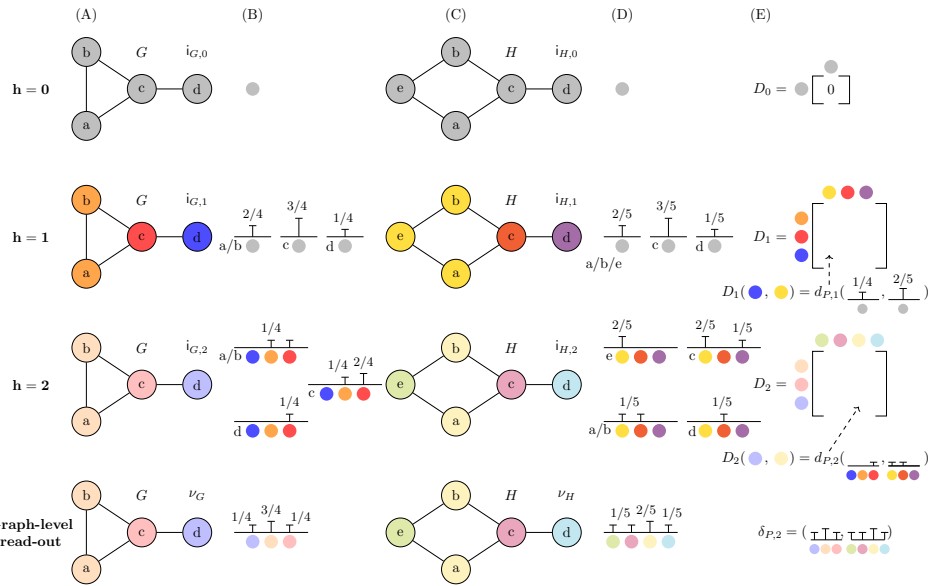

Figure 1: Illustration of the procedure to compute the distance $\delta_\mathsf{P}$ between graphs $G$ and $H$. Columns A and C show the colors obtained by 1-$\mathsf{WL}$ iterations on graphs $G$ and $H$, respectively. Columns B and D show the iterated degree measures (IDMs) $\mathsf{i}_{G,h}$ and $\mathsf{i}_{H,h}$ for iterations $h = 1, 2$ (see Eq. (1)), and the output distributions of iterated degree measures (DIDMs) $\nu_G$ and $\nu_H$ (see Eq. (2)). Column E depicts the recursive construction to compute the distance $\delta_\mathsf{P}$ between the IDMs from columns B and D (outlined in Section 3, detailed in Appendix C.2.2).

isomorphism test [88] and universal approximation theorems [7, 48]. Works following the first approach study if an MPNN, by choosing appropriate weights, can distinguish the same pairs of non-isomorphic graphs as the 1-$\mathsf{WL}$ or its more powerful generalization the $k$-$\mathsf{WL}$. Here, an MPNN distinguishes two non-isomorphic graphs if it can compute different vectorial representations for the two graphs. Specifically, Morris et al. [89] and Xu et al. [115] showed that the 1-$\mathsf{WL}$ limits the expressive power of any possible MPNN architecture in distinguishing non-isomorphic graphs. In turn, these results have been generalized to the $k$-$\mathsf{WL}$, see, e.g., Azizian and Lelarge [7], Geerts [47], Maron et al. [81], Morris et al. [89, 91, 93]. Works following the second approach study, which functions over the domain of graphs, can be approximated arbitrarily close by an MPNN [7, 28, 48, 79]; see also next paragraph. Further, see Appendix B for an extended discussion of related works about MPNNs' expressivity.

**Limitations of current universal approximation theorems for MPNNs.** *Universal approximation theorems* assume that the domain of the network is a compact metric space and show that a neural network can approximate any continuous function over this space. Current approaches studying MPNNs' universal approximation capabilities employ the (graph) edit distance to define the metric on the space of graphs, e.g., see [7]. However, the edit distance is not a natural notion of similarity for practical machine learning on graphs. That is, any two graphs on a different number of vertices are far apart, not fully reflecting the similarity of real-world graphs. More generally, the same holds for any pair of non-isomorphic graphs. Hence, any rewiring of a graph leads to a far-away graph. However, we would like to interpret the rewiring of a small number of edges in a large graph as a small perturbation close to the original graph. Additionally, since the edit metric is not compact, the Stone–Weierstrass theorem, the primary tool in universal approximation analysis, cannot be applied directly to the whole space of graphs. For example, [26] uses other non-compact metrics to circumvent this, artificially choosing a compact subset of graphs from the whole space by uniformly limiting the size of the graphs and the edge weights. Alternatively, Chen et al. [28] resorted to the graph signal viewpoint, allowing real-valued edge features, and showed that the algorithmic alignment of GNNs with graph isomorphism algorithms can be utilized to prove universal approximation theorems for MPNNs. In contrast, here we suggest using graph similarity measures from graphon analysis, for which simple graphs of arbitrarily different sizes can be close to each other and by which the space of all graphs is dense in the compact metric space of graphons, allowing us to use the Stone–Weierstrass theorem directly; see also Appendix B.2 for an extended discussion on graph metrics beyond the edit distance.

**Graphon theory.** The book of Lovász [78] provides a thorough treatment of *graphons*, which emerged as limit objects for sequences of graphs in the theory of dense graph limits developed by Borgs et al. [20, 21], Lovász and Szegedy [77]. These limit objects allow the completion of the space of all graphs to a compact metric space. When endowed with the *cut distance*, first introduced by Frieze and Kannan [41], the space of graphons is compact [78, Theorem 9.23]). We can interpret graphons in two ways. First, as a weighted graph on the continuous vertex set $[0, 1]$. Secondly, we can think of every point from $[0, 1]$ as an infinite set of vertices and two points in $[0, 1]$ as being connected by a random bipartite graph with edge density given by the graphon. This second point of view naturally leads to using graphons as generative models of graphs and the theory of graph limits. Our work follows the first point of view, and we emphasize that we *do not* use graphons as a generative model. This also means that we do not need large graphs for the asymptotics to work since every graph—and, in particular, every small graph—is also a graphon. Grebík and Rocha [52] generalized the 1-WL test and various of its characterizations to graphons, while Böker [19] did this for the $k$-WL test.

**Graphon theory in graph machine learning.** Keriven et al. [64], Maskey et al. [83], Ruiz et al. [103] use graphons to analyze graph signal processing and MPNNs. These papers assume a single fixed graphon generating the data, i.e., any graph from the dataset is randomly sampled from this graphon, and showed that spectral MPNNs on graphs converge to spectral MPNNs on graphons as the size of the sampled graphs increases. Maskey et al. [82] developed a generalization analysis of MPNNs, assuming a pre-defined finite set of graphons. Further, Keriven et al. [65] compared the expressivity of two types of spectral MPNNs on spaces of graphons, assuming graphons are Lipschitz continuous kernels. To that, the metric on the graphon space is taken as the $L_\infty$ distance between graphons as functions. However, the paper does not directly characterize the separation power of the studied classes of MPNNs, and it requires the choice of an arbitrary compact subset to perform the analysis. In contrast, in the current paper, we use graphon analysis to endow the domain of definition of MPNNs, the set of all graphs, with a "well-behaved" structure describing a notion of natural graph similarity and allowing us to analyze properties of MPNNs regardless of any model of the data distribution.

## 2   Background

Here, we provide the necessary background and define notation.

**Analysis.** We denote the Lebesgue measure on $[0, 1]$ by $\lambda$ and consider measurability w.r.t. the Borel $\sigma$-algebra on $[0, 1]$. Let $(X, \mathcal{B})$ be a standard Borel space, where we sometimes just write $X$ when $\mathcal{B}$ is understood and then use $\mathcal{B}(X)$ to explicitly denote $\mathcal{B}$. For a measure $\mu$ on $X$, we let $\|\mu\| := \mu(X)$ denote its *total mass*, and for a standard Borel space $(Y, \mathcal{C})$ and a measurable map $f : X \to Y$, let the *push-forward* $f_*\mu$ *of* $\mu$ *via* $f$ be defined by $f_*\mu(A) := \mu(f^{-1}(A))$ for every $A \in \mathcal{C}$. Let $\mathscr{P}(X)$ and $\mathscr{M}_{\leq 1}(X)$ denote the spaces of all probability measures on $X$ and all measures of total mass at most one on $X$, respectively. Let $C_b(X)$ denote the set of all bounded continuous real-valued functions on $X$. We endow $\mathscr{P}(X)$ and $\mathscr{M}_{\leq 1}(X)$ with the topology generated by the maps $\mu \mapsto \int_X f d\mu$ for $f \in C_b(X)$, the *weak topology* (*weak\* topology* in functional analysis); see [63, Section 17.E] or [17, Chapter 8]. Then, $\mathscr{P}(X)$ and $\mathscr{M}_{\leq 1}(X)$ are again standard Borel spaces, and if $K$ is a compact metric space, then $\mathscr{P}(K)$ and $\mathscr{M}_{\leq 1}(K)$ are compact metrizable; see [63, Theorem 17.22]. For a sequence $(\mu_i)_i$ of measures and a measure $\mu$, we have $\mu_i \to \mu$ if and only if $\int_X f d\mu_i \to \int_X f d\mu$ for every $f \in C_b(X)$, and for measures $\mu, \nu$, we have $\mu = \nu$ if and only if $\int_X f d\mu = \int_X f d\nu$ for every $f \in C_b(X)$. In both statements, we may replace $C_b(X)$ by a dense (w.r.t. the sup norm) subset. See Appendix A.3 for basics on topology. We denote a function $f : A \to B$ also by $f(-)$ or $f_-$ in case the evaluation of $f$ on a point $a \in A$ is denoted by $f(a)$ or $f_a$, respectively.

**Graphs and graphons.** A *graph* $G$ is a pair $(V(G), E(G))$ with *finite* sets of *vertices* or *nodes* $V(G)$ and *edges* $E(G) \subseteq \{\{u, v\} \subseteq V(G) \mid u \neq v\}$. If not otherwise stated, we set $n := |V(G)|$, and the graph is of *order* $n$. We also call the graph $G$ an $n$-order graph. For ease of notation, we denote the edge $\{u, v\}$ in $E(G)$ by $uv$ or $vu$. The *neighborhood* of $v$ in $V(G)$ is denoted by $N(v) := \{u \in V(G) \mid vu \in E(G)\}$ and the *degree* of a vertex $v$ is $|N(v)|$. Two graphs $G$ and $H$ are *isomorphic* and we write $G \simeq H$ if there exists a bijection $\varphi : V(G) \to V(H)$ preserving the adjacency relation, i.e., $uv$ is in $E(G)$ if and only if $\varphi(u)\varphi(v)$ is in $E(H)$. Then $\varphi$ is an *isomorphism* between $G$ and $H$.

A *kernel* is a measurable function $U : [0, 1]^2 \to \mathbb{R}$, and a symmetric measurable function $W : [0, 1]^2 \to [0, 1]$ is called a *graphon*. The set of all graphons is denoted by $\mathcal{W}$. Graphons generalize graphs in the following way. Every graph $G$ can be viewed as a graphon $W_G$ by partitioning $[0, 1]$ into $n$ intervals

$(I_v)_{v \in V(G)}$, each of mass $1/n$, and letting $W_G(x, y)$ for $x \in I_u, y \in I_v$ be one or zero depending on whether $uv$ is an edge in $G$ or not. The *homomorphism density* $t(F, W)$ of a graph $F$ in a graphon $W$ is $t(F, W) := \int_{[0,1]^{V(F)}} \prod_{ij \in E(F)} W(x_i, x_j) \, d(\bar{x})$, where $\bar{x}$ is the tuple of variables $x_v$ for $v \in V(G)$.

**The tree distance.** A graphon $W$ defines an operator $T_W \colon L^2([0,1]) \to L^2([0,1])$ on the space $L^2([0,1])$ of square-integrable functions modulo equality almost everywhere by setting $(T_W f)(x) := \int_{[0,1]} W(x, y) f(y) \, d\lambda(y)$ for every $x \in X$ and every $f \in L^2([0,1])$. Böker [18] defined the *tree distance* of two graphons $U$ and $W$ by $\delta_\square^{\mathcal{T}}(U, W) := \inf_S \sup_{f,g} |\langle f, (T_U \circ S - S \circ T_W) g \rangle|$, where the supremum is taken over all measurable functions $f, g \colon [0, 1] \to [0, 1]$ and the infimum is taken over all *Markov operators* $S$, i.e., operators $S \colon L^2([0, 1]) \to L^2([0, 1])$ such that $S(f) \geq 0$ for every $f \geq 0$, $S(\mathbf{1}_{[0,1]}) = \mathbf{1}_{[0,1]}$, and $S^*(\mathbf{1}_{[0,1]}) = \mathbf{1}_{[0,1]}$ also for the *Hilbert adjoint* $S^*$. Markov operators are the infinite-dimensional analog to doubly stochastic matrices, see [39] for a thorough treatment of Markov operators. One can verify that the tree distance is a lower bound for the *cut distance* [18].

**Iterated measures.** Here, we define *iterated degree measures (IDMs)*, which is basically a sequence of measures, by adapting the definition of Grebík and Rocha [52]. Let $\mathbb{M}_0 := \{1\}$ and inductively define $\mathbb{M}_{h+1} := \mathscr{M}_{\leq 1}(\mathbb{M}_h)$ for every $h \geq 0$. Then, the spaces $\mathbb{M}_0, \mathbb{M}_1, \ldots$ are all compact metrizable. For $0 \leq h < \infty$, inductively define the *projection* $p_{h+1,h} \colon \mathbb{M}_{h+1} \to \mathbb{M}_h$ by letting $p_{1,0}$ be the trivial map and, for $h > 0$, letting $p_{h+1,h}(\alpha) := (p_{h,h-1})_* \alpha$ for every $\alpha \in \mathbb{M}_{h+1} = \mathscr{M}_{\leq 1}(\mathbb{M}_h)$. This extends to $p_{h,\ell} \colon \mathbb{M}_h \to \mathbb{M}_\ell$ for $0 \leq \ell \leq h < \infty$ by composition in the obvious way. Let

$$\mathbb{M} := \mathbb{M}_\infty := \Big\{ (\alpha_h)_h \in \prod_{h \in \mathbb{N}} \mathbb{M}_h \mid p_{h+1,h}(\alpha_{h+1}) = \alpha_h \text{ for every } h \in \mathbb{N} \Big\}$$

be the *inverse limit* of $\mathbb{M}_0, \mathbb{M}_1, \ldots$; see the Kolmogorov Consistency Theorem [63, Theorem 17.20]). Then, $\mathbb{M}$ is compact metrizable [52, Claim 6.2]. For $0 \leq h < \infty$, let $p_{\infty,h} \colon \mathbb{M} \to \mathbb{M}_h$ denote the projection to the $h$th component. We remark that we slightly simplified the definition of Grebík and Rocha [52] by not including previous IDMs from $\mathbb{M}_{h'}$ for $h' \leq h$ in $\mathbb{M}_{h+1}$ and directly defining $\mathbb{M}$ as the inverse limit; corresponding to the definition of the space $\mathbb{P}$ in [52]. These changes yield equivalent definitions that simplify the exposition.

**The 1-WL for graphons.** See Appendix A.1 for the standard definition of 1-WL on graphs. Grebík and Rocha [52] generalized 1-WL to graphons by defining a map $i_W \colon [0, 1] \to \mathbb{M}$, mapping every point of the graphon $W \in \mathcal{W}$ to an iterated degree measure as follows. First, inductively define the map $i_{W,h} \colon [0, 1] \to \mathbb{M}_h$ by setting $i_{W,0}(x) := 1$ for every $x \in [0, 1]$ and

$$i_{W,h+1}(x) := A \mapsto \int_{i_{W,h}^{-1}(A)} W(x, y) d\lambda(y), \tag{1}$$

for all $x \in [0, 1], A \in \mathcal{B}(\mathbb{M}_h)$, and $h \geq 0$. Intuitively, $i_{W,h}(x)$ is the color assigned to point $x$ after $h$ iterations of 1-WL. Observe that $i_{W,1}(x)$ encodes the degree of point $x$, and $i_{W,h}(x)$ for $h > 1$ represents the iterated degree sequence information. Then, we define $i_W := i_{W,\infty} \colon [0, 1] \to \mathbb{M}$ by $i_W(x) := i_{W,\infty}(x) := \prod_{h \in \mathbb{N}} i_{W,h}(x)$ and let

$$\nu_W := \nu_{W,\infty} := (i_W)_* \lambda \in \mathscr{P}(\mathbb{M}) \tag{2}$$

be the *distribution of iterated degree measures (DIDM) of* $W$. In other words, $\nu_W(A)$ is the volume that the colors in $A$ occupy in the graphon domain $[0, 1]$. Then, the 1-*WL test on graphons* is the mapping that takes a graphon $W \in \mathcal{W}$ and returns $\nu_W$. In addition to $\nu_W$, we also define $\nu_{W,h} := (i_{W,h})_* \lambda \in \mathscr{P}(\mathbb{M}_h)$ for $0 \leq h < \infty$, corresponding to running 1-WL for $h$ rounds.

While every DIDM of a graphon is a measure from the compact space $\mathscr{P}(\mathbb{M})$, not every measure in $\mathscr{P}(\mathbb{M})$ is the DIDM of a graphon. Grebík and Rocha [52] address this by giving a definition of a DIDM that is independent of a specific graphon. For us, it suffices to remark that the set $\mathbb{D}_h := \{\nu_{W,h} \mid W \text{ graphon}\} \subseteq \mathscr{P}(\mathbb{M}_h)$ is compact as it is the image of the compact space of graphons [78, Theorem 9.23] under a continuous function [52]. For us, this means that $\mathbb{D}_h$ and $\mathscr{P}(\mathbb{M}_h)$ can be used interchangeably in our arguments, and we do not have to be overly careful with distinguishing them. For simplicity, we simply stick to $\mathscr{P}(\mathbb{M}_h)$ and refer to all elements of $\mathscr{P}(\mathbb{M}_h)$ as *DIDMs*.

**Message-passing graph neural networks.** MPNNs learn a $d$-dimensional real-valued vector for each vertex in a graph by aggregating information from neighboring vertices; see Appendix A.2 for more details. Here, we consider MPNNs where both the update functions and the readout functions are Lipschitz continuous and use sum aggregation normalized by the order of the graph. Formally, we first let

$\boldsymbol{\varphi} = (\varphi_i)_{i=0}^L$ denote a tuple of continuous functions $\varphi_0 \colon \mathbb{R}^0 \to \mathbb{R}^{d_0}$ and $\varphi_t \colon \mathbb{R}^{d_{t-1}} \to \mathbb{R}^{d_t}$ for $t \in [L]$, where we simply view $\varphi_0$ as an element of $\mathbb{R}^{d_0}$. Furthermore, let $\psi$ denote a continuous function $\psi \colon \mathbb{R}^{d_L} \to \mathbb{R}^d$. For a graph $G$, an MPNN initializes a feature $\mathbf{h}_v^{(0)} := \varphi_0 \in \mathbb{R}^{d_0}$. Then, for $t \in [L]$, we compute $\mathbf{h}_-^{(t)} \colon V(G) \to \mathbb{R}^{d_t}$ and the single graph-level feature $\mathbf{h}_G \in \mathbb{R}^d$ after $L$ layers by

$$\mathbf{h}_v^{(t)} := \varphi_t\left(\frac{1}{|V(G)|} \sum_{u \in N(v)} \mathbf{h}_u^{(t-1)}\right) \qquad \text{and} \qquad \mathbf{h}_G := \psi\left(\frac{1}{|V(G)|} \sum_{v \in V(G)} \mathbf{h}_v^{(L)}\right).$$

For a graphon $W \in \mathcal{W}$, an MPNN initializes a feature $\mathbf{h}_x^{(0)} := \varphi_0 \in \mathbb{R}^{d_0}$ for $x \in [0,1]$. Then, for $t \in [L]$, we compute $\mathbf{h}_-^{(t)} \colon [0,1] \to \mathbb{R}^{d_t}$ and the single graphon-level feature $\mathbf{h}_W \in \mathbb{R}^d$ after $L$ layers by

$$\mathbf{h}_x^{(t)} := \varphi_t\left(\int_{[0,1]} W(x,y)\mathbf{h}_y^{(t-1)} \, d\lambda(y)\right) \qquad \text{and} \qquad \mathbf{h}_W := \psi\left(\int_{[0,1]} \mathbf{h}_x^{(L)} \, d\lambda(x)\right).$$

This generalizes the previous definition, i.e., for a graph $G$ and its (induced) graphon $W_G$, we have $\mathbf{h}_G = \mathbf{h}_{W_G}$ and $\mathbf{h}_v^{(t)} = \mathbf{h}_x^{(t)}$ for all $t \in [L]$, $v \in V(G)$, and $x \in I_v$; see Appendix C.1.

We now extend the definition of MPNNs to IDMs. While the above definition of $\mathbf{h}_-^{(t)}$ depends on a specific graphon $W$, an IDM already carries the aggregated information of its neighborhood. Hence, the initial feature $\mathbf{h}_\alpha^{(0)} := \varphi_0 \in \mathbb{R}^{d_0}$ for $\alpha \in \mathbb{M}_0$ and $\mathbf{h}_-^{(t)} \colon \mathbb{M}_t \to \mathbb{R}^{d_t}$ *are defined for all IDMs at once*. The intuition is that MPNNs cannot assign at layer $t$ different feature values to nodes that have the same color at step $t$ of the graphon 1-WL algorithm. Hence, it is enough to only consider an assignment between colors and feature values, and not consider the graph/graphon structure directly. Then, for a DIDM $\nu \in \mathscr{P}(\mathbb{M}_L)$, we define the single DIDM-level feature $\mathbf{h}_\nu \in \mathbb{R}^d$. Formally, we let

$$\mathbf{h}_\alpha^{(t)} := \varphi_t\left(\int_{\mathbb{M}_{t-1}} \mathbf{h}_-^{(t-1)} \, d\alpha\right) \qquad \text{and} \qquad \mathbf{h}_\nu := \psi\left(\int_{\mathbb{M}_L} \mathbf{h}_-^{(L)} \, d\nu\right).$$

That is, messages are aggregated via the IDM itself. In addition to $\mathbf{h}_-^{(t)} \colon \mathbb{M}_t \to \mathbb{R}^{d_t}$, and $\mathbf{h}_- \colon \mathscr{P}(\mathbb{M}_L) \to \mathbb{R}^d$, we define $\mathbf{h}_-^{(t)} \colon \mathbb{M} \to \mathbb{R}^{d_t}$ and $\mathbf{h}_- \colon \mathscr{P}(\mathbb{M}) \to \mathbb{R}^d$ by setting $\mathbf{h}_\alpha^{(t)} := \mathbf{h}_{p_{\infty,t}(\alpha)}^{(t)}$ for every $\alpha \in \mathbb{M}$ and $\mathbf{h}_\nu := \mathbf{h}_{(p_{\infty,L})_* \nu}$ for every $\nu \in \mathscr{P}(\mathbb{M})$; it will always be clear from the context which of these functions we mean. These definitions of MPNNs on IDMs extend the previous definitions of MPNNs on graphons via the following identities. For a graphon $W \in \mathcal{W}$, we have $\mathbf{h}_W = \mathbf{h}_{\nu_{W,L}} = \mathbf{h}_{\nu_W}$ and $\mathbf{h}_x^{(t)} = \mathbf{h}_{i_{W,t}(x)}^{(t)} = \mathbf{h}_{i_W(x)}^{(t)}$ for almost every $x \in [0,1]$; see Appendix C.1. That is, the feature values of an MPNN on a graphon $W$ are equal to the feature values of that MPNN on the (D)IDM computed by the 1-WL on $W$. We call a tuple $\boldsymbol{\varphi}$ as defined above an *(L-layer) MPNN model* if $\varphi_t$ is Lipschitz continuous on $\{\int_{\mathbb{M}_{t-1}} \mathbf{h}_-^{(t-1)} \, d\alpha \mid \alpha \in \mathbb{M}_t\}$ for every $t \in [L]$, and we call $\psi$ as defined above *Lipschitz* if it is Lipschitz continuous on $\{\int_{\mathbb{M}_L} \mathbf{h}_-^{(L)} \, d\nu \mid \nu \in \mathscr{P}(\mathbb{M}_L)\}$. We use $\|-\|_L$ to denote the Lipschitz constants on these sets. In this paper, $\boldsymbol{\varphi}$ and $\psi$ always denote an MPNN model and a Lipschitz function, respectively. We use the term $\infty$-layer MPNN model to refer to an $L$-layer MPNN model for an arbitrary $L$.

## 3 Metrics on iterated degree measures

Chen et al. [26] recently introduced the *Weisfeiler–Leman distance*, a polynomial-time computable pseudometric on graphs combining the 1-WL test with the well-known *Wasserstein metric* from optimal transport [112], where their approach resembles that of iterated degree measures as introduced by Grebík and Rocha [52]. To use metrics from optimal transport, Chen et al. [26] resorted to mean aggregation instead of sum aggregation to obtain probability measures instead of finite measures with total mass at most one. Using mean aggregation, however, is different from the 1-WL test, which relies on sum aggregation. That is, sum aggregation allowed the algorithm to start with a constant coloring, something impossible with mean aggregation, potentially leading to a constant coloring. Chen et al. [26] circumvented this problem by encoding vertex degrees and the total number of vertices in the initial coloring.

Here, we show that the *Prokhorov metric* [99] and an unbalanced variant of the Wasserstein metric can be beautifully integrated into the theory of iterated degree measures, eliminating the need to work around the limits of mean aggregation. Both metrics metrize the weak topology, which is precisely the topology of the

space $\mathbb{M}_h$ of IDMs is endowed with; see Section 2. In modern-day literature, the Prokhorov metric is usually only defined for probability measures [35, Section 11.3], yet the original definition by Prokhorov [99] already was for finite measures. That is, let $(S, d)$ be a complete separable metric space with Borel $\sigma$-algebra $\mathcal{B}$. For a subset $A \subseteq S$ and $\varepsilon \geq 0$, let $A^\varepsilon := \{y \in S \mid d(x, y) < \varepsilon \text{ for some } x \in A\}$, and define the *Prokhorov metric* $\mathsf{P}$ on $\mathcal{M}_{\leq 1}(S)$ by

$$\mathsf{P}(\mu, \nu) := \inf\{\varepsilon > 0 \mid \mu(A) \leq \nu(A^\varepsilon) + \varepsilon \text{ and } \nu(A) \leq \mu(A^\varepsilon) + \varepsilon \text{ for every } A \in \mathcal{B}\}.$$

As the name suggests, $\mathsf{P}$ is a metric on $\mathcal{M}_{\leq 1}(S)$ [99, Section 1.4], and moreover, convergence in $\mathsf{P}$ is equivalent to convergence in the weak topology [99, Theorem 1.11]. For the *unbalanced Wasserstein metric* let $\mu, \nu \in \mathcal{M}_{\leq 1}(S)$, where we assume $\|\mu\| \geq \|\nu\|$ without loss of generality, and define

$$\mathsf{W}(\mu, \nu) := \|\mu\| - \|\nu\| + \inf_{\gamma \in \mathcal{M}(\mu, \nu)} \int_{S \times S} d(x, y) \, d\gamma(x, y),$$

where $\mathcal{M}(\mu, \nu)$ is the set of all measures $\gamma \in \mathcal{M}_{\leq 1}(S \times S)$ such that $(p_1)_* \gamma \leq \mu$ and $(p_2)_* \gamma = \nu$. Here, $p_1$ and $p_2$ are the projections from $S \times S$ upon $S$ to the first and the second component, respectively, i.e., $(p_1)_* \gamma(A) = \gamma(A \times S)$ and $(p_2)_* \gamma(A) = \gamma(S \times A)$. We prove that $\mathsf{W}$ is a well-defined metric on $\mathcal{M}_{\leq 1}(S, \mathcal{B})$ that coincides with the Wasserstein distance [35, Section 11.8] on probability measures. Furthermore, it satisfies $\mathsf{W}(\mu, \nu) \leq 2\mathsf{P}(\mu, \nu) \leq 4\sqrt{\mathsf{W}(\mu, \nu)}$ for all $\mu, \nu \in \mathcal{M}_{\leq 1}(S)$, which implies that it metrizes the weak topology; see Appendix C.2.

The metric $\mathsf{P}$ is used to define the metric $d_{\mathsf{P}, h}$ on $\mathbb{M}_h$ for $0 \leq h \leq \infty$ as follows. Let $d_{\mathsf{P}, 0}$ be the trivial metric on the one-point space $\mathbb{M}_0$ and, for $h \geq 0$, inductively let $d_{\mathsf{P}, h+1}$ be the Prokhorov metric on $(\mathbb{M}_{h+1}, d_{\mathsf{P}, h})$. Then, define $d_{\mathsf{P}, \infty}$ on $\mathbb{M}_\infty$ by setting $d_{\mathsf{P}}(\alpha, \beta) := d_{\mathsf{P}, \infty}(\alpha, \beta) := \sup_{h \in \mathbb{N}} \frac{1}{h} \cdot d_{\mathsf{P}, h}(\alpha_h, \beta_h)$ for $\alpha, \beta \in \mathbb{M}_\infty$. The factor of $1/h$ is included in this definition on purpose to ensure that $d_{\mathsf{P}, \infty}$ metrizes the product topology and not the uniform topology. The metric $d_{\mathsf{W}, h}$ on $\mathbb{M}_h$ is defined completely analogously via $\mathsf{W}$ instead of $\mathsf{P}$.

The metrics $d_{\mathsf{P}, h}$ and $d_{\mathsf{W}, h}$ on $\mathbb{M}_h$ allow us, for example, to compare the IDM of a point in a graphon to the IDM of a point in another graphon. To compare two graphons' distributions on iterated degree measures, we let $\delta_{\mathsf{P}, h}$ be the Prokhorov metric on $(\mathscr{P}(\mathbb{M}_h), d_{\mathsf{P}, h})$ for $0 \leq h \leq \infty$ and again define $\delta_{\mathsf{W}, h}$ analogously via the distance $\mathsf{W}$. We note that these metrics directly apply to graphons $U, W \in \mathcal{W}$ by simply comparing their DIDMs $\nu_{U, h}$ and $\nu_{W, h}$.

**Theorem 2.** Let $0 \leq h \leq \infty$. The metrics $d_{\mathsf{P}, h}$ and $d_{\mathsf{W}, h}$ are well-defined and metrize the topology of $\mathbb{M}_h$. The metrics $\delta_{\mathsf{P}, h}$ and $\delta_{\mathsf{W}, h}$ are well-defined and metrize the topology of $\mathscr{P}(\mathbb{M}_h)$. Moreover, these metrics are computable on graphs in time polynomial in the size of the input graphs and $h$, up to an additive error of $\varepsilon$ in the case of $d_{\mathsf{W}, \infty}$ and $\delta_{\mathsf{W}, \infty}$.

While these metrics are polynomial-time computable, in Appendix C.2, we derive the same impractical upper bound of $\mathcal{O}(h \cdot n^5 \cdot \log n)$ for $\delta_{\mathsf{W}, h}$ as Chen et al. [26] get for their Weisfeiler–Leman distance and the even worse bound of $\mathcal{O}(h \cdot n^7)$ for $\delta_{\mathsf{P}, h}$. This means that these metrics are not suitable as a computational tool in practice. Theorem 1 hints that we can instead use easy-to-compute MPNNs to lower-bound these metrics, which leads to our experiments in Section 6.

MPNNs are Lipschitz in the metrics we defined, where the Lipschitz constant only depends on basic properties of the MPNN model. That is, if two graphons are close in our metrics, then MPNNs outputs for *all* MPNN models up to a specific Lipschitz constant are close. Formally, let $\boldsymbol{\varphi} = (\varphi_i)_{i=0}^L$ be an MPNN model with $L$ layers, and for $t \in \{0, \ldots, L\}$, let $\boldsymbol{\varphi}_t := (\varphi_i)_{i=0}^t$. Then, we inductively define the *Lipschitz constant* $C_{\boldsymbol{\varphi}} \geq 0$ of $\boldsymbol{\varphi}$ by $C_{\boldsymbol{\varphi}_0} := 0$ for $t = 0$ and $C_{\boldsymbol{\varphi}_t} := \|\varphi_t\|_L \cdot (\|\mathbf{h}_-^{(t-1)}\|_\infty + C_{\boldsymbol{\varphi}_{t-1}})$ for $t > 0$. This essentially depends on the product of the Lipschitz constants of the functions in $\boldsymbol{\varphi}$, and the bounds for the MPNN output values, which are finite since a continuous function on a compact set attains its maximum. Including these bounds in the constant is necessary since we consider sum aggregation. That is, a constant function mapping all inputs to some $c \in \mathbb{R}$ has Lipschitz constant zero, but when integrated with measures of total mass zero and one, for example, the difference of the outputs is $c$. We define $C_{(\boldsymbol{\varphi}, \psi)}$ for Lipschitz $\psi$ analogously by essentially viewing $(\boldsymbol{\varphi}, \psi)$ as an MPNN model.

**Lemma 3.** Let $\boldsymbol{\varphi}$ be an $L$-layer MPNN model for $L \in \mathbb{N}$ and $\psi$ be Lipschitz. Then,

$$\|\mathbf{h}_\alpha^{(L)} - \mathbf{h}_\beta^{(L)}\|_2 \leq C_{\boldsymbol{\varphi}} \cdot d_{\mathsf{W}, L}(\alpha, \beta) \qquad \text{and} \qquad \|\mathbf{h}_\mu - \mathbf{h}_\nu\|_2 \leq C_{(\boldsymbol{\varphi}, \psi)} \cdot \delta_{\mathsf{W}, L}(\mu, \nu)$$

for all $\alpha, \beta \in \mathbb{M}_L$ and all $\mu, \nu \in \mathscr{P}(\mathbb{M}_L)$, respectively. These inequalities also hold for $d_{\mathsf{W}, \infty}$ and $\delta_{\mathsf{W}, \infty}$ with an additional factor of $L$ in the Lipschitz constant.

# 4 Universality of message-passing graph neural networks

In this section, we prove a universal approximation theorem for MPNNs on IDMs and DIDMs, deriving our main result from it. For $0 \leq L \leq \infty$, let $\mathcal{N}_L^n \subseteq C(\mathbb{M}_L, \mathbb{R}^n)$ denote the set of all functions $\mathbf{h}_-^{(L)} \colon \mathbb{M}_L \to \mathbb{R}^n$ for an $L$-layer MPNN model $\varphi$ with $d_L = n$. Similarly, let

$$\mathcal{NN}_L^n := \{\mathbf{h}_- \mid \varphi \; L\text{-layer MPNN model}, \psi \colon \mathbb{R}^{d_L} \to \mathbb{R}^n \text{ Lipschitz}\} \subseteq C(\mathscr{P}(\mathbb{M}_L), \mathbb{R}^n)$$

be the set of all functions computed by an MPNN after a global readout. Our universal approximation theorem, Theorem 4, shows that all continuous functions on IDMs and DIDMs, i.e., functions on graphons that are invariant w.r.t. the (colors of) the 1-WL test, can be approximated by MPNNs. Hence, our result extends the universal approximation result of Chen et al. [26] for *measure Markov chains* in two ways. First, measure Markov chains are restricted to finite spaces by definition, which is not the case for graphons and our universal approximation theorem. Secondly, the spaces $\mathbb{M}_L$ and $\mathscr{P}(\mathbb{M}_L)$ are compact, which means we obtain a universal approximation theorem for the whole space of graphons, including all graphs, not restricted to an artificially chosen compact subset.

**Theorem 4.** Let $0 \leq L \leq \infty$. Then, $\mathcal{N}_L^1$ is dense in $C(\mathbb{M}_L, \mathbb{R})$ and $\mathcal{NN}_L^1$ is dense in $C(\mathscr{P}(\mathbb{M}_L), \mathbb{R})$.

The proof of Theorem 4 is elegant and does not rely on encoding the 1-WL test as an MPNN. That is, it follows by inductive applications of the Stone–Weierstrass theorem [35, Theorem 2.4.11] combined with the definition of IDMs. It is strikingly similar to the proof of Grebík and Rocha [52] for a similar result concerning tree homomorphism densities; see Appendix D.

While the second statement of Theorem 4, i.e., the graphon-level approximation, is interesting in its own right, the crux of Theorem 4 lies in its first statement, namely, that $\mathcal{N}_L^1$ is dense in $C(\mathbb{M}_L, \mathbb{R})$, immediately implying that the topology induced by MPNNs on $\mathscr{P}(\mathbb{M}_L)$ is the weak topology, i.e., the topology we endowed this space within Section 2.

**Corollary 5.** Let $0 \leq L \leq \infty$ and $n > 0$. Let $\nu \in \mathscr{P}(\mathbb{M}_L)$ and $(\nu_i)_i$ be a sequence with $\nu_i \in \mathscr{P}(\mathbb{M}_L)$. Then, $\nu_i \to \nu$ if and only if $\mathbf{h}_{\nu_i} \to \mathbf{h}_\nu$ for all $L$-layer MPNN models $\varphi$ and Lipschitz $\psi \colon \mathbb{R}^{d_L} \to \mathbb{R}^n$.

By combining standard compactness arguments with Theorem 2 and Corollary 5, we can now prove that two graphons are close in our metrics if and only if the output of all possible MPNNs, up to a specific constant and number of layers, is close. Formally, the forward direction of this equivalence is just Lemma 3, while the backward direction reads as follows.

**Theorem 6.** Let $n > 0$ be fixed. For every $\varepsilon > 0$, there are $L \in \mathbb{N}, C > 0$, and $\delta > 0$ such that, for all graphons $U$ and $W$, if $\|\mathbf{h}_U - \mathbf{h}_W\|_2 \leq \delta$ for every $L'$-layer MPNN model $\varphi$ and Lipschitz $\psi \colon \mathbb{R}^{d_{L'}} \to \mathbb{R}^n$ with $L' \leq L$ and $C_{(\varphi, \psi)} \leq C$, then $\delta_\mathsf{P}(U, W) \leq \varepsilon$.

We stress that the constants $L$, $C$, and $\delta$ in the theorem statement are independent of the graphons $U$ and $W$. The proof of Theorem 6 is simple: one assumes that the statement does not hold to obtain two sequences of counterexamples, which have to have convergent subsequences by compactness, and the limit graphons allow us to derive a contradiction. This establishes the equivalence between our metrics and MPNNs stated in Theorem 1. Analogous reasoning together with the universality of tree homomorphism densities [52], cf. Appendix D, yields the equivalence between our metrics and tree homomorphism densities. Then, the missing equivalence to the tree distance follows from the result of Böker [18], which connects the tree distance to tree homomorphism densities. See Appendix C.4 for the formal statements of all these equivalences as epsilon-delta-statements and their proofs.

# 5 Extension to graphons with signals

Our focus in this work lies on MPNNs' ability to distinguish the structure of graphons. However, the definitions of IDMs, MPNNs, and our metrics can be adapted to graphons with signals, i.e., graphons $W$ equipped with a measurable *signal function* $\ell \colon [0, 1] \to K$, where $(K, d)$ is some fixed compact metric space. In the following, we briefly sketch how to do this. First, replace the one-point space $\mathbb{M}_0$ by $(K, d)$ and modify the 1-WL for graphons by setting $\mathsf{i}_{(W,\ell),0} := \ell$, i.e., use the signal function as the initial coloring. For $h > 0$, adapt the definition of the IDM spaces $\mathbb{M}_h$ and of the refinement rounds $\mathsf{i}_{(W,\ell),h}$ of the 1-WL for graphons to include the previous IDM of a point in its new IDM, like in the original definition of Grebík and Rocha [52]. Omitting the previous IDM, as we have done before in our definition, is only reasonable if the initial coloring is constant. Then, since $\mathbb{M}_0 = (K, d)$ is a compact metric space,

the spaces $\mathbb{M}_h$ are compact metrizable as before. Modify the definition of an MPNN model $\varphi$ such that $\varphi_0$ is a Lipschitz function $K \to \mathbb{R}^{d_0}$. Finally, adapt the definition of the metrics $d_{\mathsf{P},h}$ and $d_{\mathsf{W},h}$ by letting $d_{\mathsf{P},0} := d_{\mathsf{W},0} := d$ be the metric of the compact metric space $(K, d)$ and then adapting $d_{\mathsf{P},h}$ and $d_{\mathsf{W},h}$ for $h > 0$ to the modified definition of $\mathbb{M}_h$ by using the product metric. Then, the proofs of our results can be adapted to this more general setting. In particular, one can prove a universal approximation theorem since Lipschitz functions on $K$ are dense in $C(K)$.

## 6  Experimental evaluation

In the following, we investigate the applicability of our theory on real-world prediction tasks. Specifically, we answer the following questions.

**Q1** To what extent do our graph metrics $\delta_{\mathsf{P}}$ and $\delta_{\mathsf{W}}$ act as a proxy for distances between MPNNs' vectorial representations?

**Q2** Our theoretical results imply that untrained MPNNs can be as effective as their trained counterparts when using *enough* of them. Can untrained MPNNs remain competitive when only using a finite number of them (measured by the hidden dimensionality)?

The source code of all methods and evaluation protocols are available at `https://github.com/nhuang37/finegrain_expressivity_GNN`. We conducted all experiments on a server with 256 GB RAM and four NVIDIA RTX A5000 GPU cards.

**Fine-grained expressivity comparisons of MPNNs.** To answer **Q1**, we construct a graph sequence that converges in our graph metrics. Given an MPNN, we compute the sequence of its embeddings on such graphs and the corresponding embedding distance using the $\ell_2$-norm. Hence, comparing different MPNNs amounts to comparing the convergence rate of their Euclidean embedding distances concerning the graph distances. Concretely, we simulate a sequence of 50 random graphs $\{G_i\}$ for $i \in [50]$ with 30 vertices using the stochastic block model, where $G_i \sim \mathrm{SBM}(p, q_i)$, with $p = 0.5$ and $q_i \in [0.1, 0.5]$ increases equidistantly. Let $G$ denote the last graph in the sequence, and observe that $G$ is sampled from an Erdős–Rényi model, i.e., $G \sim \mathrm{ER}(p)$. For $i \in [50]$, we compute the Wasserstein distance $\delta_{\mathsf{W},h}(G_i, G)$ and the Euclidean distance $\|\mathbf{h}_{G_i} - \mathbf{h}_G\|_2$.[1] For demonstration purposes, we compare two common MPNN layers, GIN [115] and GraphConv [89], using sum aggregation normalized by the graph's order, varying the hidden dimensions and the number of layers.

Figure 2 visualizes their normalized embedding distance and normalized graph distance, with an increasing number of hidden dimensions, from left to right. GIN, top-row, and GraphConv, bottom-row, produce more discriminative embeddings as the number of hidden dimensions increases, supporting Theorem 4. Each point corresponds to a different (untrained) MPNN. Note that we interpret the hidden dimension $w$ (i.e., width) as concatenating $w$ random MPNNs. We observe similar behavior when increasing the number of layers; see Figure 3 in the appendix. Untrained GraphConv embeddings are more robust than untrained GIN embeddings regarding the choice of hidden dimensions and number of layers. Figure 4 in the appendix shows the same experiments on the real-world dataset MUTAG, part of the TUDataset [90]. We observe that increasing the number of hidden dimensions improves performance. Nonetheless, increasing the number of layers seems to first improve and then degrade performance. This observation coincides with the downstream graph classification performance, as discussed in the next section.

**The surprising effectiveness of untrained MPNNs.** To answer **Q2**, we compare popular MPNN architectures, i.e., GIN and GraphConv, with their untrained counterparts. For untrained MPNNs, we freeze their input and hidden layer weights that are randomly initialized and only optimize for the output layer(s) used for the final prediction. We benchmark on a subset of the established TUDataset [90]. For each dataset, we run *paired* experiments of trained and untrained MPNNs on the same ten random splits (train/test) and 10-fold cross-validation splits, using the evaluation protocol outlined in Morris et al. [90]. We report the test accuracy with 10-run standard deviation in Table 1 and the mean training time per epoch with standard deviation in Table 2 in the appendix. Table 1 and Table 2 show that untrained MPNNs with sufficient hidden dimensionality perform competitively as trained MPNNs while being significantly faster, with 20%-46% time savings. As shown in Figure 6 in the appendix, increasing the hidden dimension (i.e., the number of MPNN models) improves the performance of untrained MPNNs. Our theory states that it is

---

[1] We compute $\delta_{\mathsf{W},h}$ via a min-cost-flow algorithm and terminate at most 3 iterations after the Weisfeiler–Leman colors stabilize.

Table 1: Untrained MPNNs show competitive performance as trained MPNNs given sufficiently large hidden dimensionality (3-layer, 512-hidden-dimension). To be consistent with our theory, we use standard architectures with sum aggregation, layer-wise $1/V(G)$ normalization, and mean pooling, denoted by "MPNN-m." We report the mean accuracy $\pm$ std over ten data splits.

| Accuracy ↑ | Mutag | Imdb-Binary | Imdb-Multi | NCI1 | Proteins | Reddit-Binary |
|---|---|---|---|---|---|---|
| GIN-m (trained) | $79.01 \pm 2.24$ | $69.96 \pm 1.43$ | $46.29 \pm 0.76$ | $\mathbf{78.61 \pm 0.34}$ | $\mathbf{73.51 \pm 0.47}$ | $\mathbf{89.73 \pm 0.37}$ |
| GIN-m (untrained) | $\mathbf{82.56 \pm 3.12}$ | $\mathbf{70.70 \pm 0.60}$ | $\mathbf{47.59 \pm 0.95}$ | $77.82 \pm 0.55$ | $73.45 \pm 0.30$ | $82.32 \pm 0.45$ |
| GraphConv-m (trained) | $\mathbf{81.62 \pm 2.08}$ | $59.14 \pm 1.93$ | $38.75 \pm 1.62$ | $\mathbf{63.28 \pm 0.6}$ | $71.49 \pm 0.67$ | $\mathbf{82.4 \pm 0.19}$ |
| GraphConv-m (untrained) | $78.03 \pm 1.57$ | $\mathbf{65.77 \pm 1.32}$ | $\mathbf{43.29 \pm 0.96}$ | $62.36 \pm 0.45$ | $\mathbf{71.83 \pm 0.42}$ | $77.15 \pm 0.29$ |

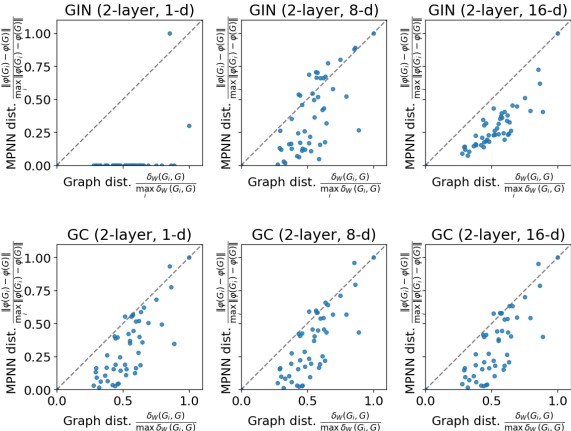

Figure 2: MPNNs preserve graph distance better when increasing the number of hidden dimensions. Comparatively, untrained GIN embeddings are more sensitive than untrained GraphConv to changes in the number of hidden dimensions.

necessary to use all MPNNs to preserve graph distance, which is impossible to compute in practice. Nonetheless, our experiment shows that using enough of them suffices for graph classification tasks.

We also investigate the effect of the number of layers, i.e., the 1-WL iteration in IDM. As shown in Figure 7 in the appendix, increasing the number of layers first improves and then degrades untrained MPNNs' performance, which is likely due to the changes in MPNNs' ability to preserve graph distance observed as in Figure 5 in the appendix.

## 7  Conclusion

This work devised a deeper understanding of MPNNs' capacity to capture graph structure, precisely determining when they learn similar vectorial representations. To that, we developed a comprehensive theory of graph metrics on graphons, demonstrating that two graphons are close in our metrics if, and only if, the outputs of all possible MPNNs are close, offering a more nuanced understanding of their ability to capture graph structure similarity. In addition, we established a connection between the continuous extensions of 1-WL and MPNNs to graphons, tree distance, and tree homomorphism densities. Our experimental study confirmed the validity of our theory in real-world prediction tasks. *In summary, our work establishes the first rigorous connection between the similarity of graphs and their learned vectorial presentations, paving the way for a more nuanced understanding of MPNNs' expressivity and robustness abilities and their connection to graph structure.*

Looking forward, future research could focus on extending all characterizations of Theorem 1 to graphons with signals. While we briefly sketched in Section 5 how to do this for MPNNs and the metrics we defined, this still presents a challenge as tree distance and tree homomorphism densities do not readily generalize to graphons with signals. A different direction could be an extension of our theory to different aggregation functions like max- or mean-aggregation. Since the 1-WL paradigm of summing over neighbors is crucial in our proofs, it is not clear how one would approach this. Additionally, further quantitative versions of equivalences in Theorem 1, not resorting to epsilon-variants statements, and generalizing our results to the $k$-WL are interesting avenues for future exploration.

## Acknowledgments and disclosure of funding

This research project was started at the BIRS 2022 Workshop "Deep Exploration of non-Euclidean Data with Geometric and Topological Representation Learning" held at the UBC Okanagan campus in Kelowna, B.C. Jan Böker is funded by the European Union (ERC, SymSim, 101054974). Views and opinions expressed are, however those of the author only and do not necessarily reflect those of the European Union or the European Research Council. Neither the European Union nor the granting authority can be held responsible for them. Ron Levie is partially funded by ISF (Israel Science Foundation) grant # 1937/23. Ningyuan Huang is partially supported by the MINDS Data Science Fellowship from Johns Hopkins University. Soledad Villar is partially funded by the NSF–Simons Research Collaboration on the Mathematical and Scientific Foundations of Deep Learning (MoDL) (NSF DMS 2031985), NSF CISE 2212457, ONR N00014-22-1-2126 and an Amazon AI2AI Faculty Research Award. Christopher Morris is partially funded by a DFG Emmy Noether grant (468502433) and RWTH Junior Principal Investigator Fellowship under Germany's Excellence Strategy.

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

# A  Additional background

Here, we provide additional background.

## A.1  The Weisfeiler–Leman algorithm and related topics

The 1-WL or color refinement is a well-studied heuristic for the graph isomorphism problem, originally proposed by Weisfeiler and Leman [114].[2] Intuitively, the algorithm determines if two graphs are non-isomorphic by iteratively coloring or labeling vertices. Given an initial coloring or labeling of the vertices of both graphs, e.g., their degree or application-specific information, in each iteration, two vertices with the same label get different labels if the number of identically labeled neighbors is unequal. These labels induce a vertex partition, and the algorithm terminates when at some iteration, the algorithm does not refine the current partition, i.e., when a *stable coloring* or *stable partition* is obtained. Then, if the number of vertices annotated with a specific label is different in both graphs, we can conclude that the two graphs are not isomorphic. It is easy to see that the algorithm cannot distinguish all non-isomorphic graphs [24]. Nonetheless, it is a powerful heuristic that can successfully test isomorphism for a broad class of graphs [10].

Formally, let $G = (V(G), E(G), \ell)$ be a labeled graph, i.e., a graph equipped with a (vertex-)label function $\ell \colon V(G) \to \mathbb{N}$. In each iteration, $t > 0$, the 1-WL computes a vertex coloring $C_t^1 \colon V(G) \to \mathbb{N}$, depending on the coloring of the neighbors. That is, in iteration $t > 0$, we set

$$C_t^1(v) \coloneqq \mathsf{RELABEL}\Big(\big(C_{t-1}^1(v), \{\!\{ C_{t-1}^1(u) \mid u \in N(v) \}\!\}\big)\Big),$$

for all vertices $v$ in $V(G)$, where RELABEL injectively maps the above pair to a unique natural number, which has not been used in previous iterations. In iteration 0, the coloring $C_0^1 \coloneqq \ell$. To test if two graphs $G$ and $H$ are non-isomorphic, we run the above algorithm in "parallel" on both graphs. If the two graphs have a different number of vertices colored $c$ in $\mathbb{N}$ at some iteration, the 1-WL *distinguishes* the graphs as non-isomorphic. Moreover, if the number of colors between two iterations, $t$ and $(t+1)$, does not change, i.e., the cardinalities of the images of $C_t^1$ and $C_{i+t}^1$ are equal, or, equivalently,

$$C_t^1(v) = C_t^1(w) \iff C_{t+1}^1(v) = C_{t+1}^1(w),$$

for all vertices $v$ and $w$ in $V(G)$, the algorithm terminates. For such $t$, we define the *stable coloring* $C_\infty^1(v) = C_t^1(v)$, for $v$ in $V(G)$. The stable coloring is reached after at most $\max\{|V(G)|, |V(H)|\}$ iterations [53]. The algorithm can be generalized to the $k$-WL, leading to a strict boost in expressive power in distinguishing non-isomorphic graphs.

**Properties of the Weisfeiler–Leman algorithm**  The Weisfeiler–Leman algorithm constitutes one of the earliest and most natural approaches to isomorphism testing [113, 114], and the theory community has heavily investigated it over the last few decades [53]. Moreover, the fundamental nature of the $k$-WL is evident from various connections to other fields such as logic, optimization, counting complexity, and quantum computing. The power and limitations of the $k$-WL can be neatly characterized in terms of logic and descriptive complexity [8, 62], Sherali-Adams relaxations of the natural integer linear optimization problem for the graph isomorphism problem [5, 57, 80], homomorphism counts [34], quantum isomorphism games [6], graph decompositions [68]. In their seminal paper, Cai et al. [24] showed that, for each $k$, a pair of non-isomorphic graphs of size $\mathcal{O}(k)$ exists not distinguished by the $k$-WL. Kiefer [66] thoroughly surveys more background and related results concerning the expressive power of the $k$-WL. For $k = 1$, the power of the algorithm has been completely characterized [3, 70]. Moreover, upper bounds on the running time [15] and the number of iterations for $k = 1$ [67], for the folklore $k = 2$ [69, 76], and general $k$ [59] have been shown. For $k$ in $\{1, 2\}$, Arvind et al. [4] studied the abilities of the folklore $k$-WL to detect and count fixed subgraphs, extending the work of Fürer [42]. The former was refined in [29]. Kiefer et al. [71] showed that the folklore 3-WL completely captures the structure of planar graphs. The algorithm for logarithmic $k$ plays a prominent role in the recent result of [9] improving the best-known running time for the graph isomorphism problem. Recently, Grohe et al. [58] introduced the framework of Deep Weisfeiler–Leman algorithms, which allow the design of a more powerful graph isomorphism test than Weisfeiler–Leman type algorithms. Finally, the emerging

---

[2]Strictly speaking, the 1-WL and color refinement are two different algorithms. That is, the 1-WL considers neighbors and non-neighbors to update the coloring, resulting in a slightly higher expressive power when distinguishing vertices in a given graph; see [55] for details. For brevity, we consider both algorithms to be equivalent.

connections between the Weisfeiler–Leman paradigm and graph learning are described in two recent surveys [54, 92].

**Fractional isomorphisms**   A matrix $X \in \mathbb{R}^{I \times J}$ is called *doubly stochastic* if all entries are nonnegative and we have $\sum_{i \in I} X_{ij} = 1$ for every $j \in J$ and $\sum_{j \in J} X_{ij} = 1$ for every $i \in I$, i.e., all columns and rows sum to 1. Note that such a matrix is necessarily square. A *fractional isomorphism* between graphs $G$ and $H$ is a doubly stochastic matrix $X \in \mathbb{R}^{V(G) \times V(H)}$ such that $A_G X = X A_H$, where $A_G$ and $A_H$ are the adjacency matrices of $G$ and $H$, respectively. $G$ and $H$ are called *fractionally isomorphic* if there is a fractional isomorphism between $G$ and $H$. A result due to Tinhofer [109, 110] states that the 1-WL test does not distinguish $G$ and $H$ if and only if they are fractionally isomorphic.

**Graph homomorphisms**   Given two graphs $F$ and $G$, a *homomorphism* from $F$ to $G$ is a mapping $h \colon V(F) \to V(G)$ such that $h(u)h(v) \in E(G)$ for every $uv \in E(F)$. Let $\hom(F, G)$ denote the number of homomorphisms from $F$ to $G$. Then, $t(F, G) := \hom(F, G) / |V(G)|^{|V(F)|}$ is called the *homomorphism density of $F$ in $G$*. We have $t(F, G) = t(F, W_G)$ for the graphon $W_G$ induced by $G$. A result due to Dvořák [37] and Dell et al. [34] states that the 1-WL test does not distinguish $G$ and $H$ if and only if we have $t(T, G) = t(T, H)$ for every tree $T$.

## A.2   Message-passing graph neural networks

Intuitively, MPNNs learn a vectorial representation, i.e., a $d$-dimensional real-valued vector, representing each vertex in a graph by aggregating information from neighboring vertices. Formally, let $G$ be graph with initial vertex features $\mathbf{h}_v^{(0)}$ in $\mathbb{R}^d$ for $v \in V(G)$. An MPNN architecture consists of a stack of neural network layers, i.e., a composition of permutation-invariant parameterized functions. Similarly to the 1-WL, each layer aggregates local neighborhood information, i.e., the neighbors' features, around each vertex and then passes this aggregated information on to the next layer. Following Gilmer et al. [50] and Scarselli et al. [105], in each layer, $t > 0$, we compute vertex features

$$\mathbf{h}_v^{(t)} := \mathsf{UPD}^{(t)} \Big( \mathbf{h}_v^{(t-1)}, \mathsf{AGG}^{(t)} \big( \{\!\!\{ \mathbf{h}_u^{(t-1)} \mid u \in N(v) \}\!\!\} \big) \Big) \in \mathbb{R}^d,$$

where $\mathsf{UPD}^{(t)}$ and $\mathsf{AGG}^{(t)}$ may be differentiable parameterized functions, e.g., neural networks.[3] In the case of graph-level tasks, e.g., graph classification, one uses

$$\mathbf{h}_G := \mathsf{READOUT} \big( \{\!\!\{ \mathbf{h}_v^{(L)} \mid v \in V(G) \}\!\!\} \big) \in \mathbb{R}^d,$$

to compute a single vectorial representation based on learned vertex features after iteration $L$. Again, READOUT may be a differentiable parameterized function. To adapt the parameters of the above three functions, they are optimized end-to-end, usually through a variant of stochastic gradient descent, e.g., [72], together with the parameters of a neural network used for classification or regression.

## A.3   Topology

We recall some basics from Dudley [35]. Given a set $X$, a *topology* on $X$ is a collection $\mathcal{T} \subseteq 2^X$ of subsets of $X$ with $\varnothing, X \in \mathcal{T}$ such that $\mathcal{T}$ is closed under finite intersections and arbitrary unions. Then, the elements of $\mathcal{T}$ are called *open sets* and $(X, \mathcal{T})$ is called a *topological space*. Given two topological spaces, $(X, \mathcal{T})$ and $(Y, \mathcal{U})$, a function $f \colon X \to Y$ is called *continuous* if $f^{-1}(U) \in \mathcal{T}$ for every $U \in \mathcal{U}$. If $S$ and $I$ are any sets and $f_i \colon S \to X_i$ a function, where $(X_i, \mathcal{T}_i)$ is a topological space, for every $i \in I$, then there is a smallest topology $\mathcal{T}$ on $S$ for which every $f_i$ is continuous, called *the topology generated by the $f_i$*. Given a topological space $(X, \mathcal{T})$, one can define *nets* and the notion of *convergence* of a net; nets do in general topological spaces what sequences do in metric spaces.

Given a pseudometric space $(X, d)$, the collection $\mathcal{T}$ of all unions of open balls $B(x, r) := \{y \in X \mid d(x, y) < r\}$, where $x \in X$ and $r > 0$, is a topology, and $\mathcal{T}$ is called *metrizable* and *metrized* by $d$. If $\mathcal{T}$ is metrizable, then a set $U \subseteq S$ is open if and only if for every $x \in U$ and sequence $x_i \to x$, there is some $j$ with $x_i \in U$ for every $i \geq j$. Hence, two metrizable topologies are equal if they have the same convergent sequences. A topological space $(X, \mathcal{T})$ is called *Hausdorff* if, for all $x \neq y$ in $X$, there are

---

[3]Strictly speaking, Gilmer et al. [50] consider a slightly more general setting in which vertex features are computed by $\mathbf{h}_v^{(t)} := \mathsf{UPD}^{(t)} \Big( \mathbf{h}_v^{(t-1)}, \mathsf{AGG}^{(t)} \big( \{\!\!\{ (\mathbf{h}_v^{(t-1)}, \mathbf{h}_u^{(t-1)}, l(v, u)) \mid u \in N_G(v) \}\!\!\} \big) \Big)$.

open sets $U$ and $V$ with $x \in U$, $y \in V$, and $U \cap V = \varnothing$. A pseudometric space $(X, d)$ is Hausdorff if and only if $d$ is a metric. A topological space $(X, \mathcal{T})$ is called *separable* if $X$ has a countable dense subset, where a subset of $X$ is called *dense* if its closure is $X$.

A topological space $(X, \mathcal{T})$ is called *compact* if whenever $\mathcal{U} \subseteq \mathcal{T}$ and $X = \bigcup \mathcal{U}$, there is a finite $\mathcal{V} \subseteq \mathcal{U}$ such that $X = \bigcup \mathcal{V}$. If $(X, \mathcal{T})$ is a compact topological space and $f \colon X \to Y$ a surjective continuous function from $X$ to another topological space $(Y, \mathcal{U})$, then $(Y, \mathcal{U})$ is compact. By Tychonoff's theorem, the product of any collection of compact topological spaces is compact w.r.t. the product topology. A metric space $(X, d)$ is compact if and only if $(X, d)$ is complete and totally bounded. Every compact metric space is separable.

Given a set $X$, a $\sigma$-*algebra* is a collection $\mathcal{A} \subseteq 2^X$ of subsets of $X$ with $\varnothing, X \in \mathcal{A}$ such that $\mathcal{A}$ is closed under complements and countable unions. Then, $(X, \mathcal{A})$ is called a *measurable space*. For every collection $\mathcal{C} \subseteq 2^X$, there is a smallest $\sigma$-algebra including $\mathcal{C}$, called *the $\sigma$-algebra generated by $\mathcal{C}$*. A $\sigma$-algebra that is generated by a topology is called a *Borel $\sigma$-algebra*; its elements are called *Borel sets*. A measurable space $(X, \mathcal{B})$ is called a *standard Borel space* if there is a separable completely metrizable topology $\mathcal{T}$ on $X$ such that $\mathcal{B}$ is the $\sigma$-algebra generated by $\mathcal{T}$. Here, a topology is called completely metrizable if there is a metric $d$ such that $(X, d)$ is a complete metric space and $d$ metrizes $\mathcal{T}$. Hence, every compact metric space with its Borel $\sigma$-algebra is a standard Borel space.

Let $(X, \mathcal{A})$ be a measurable space. A function $\mu \colon \mathcal{A} \to [0, \infty]$ is called a *measure* if $\mu(\varnothing) = 0$ and it is countably additive, i.e., if whenever $A_n \in \mathcal{A}$ for $n = 1, 2, \dots$ are pairwise disjoint, then $\mu(\bigcup_{n \geq 1} A_n) = \sum_{n \geq 1} \mu(A_n)$. The measure $\mu$ is called *finite* if $\mu(X) < \infty$.

## A.4  The Stone–Weierstrass theorem

The Stone–Weierstrass theorem is the primary tool we use for proving our universal approximation theorem. Here, we state the formulation from [35, Theorem 2.4.11]. An *algebra* is a vector space that is additionally closed under multiplication. Recall that, for a compact Hausdorff space $K$, we denote by $C(K, \mathbb{R})$ the set of all real-valued continuous functions on $K$ and $d_{\sup}(f, g) \coloneqq \sup_{x \in K} |f(x) - g(x)|$ for $f, g \in C(K, \mathbb{R})$.

**Theorem 7** (Stone–Weierstrass theorem)**.** Let $K$ be a compact Hausdorff space and let $\mathcal{F}$ be an algebra included in $C(K, \mathbb{R})$ such that $\mathcal{F}$ separates points and contains the constants. Then, $\mathcal{F}$ is dense in $C(K, \mathbb{R})$ for $d_{\sup}$.

## A.5  The cut distance

The *cut norm* of a kernel $U$ is $\|U\|_{\square} \coloneqq \sup_{S, T} |\int_{S \times T} W(x, y)\, dx dy|$, where the supremum is taken over all measurable sets $S, T \subseteq [0, 1]$. The cut distance of two graphons $U$ and $W$ is defined by $\delta_{\square}(U, W) \coloneqq \inf_{\varphi} \|U^{\varphi} - W\|_{\square}$, where the infimum is taken over all invertible measure-preserving maps $\varphi \colon [0, 1] \to [0, 1]$ and $U^{\varphi}(x, y) \coloneqq U(\varphi(x), \varphi(y))$ for all $x, y \in X$. See the book of Lovász [78] for a thorough exposition.

## A.6  Distributions on iterated degree measures

Grebík and Rocha [52] give the following definition of a DIDM, which we slightly adapt to our definitions. The Kolmogorov Consistency Theorem yields that, for every $\alpha \in \mathbb{M}$, there is a unique $\mu_{\alpha} \in \mathcal{M}_{\leq 1}(\mathbb{M})$ such that $(p_{\infty, h})_* \mu_{\alpha} = \alpha_{h+1}$ for every $h \in \mathbb{N}$. Then, a probability measure $\nu \in \mathscr{P}(\mathbb{M})$ is called a *distribution on iterated degree measures* (DIDM) if $\mu_{\alpha}$ is absolutely continuous w.r.t. $\nu$ with the Radon-Nikodym derivative satisfying $0 \leq \frac{d\mu_{\alpha}}{d\nu} \leq 1$ for $\nu$-almost every $\alpha$; intuitively, this ensures that, if we have some mass $m$ of points of color $c$, then every point in the graph may have at most $m$ neighbors of color $c$. Then, $\nu_W$ for a graphon $W$ satisfies this definition, and conversely, that every DIDM defines a kernel on $\mathbb{M} \times \mathbb{M}$ that is bounded by one almost everywhere.

# B  Extended related work

Here, we provided an extended discussion on related work.

## B.1 Limitations of MPNNs and the 1-WL

Recently, connections between GNNs and Weisfeiler–Leman type algorithms have been shown [12, 49, 89, 115]. Specifically, Morris et al. [89] and Xu et al. [115] showed that the 1-WL limits the expressive power of any possible GNN architecture in distinguishing non-isomorphic graphs. In turn, these results have been generalized to the $k$-WL, see, e.g., Azizian and Lelarge [7], Geerts [47], Maron et al. [81], Morris et al. [89, 91, 93], and connected to permutation-equivariant functions approximation over graphs, see, e.g., Azizian and Lelarge [7], Chen et al. [28], Geerts and Reutter [48], Maehara and NT [79]. Further, Aamand et al. [1] devised an improved analysis using randomization. Recent works have extended the expressive power of GNNs, e.g, by using random features [2, 32, 104], equivariant graph polynomials [100], or homomorphism and subgraph counts [13, 22, 30, 95], see Morris et al. [92] for a thorough survey. Recently, Grohe [56] showed tight connections between GNNs' expressivity and circuit complexity. Moreover, Rosenbluth et al. [102] investigated the expressive power of different aggregation functions beyond sum aggregation.

## B.2 Graph metrics beyond edit distance

As discussed before, the edit distance does not capture the structural similarity of graphs well. A metric that arguably captures the similarity of graphs well is the *cut distance*, which was introduced by Frieze and Kannan [41] and provides the central notion of convergence in the theory of dense graph limits. Graph similarity can also be defined via graph homomorphism densities, and as part of the aforementioned theory, Borgs et al. [20], Lovász and Szegedy [77] proved that this is equivalent to the cut distance. Böker [18] introduced a variant of the cut distance based on the well-known system of linear equations defining fractional isomorphism, the *tree distance*, which is equivalent to similarity defined via tree homomorphism densities. To study MPNNs' stability properties, Chuang and Jegelka [31] introduced *Tree Mover's Distance* to compare two graphs via hierarchical optimal transport between their computation trees. Similarly, Chen et al. [26] introduced the *Weisfeiler–Leman distance*, a polynomial-time computable pseudometric on graphs by combining techniques from optimal transport with the 1-WL, and gave an interpretation in terms of stochastic processes [27].

# C  Proofs

Here, we collect proofs that we omitted from the main body of the paper.

## C.1  Equivalence of message-passing graph neural networks

Here, we show that, first, for a graph $G$, the output of an MPNN on $G$ equals the output of the MPNN on the corresponding graphon $W_G$ and, second, the output of an MPNN on a graphon $W$ equals the output of the corresponding MPNN on the corresponding DIDM $\nu_W$ of $W$. Hence, it suffices to consider MPNNs on DIDMs. We note that, while $\varphi$ is an MPNN model and $\psi$ is Lipschitz in the following theorems, we actually do not need Lipschitz continuity and continuous functions would suffice.

**Theorem 8.** Let $G$ be a graph and $\varphi$ be an $L$-layer MPNN model. Let $(I_v)_{v \in V(G)}$ be the partition of $[0,1]$ used in the construction of $W_G$ from $G$. Then, for all $t \in [L]$, $v \in V(G)$, and $x \in I_v$,

$$\mathbf{h}_v^{(t)} = \mathbf{h}_x^{(t)}.$$

*Proof.* We prove the claim by induction on $t$.

*Base of the induction.* The first space of colors is $\mathbb{M}_0 = \{1\}$. Then, for all $v \in V(G)$ and $x \in I_v$,

$$\mathbf{h}_v^{(0)} = \varphi_0 = \mathbf{h}_x^{(0)}.$$

*Induction step.* The induction assumption is that $\mathbf{h}_v^{(t-1)} = \mathbf{h}_x^{(t-1)}$ for all $v \in V(G)$ and $x \in I_v$. Then, for all $v \in V(G)$ and $x \in I_v$, we have

$$\mathbf{h}_x^{(t)} = \varphi_t\left(\int_{[0,1]} W_G(x,y)\mathbf{h}_y^{(t-1)}\, d\lambda(y)\right) = \varphi_t\left(\sum_{u \in V(G)} \int_{I_u} W_G(x,y)\mathbf{h}_y^{(t-1)}\, d\lambda(y)\right)$$

$$= \varphi_t\left(\sum_{u \in V(G)} \int_{I_u} W_G(x,y)\mathbf{h}_u^{(t-1)}\, d\lambda(y)\right)$$

$$= \varphi_t\left(\frac{1}{|V(G)|} \sum_{u \in N(v)} \mathbf{h}_u^{(t-1)}\right) = \mathbf{h}_v^{(t)}.$$

$\square$

**Theorem 9.** Let $G$ be a graph, let $\varphi$ be an $L$-layer MPNN model, and let $\psi$ be Lipschitz. Let $(I_v)_{v \in V(G)}$ be the partition of $[0,1]$ used in the construction $W_G$ from $G$. Then,

$$\mathbf{h}_G = \mathbf{h}_{W_G}.$$

*Proof.* With Theorem 8, we get

$$\mathbf{h}_{W_G} = \psi\left(\int_{[0,1]} \mathbf{h}_x^{(L)}\, d\lambda(x)\right) = \psi\left(\sum_{v \in V(G)} \int_{I_v} \mathbf{h}_x^{(L)}\, d\lambda(x)\right)$$

$$= \psi\left(\sum_{v \in V(G)} \int_{I_v} \mathbf{h}_v^{(L)}\, d\lambda(x)\right)$$

$$= \psi\left(\frac{1}{|V(G)|} \sum_{v \in V(G)} \mathbf{h}_v^{(L)}\right) = \mathbf{h}_G.$$

$\square$

To prove the equivalence of an MPNN on a graphon $W$ and the corresponding DIDM $\nu_W$, we need the following definition. Let $\delta \colon [0,1] \to \mathbb{R}$ be a nonnegative measurable function. Then, define the measure $\nu_\delta$ by

$$\nu_\delta(A) := \int_A \delta\, d\lambda$$

for measurable $A \subseteq [0,1]$. The following lemma, which can be interpreted as the "well-definedness of the change of variable $z = \delta(y)$", is taken from [16, Theorem 16.11].

**Lemma 10.** Let $\delta : [0, 1] \to \mathbb{R}$ be a nonnegative measurable function. Then, a measurable function $f : [0, 1] \to \mathbb{R}$ is integrable with respect to $\nu_\delta$ if and only if $f\delta$ is integrable with respect to $\lambda$, in which case

$$\int_A f \, d\nu_\delta = \int_A f\delta \, d\lambda$$

holds for every measurable $A$.

**Theorem 11.** Let $W \in \mathcal{W}$ and $\varphi$ be an $L$-layer MPNN model. Then, for every $t \in [L]$ and almost every $x \in [0, 1]$,

$$\mathbf{h}_x^{(t)} = \mathbf{h}_{i_{W,t}(x)}^{(t)} = \mathbf{h}_{i_W(x)}^{(t)}.$$

*Proof.* We only have to show the first equality. Then, the second follows directly from the definitions of $\mathbf{h}_-^{(t)} : \mathbb{M} \to \mathbb{R}^{d_t}$ and $i_W$.

*Base of the induction.* The first space of colors is $\mathbb{M}_0 = \{1\}$. Then,

$$\mathbf{h}_x^{(0)} = \varphi_0 = \mathbf{h}_{i_{W,t}(x)}^{(0)}.$$

*Induction step.* The induction assumption is that $\mathbf{h}_x^{(t-1)} = \mathbf{h}_{i_{W,t-1}(x)}^{(t-1)}$ for almost every $x \in [0, 1]$. Let us prove the induction step for $t$. We have

$$\mathbf{h}_x^{(t)} = \varphi_t \left( \int_{[0,1]} W(x, y) \mathbf{h}_y^{(t-1)} \, d\lambda(y) \right).$$

Recall that for any $A \in \mathcal{B}(\mathbb{M}_{h-1})$,

$$\left( i_{W,t}(x) \right)(A) = \int_{i_{W,t-1}^{-1}(A)} W(x, y) \, d\lambda(y),$$

so, by the notation $\nu_{W(x,-)}$ introduced before Lemma 10,

$$i_{W,t}(x) = (i_{W,t-1})_* \nu_{W(x,-)}.$$

Hence, by Lemma 10 with $\delta = W(x, -)$ and $f = \mathbf{h}_-^{(t-1)}$, we have

$$\mathbf{h}_{i_{W,t}(x)}^{(t)} = \varphi_t \left( \int_{\mathbb{M}_{t-1}} \mathbf{h}_-^{(t-1)} \, d i_{W,t}(x) \right) = \varphi_t \left( \int_{\mathbb{M}_{t-1}} \mathbf{h}_-^{(t-1)} \, d(i_{W,t-1})_* \nu_{W(x,-)} \right)$$

$$= \varphi_t \left( \int_{[0,1]} \mathbf{h}_-^{(t-1)} \circ i_{W,t-1} \, d\nu_{W(x,-)} \right)$$

$$= \varphi_t \left( \int_{[0,1]} W(x, y) \mathbf{h}_{i_{W,t-1}(y)}^{(t-1)} \, d\lambda(y) \right).$$

Hence, by the induction assumption, $\mathbf{h}_x^{(t)} = \mathbf{h}_{i_{W,t}(x)}^{(t)}$. $\qquad \square$

**Theorem 12.** Let $W \in \mathcal{W}$, let $\varphi$ be an $L$-layer MPNN model, and let $\psi$ be Lipschitz. Then,

$$\mathbf{h}_W = \mathbf{h}_{\nu_{W,L}} = \mathbf{h}_{\nu_W}.$$

*Proof.* We only have to prove the first inequality; the second follows from the first and the definition of $\nu_W$. The first inequality follows from Theorem 11:

$$\mathbf{h}_{\nu_{W,L}} = \psi \left( \int_{\mathbb{M}_L} \mathbf{h}_-^{(L)} \, d\nu_{W,L} \right) = \psi \left( \int_{\mathbb{M}_L} \mathbf{h}_-^{(L)} \, d(i_{W,L})_* \lambda \right)$$

$$= \psi \left( \int_{[0,1]} \mathbf{h}_{i_{W,L}(x)}^{(L)} \, d\lambda(x) \right)$$

$$= \psi \left( \int_{[0,1]} \mathbf{h}_x^{(L)} \, d\lambda(x) \right)$$

$$= \mathbf{h}_W.$$

$\qquad \square$

## C.2 Metrics on iterated degree measures

We first prove Theorem 2 for the metrics based on the Prokhorov metric. Then, we prove the inequalities between the Prokhorov metric P and the unbalanced Wasserstein metric W from Section 3 and deduce Theorem 2 for the unbalanced Wasserstein metric from it. After that, we prove Lemma 3.

### C.2.1 The Prokhorov metric

To begin, let us restate Theorem 2 for better readability.

**Theorem 2.** Let $0 \leq h \leq \infty$. The metrics $d_{\mathsf{P},h}$ and $d_{\mathsf{W},h}$ are well-defined and metrize the topology of $\mathbb{M}_h$. The metrics $\delta_{\mathsf{P},h}$ and $\delta_{\mathsf{W},h}$ are well-defined and metrize the topology of $\mathscr{P}(\mathbb{M}_h)$. Moreover, these metrics are computable on graphs in time polynomial in the size of the input graphs and $h$, up to an additive error of $\varepsilon$ in the case of $d_{\mathsf{W},\infty}$ and $\delta_{\mathsf{W},\infty}$.

Let $(S, d)$ be a complete separable metric space with Borel $\sigma$-algebra $\mathcal{B}$. Recall that, for a subset $A \subseteq S$ and $\varepsilon \geq 0$, one defines $A^\varepsilon := \{y \in S \mid d(x, y) < \varepsilon \text{ for some } x \in A\}$. Then, the Prokhorov metric P on $\mathscr{M}_{\leq 1}(S, \mathcal{B})$ is given by

$$\mathsf{P}(\mu, \nu) := \inf\{\varepsilon > 0 \mid \mu(A) \leq \nu(A^\varepsilon) + \varepsilon \text{ and } \nu(A) \leq \mu(A^\varepsilon) + \varepsilon \text{ for every } A \in \mathcal{B}\}.$$

It is not hard to see that we can replace $A^\varepsilon$ by $A^{\varepsilon]} := \{y \in S \mid d(x, y) \leq \varepsilon \text{ for some } x \in A\}$ in the above definition without changing the value $\mathsf{P}(\mu, \nu)$. Moreover, if $\|\mu\| \geq \|\nu\|$, then the definition simplifies to

$$\mathsf{P}(\mu, \nu) = \inf\{\varepsilon > 0 \mid \mu(A) \leq \nu(A^\varepsilon) + \varepsilon \text{ for every } A \in \mathcal{B}\}$$

since, if $\mu(A) \leq \nu(A^\varepsilon) + \varepsilon$ for every $A \in \mathcal{B}$, then $\nu(B) \leq \mu(B^\varepsilon) + \varepsilon + \|\nu\| - \|\mu\| \leq \mu(B^\varepsilon) + \varepsilon$ for every $B \in \mathcal{B}$ [40, Lemma 4.30].

As the name suggests, P is a metric on $\mathscr{M}(S)$ [99, Section 1.4], and Prokhorov also proved that convergence in P is equivalent to convergence in the weak topology.

**Theorem 13** ([99, Theorem 1.11])**.** Let $(S, d)$ be a complete separable metric space. Then, $(\mathscr{M}(S), \mathsf{P})$ is a complete separable metric space, and convergence in P is equivalent to weak convergence of measures.

The well-definedness of $d_{\mathsf{P},h}$ and $\delta_{\mathsf{P},h}$ follows from an inductive application of this theorem.

**Lemma 14.** Let $0 \leq h \leq \infty$. The metric $d_{\mathsf{P},h}$ is well-defined and metrizes the topology of $\mathbb{M}_h$. The metric $\delta_{\mathsf{P},h}$ is well-defined and metrizes the topology of $\mathscr{P}(\mathbb{M}_h)$.

*Proof.* First, we consider $d_{\mathsf{P},h}$. Let $h \in \mathbb{N}$. To see why we even have to prove that $d_{\mathsf{P},h}$ is well-defined, note that the measures in $\mathbb{M}_{h+1} = \mathscr{M}_{\leq 1}(\mathbb{M}_h) = \mathscr{M}_{\leq 1}(\mathbb{M}_h, \mathcal{B}(\mathbb{M}_h))$ by definition are functions $\mu \colon \mathcal{B}(\mathbb{M}_h) \to \mathbb{R}$, where $\mathcal{B}(\mathbb{M}_h)$ depends on the topology of $\mathbb{M}_h$, which is the weak topology in our case. Hence, it is crucial that $d_{\mathsf{P},h}$ metrizes the weak topology on $\mathbb{M}_h$; otherwise the sets $\mathscr{M}_{\leq 1}(\mathbb{M}_h, \mathcal{B}(\mathbb{M}_h))$ and $\mathscr{M}_{\leq 1}(\mathbb{M}_h, d_{\mathsf{P},h})$ might not even be the same.

For the induction basis $h = 0$, this claim trivially holds. Then, the induction hypothesis yields that $d_{\mathsf{P},h}$ is well-defined and metrizes the weak topology on $\mathbb{M}_h$. Then, $d_{\mathsf{P},h+1}$ is well-defined as the Borel $\sigma$-algebra of $(\mathbb{M}_h, d_{\mathsf{P},h})$ equals $\mathcal{B}(\mathbb{M}_h)$ and, by Theorem 13, convergence in $d_{\mathsf{P},h+1}$ is equivalent to weak convergence on $\mathscr{M}_{\leq 1}(\mathbb{M}_h, d_{\mathsf{P},h})$, which is just weak convergence on $\mathscr{M}_{\leq 1}(\mathbb{M}_h, \mathcal{B}(\mathbb{M}_h)) = \mathbb{M}_{h+1}$. Hence, the topology induced by $d_{\mathsf{P},h+1}$ is equal to the weak topology on $\mathbb{M}_{h+1}$ as both spaces are metrizable. For $d_{\mathsf{P},\infty}$, the claim directly follows since $d_{\mathsf{P},\infty}$ is just a variant of the product metric, which metrizes the product topology.

The claim for $\delta_{\mathsf{P},h}$ then follows from the just proven claim for $d_{\mathsf{P},h}$ by the same reasoning as in the inductive step. $\qquad\square$

The following lemma states that the distances $d_{\mathsf{P},h}$ behave as expected, i.e., the distance of IDMs can only increase when $h$ increases.

**Lemma 15.** Let $0 \leq h \leq h' < \infty$. Then, $d_{\mathsf{P},h}(p_{h',h}(\alpha), p_{h',h}(\beta)) \leq d_{\mathsf{P},h'}(\alpha, \beta)$ for all $\alpha, \beta \in \mathbb{M}_h$.

*Proof.* It suffices to show that

$$d_{\mathsf{P},h}(p_{h+1,h}(\alpha), p_{h+1,h}(\beta)) \leq d_{\mathsf{P},h+1}(\alpha, \beta),$$

for all $\alpha, \beta \in \mathbb{M}_{h+1}$, where $h \geq 0$. We prove this by induction on $h$, where for $h = 0$, the statement trivially holds. For the inductive step, we assume without loss of generality that $\|\alpha\| \geq \|\beta\|$, which directly implies $\|p_{h+1,h}(\alpha)\| \geq \|p_{h+1,h}(\beta)\|$. We show that if $\varepsilon > 0$ such that $\alpha(A) \leq \beta(A^\varepsilon) + \varepsilon$ for every $A \in \mathcal{B}(\mathbb{M}_{h+1})$, then also $p_{h+1,h}(\alpha)(A) \leq p_{h+1,h}(\beta)(A^\varepsilon) + \varepsilon$. For such an $\varepsilon$, we have

$$
\begin{aligned}
p_{h+1,h}(\alpha)(A) = (p_{h,h-1})_* \alpha(A) &= \alpha(p_{h,h-1}^{-1}(A)) \\
&\leq \beta((p_{h,h-1}^{-1}(A))^\varepsilon) + \varepsilon \\
&\leq \beta(p_{h,h-1}^{-1}(A^\varepsilon)) + \varepsilon = p_{h+1,h}(\beta)(A^\varepsilon) + \varepsilon,
\end{aligned}
$$

where $\beta((p_{h,h-1}^{-1}(A))^\varepsilon) \leq \beta(p_{h,h-1}^{-1}(A^\varepsilon))$ holds by the induction hypothesis: We show that $(p_{h,h-1}^{-1}(A))^\varepsilon \subseteq p_{h,h-1}^{-1}(A^\varepsilon)$. Let $x \in (p_{h,h-1}^{-1}(A))^\varepsilon$. Then, there is a $y \in (p_{h,h-1}^{-1}(A))$ such that $d_{\mathsf{P},h}(x, y) < \varepsilon$. By the inductive hypothesis, we have

$$
d_{\mathsf{P},h-1}(p_{h,h-1}(x), p_{h,h-1}(y)) \leq d_{\mathsf{P},h}(x, y) < \varepsilon.
$$

Since $p_{h,h-1}(y) \in A$, we have $p_{h,h-1}(x) \in A^\varepsilon$, and thus, $x \in p_{h,h-1}^{-1}(A)$. $\qquad \square$

### C.2.2 Computability of the Prokhorov metric

It remains to prove that $d_{\mathsf{P},h}$ and $\delta_{\mathsf{P},h}$ are polynomial-time computable. A paper of Garel and Massé [46] is concerned with computing the Prokhorov metric of (possibly non-discrete) probability distributions. To this end, they first quantize the distributions and then show that the Prokhorov metric of finitely-supported probability distributions can be computed exactly by solving a series of maximum-flow problems; this latter part is based on results by Schay [106] and García-Palomares and Evarist Giné [45], which essentially shows how the Prokhorov metric can be formulated as a linear-optimization problem. By generalizing this linear program to finite measures, we can adapt the algorithm of Garel and Massé [46] to obtain an algorithm for computing the Prokhorov distance of finite measures.

**Theorem 16.** Let $\mu, \nu \in \mathcal{M}(S)$, where $(S, d)$ is a finite metric space with $S = \{x_1, \ldots, x_n\}$. Then, the Prokhorov metric $\mathsf{P}(\mu, \nu)$ can be computed in time polynomial in $n$ and the number of bits needed to encode $d$, $\mu$, and $\nu$.

*Proof.* Without loss of generality, we assume that $\|\mu\| \geq \|\nu\|$. Then,

$$
\begin{aligned}
\mathsf{P}(\mu, \nu) &= \inf\{\varepsilon > 0 \mid \mu(A) \leq \nu(A^\varepsilon) + \varepsilon \text{ for every } A \subseteq S\} \\
&= \inf\{\varepsilon > 0 \mid \mu(A) \leq \nu(A^{\varepsilon]}) + \varepsilon \text{ for every } A \subseteq S\} \\
&= \inf\{\varepsilon \geq 0 \mid \varepsilon \geq \rho(\varepsilon)\},
\end{aligned}
$$

where

$$
\rho(\varepsilon) := \inf\{\eta > 0 \mid \mu(A) \leq \nu(A^{\varepsilon]}) + \eta \text{ for every } A \subseteq S\}
$$

for $\varepsilon > 0$. The following claim generalizes an observation of Schay [106, Theorem 1] and García-Palomares and Evarist Giné [45, Lemma] to finite measures and shows that the value of $\rho(\varepsilon)$ can be expressed as a linear program.

**Claim 17.** Let $\varepsilon > 0$. Then,

$$
\rho(\varepsilon) = \|\mu\| - \max \sum_{i,j=1}^n \mathbb{1}[d(x_i, x_j) \leq \varepsilon] \cdot x_{ij},
$$

where the maximum is taken over variables $x_{ij}$ such that $\sum_{i=1}^n x_{ij} = \nu(\{x_j\})$ for every $j \in [n]$, $\sum_{j=1}^n x_{ij} \leq \mu(\{x_i\})$ for every $i \in [n]$, and $x_{ij} \geq 0$ for all $i, j \in [n]$.

*Proof.* For the sake of brevity, define $p_i := \mu(\{x_i\})$ and $q_i := \nu(\{x_i\})$ for $i \in [n]$ and $d_{ij}^\varepsilon := \mathbb{1}[d(x_i, x_j) \leq \varepsilon]$ for all $i, j \in [n]$. We have

$$
\begin{aligned}
\rho(\varepsilon) = \max_{A \subseteq S}\big(\mu(A) - \nu(A^{\varepsilon]})\big) &= \max_{\substack{v_j \in \{0,1\}, \\ w_i \in \{0,1\}, \\ v_j \geq d_{ij}^\varepsilon - 1 + w_i}} \sum_{i \in [n]} (p_i w_i - q_i v_i) \\
&= \max_{\substack{0 \leq v_j \leq 1, \\ 0 \leq w_i \leq 1, \\ v_j \geq d_{ij}^\varepsilon - 1 + w_i}} \sum_{i \in [n]} (p_i w_i - q_i v_i)
\end{aligned}
$$

where the first equality holds by definition of $\rho(\varepsilon)$, the second since one can view the sets $A$ and $A^\varepsilon$ as vectors $\bar{v}$ and $\bar{w}$, respectively, and the third, since the optimum value of a linear program is attained on an extreme point of the convex polytope specifying the feasible solutions, which are 0-1-vectors for this specific linear program. Moreover, since values $v_j > 1$ and $w_i < 0$ can always be set to 1 and 0, respectively, without decreasing the value of the target function, this further equals to

$$\max_{\substack{v_j \geq 0, \\ w_i \leq 1, \\ v_j \geq d_{ij}^\varepsilon - 1 + w_i}} \sum_{i \in [n]} (p_i w_i - q_i v_i) = \|\mu\| - \min_{\substack{u_i \geq 0, \\ v_j \geq 0, \\ u_i + v_j \geq d_{ij}^\varepsilon}} \sum_{i \in [n]} (p_i u_i + q_i v_i),$$

where the second equality results from the substitution $u_i := 1 - w_i$. Then, considering the dual to the linear minimization problem yields that this further equals to

$$\rho(\varepsilon) = \|\mu\| - \max \sum_{i,j=1}^{n} d_{ij}^\varepsilon \cdot x_{ij},$$

where the maximum is taken over variables $x_{ij}$ such that $\sum_{i=1}^n x_{ij} \leq q_j$ for every $j \in [n]$, $\sum_{j=1}^n x_{ij} \leq p_i$ for every $i \in [n]$, and $x_{ij} \geq 0$ for all $i, j \in [n]$. Since we can always increase the variables $x_{ij}$ such that $\sum_{i=1}^n x_{ij} = q_j$ for every $j \in [n]$ holds without decreasing the target function, we obtain the claim. ∎

With this claim, we can use a polynomial-time method for solving linear programs to compute $\rho(\varepsilon)$ in time polynomial in $n$ and the number of bits needed to encode $d$, $\mu$, and $\nu$. Alternatively, it is straightforward to express this linear program as a maximum-flow problem, which can also be solved in polynomial time. The mapping $\varepsilon \mapsto \rho(\varepsilon)$ is a non-increasing step function whose jumps can only occur at points in $D := \{d(x_i, x_j) \mid i, j \in [n]\}$. Let $D' := D \cup \{0\}$. Then, we have

$$\mathsf{P}(\mu, \nu) = \inf\{\varepsilon \geq 0 \mid \rho(\varepsilon) \leq \varepsilon\}$$
$$= \min\left(\{\varepsilon \in D' \mid \rho(\varepsilon) \leq \varepsilon\} \cup \{\rho(\varepsilon) \mid \varepsilon \in D' \text{ with } \rho(\varepsilon) > \varepsilon\}\right),$$

which can be computed in time polynomial in $n$ and the number of bits needed to encode $d$, $\mu$, and $\nu$ since $D$ contains at most $n^2$ values. □

Theorem 16 can then be used to compute the distance $\delta_{\mathsf{P},h}$ of two given graphs. Intuitively, we run the 1-WL on these graphs in parallel and additionally, after the $i$th refinement round, compute a matrix $D_i$ containing the distances $d_{\mathsf{P},i}$ between any pair of colors. This matrix can be computed from the new colors and $D_{i-1}$. In the end, we use $D_h$ to compute $\delta_{\mathsf{P},h}$. This is essentially the same approach that Chen et al. [26] use for showing that their Weisfeiler–Leman distance is polynomial-time computable.

**Theorem 18.** Let $0 \leq h < \infty$, and let $G$ and $H$ be graphs. Then, $\delta_{\mathsf{P},h}(G, H)$ can be computed in time polynomial in $h$ and the size of $G$ and $H$.

*Proof.* We inductively compute $D_h \in \mathbb{R}^{V(G) \times V(H)}$ with $D_{h,uv} = d_{\mathsf{P},h}(\mathsf{i}_{G,h}(u), \mathsf{i}_{H,h}(v))$. In a practical setting, it makes sense to index $D_h$ by the colors/IDMs of the vertices instead in a practical setting to reduce the size of the matrix and avoid unnecessary computations of the same value. However, for the sake of simplicity, we stick to this definition of $D_h$ here. For $h = 0$, this is just the all-zero matrix. After having computed $D_n$, for all pairs of vertices $u \in V(G), v \in V(H)$, we use the matrix $D_h$ to compute the measures $\mathsf{i}_{G,h+1}(u)$ and $\mathsf{i}_{H,h+1}(v)$ and then use Theorem 16 to compute their distance w.r.t. $d_{\mathsf{P},h+1}$. We note that we do not need the distances between the IDMs of two vertices $u, v \in V(G)$ or $u, v \in V(H)$ for this. Hence, the matrix $D_h \in \mathbb{R}^{V(G) \times V(H)}$ suffices to compute $D_{h+1} \in \mathbb{R}^{V(G) \times V(H)}$. Furthermore, we note that we actually do not need to know the precise element of $\mathbb{M}_h$ assigned to a vertex to do so: we basically run color refinement in parallel and use its renamed colors after $h$ rounds as our space $S$. From the coloring after $h$ refinement rounds and the matrix $D_h$, another application of Theorem 16 yields the distance $\delta_{\mathsf{P},h}(G, H) = \delta_{\mathsf{P},h}(\nu_{G,h}, \nu_{H,h})$. □

Let us give a rough analysis of the running time of the algorithm from Theorem 18. Let $n$ be the maximum number of vertices in the two graphs $G$ and $H$. Then, we have to compute the Prokhorov metric of two IDMs $h \cdot n^2$ times and once for the final distance $\delta_{\mathsf{P},h}(G, H)$. To compute a single such Prokhorov metric, we have to solve $n^2$ maximum-flow problems on a graph with $n$ vertices and $n^2$

edges, where a single maximum-flow problem takes time $\mathcal{O}(n^3)$ [96]. Hence, the overall running time is $\mathcal{O}(h \cdot n^2 \cdot n^2 \cdot n^3) = \mathcal{O}(h \cdot n^7)$; We will see that the distance $\delta_{\mathsf{W},h}$ can be computed in time $\mathcal{O}(h \cdot n^5 \cdot \log n)$, which is much better, but nevertheless not ideal for practical purposes. This shows why we need a different way of approximating these distances for practical purposes and leads to our experiments in Section 6.

For two graphs $G$ and $H$ on $n$ and $m$ vertices, respectively, the 1-$\mathsf{WL}$ test converges after at most $\max\{n, m\}$ refinement rounds, i.e., the color partitions are stable and cannot further be refined in subsequent rounds. For $d_{\mathsf{P},h}$, however, it is conceivable that $d_{\mathsf{P},h}$ still changes even after having obtained stable color partitions. More precisely, by Lemma 15, $d_{\mathsf{P},h}$ may still increase, i.e., we have a second convergence step where the distances $d_{\mathsf{P},h}$ converge after the color partitions have already stabilized. However, for the Prokhorov distance, we can argue that this takes, at most, polynomially many steps.

**Theorem 19.** Let $G$ and $H$ be graphs. Then, $\delta_{\mathsf{P},\infty}(G, H)$ can be computed in polynomial time.

*Proof.* Let $n$ and $m$ be the number of vertices of $G$ and $H$, respectively. Then, the masses that an IDM assigns to a color class are all of the form $i/(n \cdot m)$ for a natural number $i$, which implies that the resulting Prokhorov distance between such IDMs also is of this form. This inductively yields that all entries of the distance matrices in the computation of $\delta_{\mathsf{P},h}(G, H)$ for $h \in \mathbb{N}$ are of this form and, with the monotonicity of Lemma 15, we get that all $n \cdot m$ entries can increase at most $n \cdot m$ times, i.e., the distance matrix stays constant after at most $(n \cdot m)^2$ steps. Thus, the claim follows. $\square$

When combined with the previous analysis, this yields that $\delta_{\mathsf{P},\infty}(G, H)$ can be computed in time $\mathcal{O}(n^4 \cdot n^7) = \mathcal{O}(n^{11})$, where $n$ is the maximum number of vertices in the two input graphs.

### C.2.3 The unbalanced Wasserstein metric

Let us now turn our attention to our unbalanced Wasserstein metric. To this end, let $(S, d)$ be a complete separable metric space with Borel $\sigma$-algebra $\mathcal{B}$. We first note that the *Wasserstein distance* of two probability measures $\mu, \nu \in \mathscr{P}(S)$ is

$$\mathsf{W}(\mu, \nu) := \inf_{\gamma \in \mathcal{C}(\mu, \nu)} \int_{S \times S} d(x, y) \, d\gamma(x, y),$$

where $\mathcal{C}(\mu, \nu)$ is the set of all couplings of $\mu$ and $\nu$, i.e., probability measures $\gamma \in \mathscr{P}(S \times S)$ with marginals $\mu$ and $\nu$ or, formally, $(p_1)_* \gamma = \mu$ and $(p_2)_* \gamma = \nu$. In Section 3, for finite measures $\mu, \nu \in \mathscr{M}_{\leq 1}(S)$, where where we assume $\|\mu\| \geq \|\nu\|$ without loss of generality, we defined the unbalanced Wasserstein distance

$$\mathsf{W}(\mu, \nu) := \|\mu\| - \|\nu\| + \inf_{\gamma \in \mathcal{M}(\mu, \nu)} \int_{S \times S} d(x, y) \, d\gamma(x, y),$$

where $\mathcal{M}(\mu, \nu)$ is the set of all measures $\gamma \in \mathscr{M}_{\leq 1}(S \times S)$ such that $(p_1)_* \gamma \leq \mu$ and $(p_2)_* \gamma = \nu$. We first note that $W$ is well-defined as $\mathcal{M}(\mu, \nu)$ is non-empty since $\frac{1}{\|\mu\|}(\mu \times \nu) \in \mathcal{M}(\mu, \nu)$. On probability measures, this definition coincides with the Wasserstein distance [35, Section 11.8], and on finite histograms, it coincides with the *Earth Mover's Distance* $\widehat{EMD}_1$ introduced by Pele and Werman [97, 98]. To prove that $W$ indeed is a metric on $\mathscr{M}_{\leq 1}(S)$, we follow the proof of Pele and Werman [97] and relate our definition of $W$ to the classical definition on probability measures as follows: For a complete separable metric space $(S, d)$ where any two elements have a distance at most one and a measure $\mu \in \mathscr{M}_{\leq 1}(S, d)$ we define the space $(S^*, d^*)$ and the probability measure $\mu \in \mathscr{P}(S^*, d^*)$ by adding a new element $*$ of mass $1 - \|\mu\|$ that has distance one to all other elements. Formally, we let $S^* := S \cup \{*\}$,

$$d^*(x, y) := \begin{cases} d(x, y) & \text{if } x, y \in S, \\ 1 & \text{otherwise} \end{cases}$$

for all $x, y \in S$, and

$$\mu^*(A) := \begin{cases} \mu(A \setminus \{*\}) + (1 - \|\mu\|) & \text{if } * \in A, \\ \mu(A) & \text{otherwise} \end{cases}$$

for every $A \in \mathcal{B}(S^*, d^*)$. For measures obtained this way, we have the following.

**Lemma 20.** Let $(S, d)$ be a complete separable metric space where any two elements have a distance at most one and $\mu, \nu \in \mathscr{M}_{\leq 1}(S, d)$. Then, $\mathsf{W}(\mu^*, \nu^*) = \mathsf{W}(\mu, \nu)$.

*Proof.* Without loss of generality, we assume that $\|\mu\| \geq \|\nu\|$. To show that $\mathsf{W}(\mu^*, \nu^*) \leq \mathsf{W}(\mu, \nu)$, we show that a $\gamma \in \mathcal{M}(\mu, \nu)$ yields a $\gamma^* \in \mathcal{C}(\mu^*, \nu^*)$ such that

$$\int_{S^* \times S^*} d^*(x, y)\, d\gamma^*(x, y) = \|\mu\| - \|\nu\| + \int_{S \times S} d(x, y)\, d\gamma(x, y). \tag{3}$$

To this end, we define $\gamma^* \in \mathcal{C}(\mu^*, \nu^*)$ by

$$\gamma^*(C) := \gamma(C \cap (S \times S)) + (\mu - (p_1)_* \gamma)(\{x \in S \mid (x, *) \in C\})$$
$$+ (1 - \|\mu\|)\mathbb{1}[(*, *) \in C]$$

for every $C \in \mathcal{B}(S \times S)$. It is easy to verify that $\gamma^*$ actually is a probability measure with marginals $\mu^*$ and $\nu^*$. Then,

$$\int_{S^* \times S^*} d^*(x, y)\, d\gamma^*(x, y) = \int_{S \times S} d^*(x, y)\, d\gamma^*(x, y) + \int_{\{*\} \times S} d^*(x, y)\, d\gamma^*(x, y)$$
$$+ \int_{S \times \{*\}} d^*(x, y)\, d\gamma^*(x, y)$$
$$+ \int_{\{*\} \times \{*\}} d^*(x, y)\, d\gamma^*(x, y)$$
$$= \int_{S \times S} d(x, y)\, d\gamma(x, y) + 0 + \|\mu - (p_1)_* \gamma\| + 0$$
$$= \|\mu\| - \|\nu\| + \int_{S \times S} d(x, y)\, d\gamma(x, y).$$

For the other direction $\mathsf{W}(\mu^*, \nu^*) \geq \mathsf{W}(\mu, \nu)$, we show that a $\gamma^* \in \mathcal{C}(\mu^*, \nu^*)$ restricts to a $\gamma \in \mathcal{M}(\mu, \nu)$ such that (3) holds. Let $\gamma$ denote the restriction of $\gamma^*$ to $\mathcal{B}(S \times S)$. We prove that $\gamma \in \mathcal{M}(\mu, \nu)$. First, we have

$$(p_1)_* \gamma(A) = \gamma(A \times S) \leq \gamma^*(A \times S^*) = (p_1)_* \gamma^*(A) = \mu^*(A) = \mu(A),$$

and, analogously, $(p_2)_* \gamma(A) \leq \nu(A)$ for every $A \in \mathcal{B}(S)$. Now, assume that $(p_2)_* \gamma(A) < \nu(A)$ for some $A \in \mathcal{B}(S)$. Then,

$$\|\nu\| = \gamma(S \times A) + \gamma(S \times \bar{A}) < \nu(A) + \nu(\bar{A}) = \|\nu\|,$$

which is a contradiction. Next, observe that

$$\gamma^*(\{*\} \times S) + \gamma^*(\{*\} \times \{*\}) = 1 - \gamma^*(S \times S^*) = 1 - \mu^*(S) = 1 - \|\mu\|,$$

and also $\gamma^*(S \times \{*\}) + \gamma^*(\{*\} \times \{*\}) = 1 - \|\nu\|$. By the previous paragraph, we have $\gamma^*(S \times S) = \|\nu\|$, which yields that

$$\gamma^*(\{*\} \times S) + \gamma^*(S \times \{*\}) + \gamma^*(\{*\} \times \{*\}) = 1 - \|\nu\|.$$

We get $\gamma^*(\{*\} \times S) = 0$ and, hence, $\gamma^*(\{*\} \times \{*\}) = 1 - \|\mu\|$ and, thus, $\gamma^*(S \times \{*\}) = \|\mu\| - \|\nu\|$. From this we obtain

$$\int_{S^* \times S^*} d^*(x, y)\, d\gamma^*(x, y) = \int_{S \times S} d(x, y)\, d\gamma(x, y) + \gamma^*(\{*\} \times S) + \gamma^*(S \times \{*\})$$
$$= \|\mu\| - \|\nu\| + \int_{S \times S} d(x, y)\, d\gamma(x, y).$$

$\square$

**Corollary 21.** Let $(S, d)$ be a separable metric space. Then, $\mathsf{W}$ is a metric on $\mathscr{M}_{\leq 1}(S, d)$.

*Proof.* Follows immediately from Lemma 20 and the fact that $\mathsf{W}$ is a metric on $\mathscr{P}(S^*, d^*)$, cf. [35, Lemma 11.8.3]. $\square$

To show that W also metrizes the weak topology, we show that it is upper-bounded by the Prokhorov metric and lower-bounded by another metric that we define in the following. For two $\mu, \nu \in \mathscr{M}_{\leq 1}(S)$, let

$$\mathsf{K}(\mu, \nu) := \sup_{\substack{\|f\|_L \leq 1, \\ \|f\|_\infty \leq 1}} \left| \int f d\mu - \int f d\nu \right| \quad \text{and} \quad \mathsf{BL}(\mu, \nu) := \sup_{\|f\|_{BL} \leq 1} \left| \int f d\mu - \int f d\nu \right|$$

be the *Kantorovich-Rubinshtein distance* and the *Bounded-Lipschitz distance* of $\mu$ and $\nu$, respectively, where $\|f\|_{BL} := \|f\|_L + \|f\|_\infty$ for $f \colon S \to \mathbb{R}$. We clearly have $\mathsf{BL}(\mu, \nu) \leq \mathsf{K}(\mu, \nu) \leq 2\mathsf{BL}(\mu, \nu)$. If $(S, d)$ is separable, then $\mathsf{K}$ and $\beta$ metrize the weak topology [17, Theorem 8.3.2].

We lower-bound the unbalanced Wasserstein distance W by the Kantorovich-Rubinshtein distance K and upper-bound it by the Prokhorov distance P (with a factor of two). This directly implies that W metrizes the weak topology. These inequalities for probability measures are easily found in the literature, see Dudley [35], and we generalize them to finite measures of total mass at most one. There are two ways to go on about this. First, one could use Lemma 20 and show that the analogous statement holds for the Prokhorov distance P; then, these inequalities follow from the ones for probability measures. Secondly, one can prove these inequalities directly, which is what we do here as the proofs provide a better understanding of these metrics, although they are not as straightforward as one might expect. In particular, the proof of upper-bounding the Wasserstein distance by the Prokhorov distance in [35] is quite complicated, and we follow the simpler proofs outlined in Schay [106] and García-Palomares and Evarist Giné [45], which use the duality of linear programming. We remark that the inequalities between the Prokhorov distance P and the unbalanced Wasserstein distance W presented in Section 3 are a special case of the following lemma.

**Lemma 22.** Let $(S, d)$ be a complete separable metric space. Then, for all $\mu, \nu \in \mathscr{M}_{\leq 1}(S)$,

$$\mathsf{BL}(\mu, \nu) \leq \mathsf{K}(\mu, \nu) \leq \mathsf{W}(\mu, \nu) \leq 2\mathsf{P}(\mu, \nu) \leq 4\sqrt{\mathsf{BL}(\mu, \nu)}.$$

*Proof.* In the following, we assume $\|\mu\| \geq \|\nu\|$ without loss of generality. The first inequality follows immediately from the definition. For the second inequality, we first prove the following claim.

**Claim 23.** Let $f \colon S \to \mathbb{R}^n$ be Lipschitz. Then,

$$\left\| \int_S f d\mu - \int_S f d\nu \right\|_2 \leq \|f\|_\infty \cdot (\|\mu\| - \|\nu\|) + \|f\|_L \cdot \int_{S \times S} d(x, y) \, d\gamma(x, y)$$

for every $\gamma \in \mathcal{M}(\mu, \nu)$.

*Proof.* Let $\gamma \in \mathcal{M}(\mu, \nu)$. Then,

$$\left\| \int_S f d\mu - \int_S f d\nu \right\|_2 = \left\| \int_S f d\mu - \int_S f \, d(p_1)_* \gamma + \int_S f \, d(p_1)_* \gamma - \int_S f d\nu \right\|_2$$

$$\leq \left\| \int_S f d(\mu - (p_1)_* \gamma) \right\|_2 + \left\| \int_S f \, d(p_1)_* \gamma - \int_S f d(p_2)_* \gamma \right\|_2$$

$$= \left\| \int_S f d(\mu - (p_1)_* \gamma) \right\|_2 + \left\| \int_{S \times S} (f(x) - f(y)) d\gamma(x, y) \right\|_2$$

$$\leq \int_S \|f\|_2 d(\mu - (p_1)_* \gamma) + \int_{S \times S} \|f(x) - f(y)\|_2 d\gamma(x, y)$$

$$\leq \int_S \|f\|_\infty d(\mu - (p_1)_* \gamma) + \int_{S \times S} \|f\|_L \cdot d(x, y) \, d\gamma(x, y)$$

$$= \|f\|_\infty \cdot \|\mu - (p_1)_* \gamma\| + \|f\|_L \cdot \int_{S \times S} d(x, y) \, d\gamma(x, y)$$

$$= \|f\|_\infty \cdot (\|\mu\| - \|\nu\|) + \|f\|_L \cdot \int_{S \times S} d(x, y) \, d\gamma(x, y).$$

■

The claim implies that, for an $f: S \to \mathbb{R}$ with $\|f\|_L \leq 1$ and $\|f\|_\infty \leq 1$, we have $\left| \int_S f d\mu - \int_S f d\nu \right| \leq \mathsf{W}(\mu, \nu)$, and since this holds for every such $f$, we get $\mathsf{K}(\mu, \nu) \leq \mathsf{W}(\mu, \nu)$.

For the third inequality, we follow Schay [106] and García-Palomares and Evarist Giné [45], who proved this inequality for probability measures by using the duality of linear programming in the finite case and then lifting this result to the infinite case. Note that we already used this connection to linear programming in as Claim 17 to show that the Prokhorov metric is efficiently computable in Theorem 16.

Assume that $\|\mu\| \geq \|\nu\|$ without loss of generality. If $S$ is finite (or $\mu$ and $\nu$ are finitely supported), we use that $\mathsf{P}(\mu, \nu) = \inf\{\varepsilon > 0 \mid \rho(\varepsilon) \leq \varepsilon\}$ for the function $\rho$ defined in the proof of Theorem 16. Claim 17 expresses the value $\rho(\varepsilon)$ for an $\varepsilon > 0$ as a linear program and we can interpret the solution to this linear program as a measure $\gamma_\varepsilon \in \mathcal{M}(\mu, \nu)$ that satisfies $\rho(\varepsilon) = \|\mu\| - \gamma_\varepsilon(\Delta_\varepsilon)$, where $\Delta_\varepsilon := \{(x, y) \in S \times S \mid d(x, y) \leq \varepsilon\}$. Choose a sequence $\varepsilon_n$ with $\varepsilon_n \downarrow \mathsf{P}(\mu, \nu)$ and $\rho(\varepsilon_n) \leq \varepsilon_n$. Then,

$$\varepsilon_n \geq \rho(\varepsilon_n) = \|\mu\| - \gamma_{\varepsilon_n}(\Delta_{\varepsilon_n}) = \|\mu\| - \|\gamma_{\varepsilon_n}\| + \gamma_{\varepsilon_n}\left(\overline{\Delta_{\varepsilon_n}}\right) = \|\mu\| - \|\nu\| + \gamma_{\varepsilon_n}\left(\overline{\Delta_{\varepsilon_n}}\right)$$

and, hence,

$$\mathsf{W}(\mu, \nu) \leq \|\mu\| - \|\nu\| + \int_{S \times S} d(x, y) \, d\gamma_{\varepsilon_n}(x, y)$$

$$= \|\mu\| - \|\nu\| + \int_{\Delta_{\varepsilon_n}} d(x, y) \, d\gamma_{\varepsilon_n}(x, y) + \int_{\overline{\Delta_{\varepsilon_n}}} d(x, y) \, d\gamma_{\varepsilon_n}(x, y)$$

$$\leq \|\mu\| - \|\nu\| + \varepsilon_n \cdot \|\gamma_{\varepsilon_n}\| + \|d\|_\infty \cdot \gamma_{\varepsilon_n}\left(\overline{\Delta_{\varepsilon_n}}\right) \leq 2 \cdot \varepsilon_n,$$

which yields that $\mathsf{W}(\mu, \nu) \leq 2\mathsf{P}(\mu, \nu)$.

We can also consider the limit of the $\gamma_{\varepsilon_n}$. More precisely, since $\mathscr{P}(S \times S)$ is compact, the sequence $\gamma_{\varepsilon_n}$, by possibly considering a subsequence, converges to a measure $\gamma$ of the same total mass, which is $\|\nu\|$. For this, note that results for probability measures apply since we can just rescale a sequence since all measures have the same total mass. We then have $\gamma \in \mathcal{M}(\mu, \nu)$, cf. the case for $S$ infinite, and for $\varepsilon > \mathsf{P}(\mu, \nu)$,

$$\|\mu\| - \|\nu\| + \gamma\left(\overline{\Delta_\varepsilon}\right) \leq \|\mu\| - \|\nu\| + \liminf_{n \to \infty} \gamma_{\varepsilon_n}\left(\overline{\Delta_\varepsilon}\right)$$

$$\leq \|\mu\| - \|\nu\| + \liminf_{n \to \infty} \gamma_{\varepsilon_n}\left(\overline{\Delta_{\varepsilon_n}}\right) \leq \liminf_{n \to \infty} \varepsilon_n = \mathsf{P}(\mu, \nu),$$

where the first inequality holds by the Portmanteau theorem, which again implies $\mathsf{W}(\mu, \nu) \leq 2\mathsf{P}(\mu, \nu)$ as before.

The infinite case works by approximating $\mu$ and $\nu$ by finite measures and is mostly analogous to [45] as we can rescale a sequence of measures of the same total mass to probability measures. More precisely, one weakly approximates $\mu$ and $\nu$ by sequences of measures $\mu_n$ and $\nu_n$ of total mass $\|\mu\|$ and $\|\nu\|$, respectively, supported by the finite set $S_n$ for $n \geq 1$. Then, $\lim_{n \to \infty} \mathsf{P}(\mu_n, \nu_n) = \mathsf{P}(\mu, \nu)$ by the triangle inequality.

For $n \geq 0$, take $\gamma_n \in \mathcal{M}(\mu_n, \nu_n)$ of total mass $\|\nu\|$ for $\mu_n$ and $\nu_n$ as defined in the finite case, i.e., we have $\|\mu\| - \|\nu\| + \gamma\left(\overline{\Delta_\varepsilon}\right) \leq \mathsf{P}(\mu_n, \nu_n)$ for every $\varepsilon > \mathsf{P}(\mu_n, \nu_n)$. Since $\mu_n$ and $\nu_n$ are uniformly tight as weakly convergent sequences, the sequence $\gamma_n$ is also uniformly tight and, by possibly considering a subsequence, converges to a measure $\gamma$ of total mass $\|\nu\|$. By continuity of $p_2$, we then have that $(p_2)_* \gamma_n$ converges to $(p_2)_* \gamma$. Since $(p_2)_* \gamma_n = \nu_n$ converges to $\nu$, we get that $(p_2)_* \gamma = \nu$. Moreover, one can show that $(p_2)_* \gamma_n \leq \mu_n$ implies that $(p_2)_* \gamma \leq \mu$: For a closed set $C \in \mathcal{B}(S)$ and an $\eta > 0$, approximate $C$ by the Lipschitz function $f(x) := \max\{0, 1 - \frac{d(x, C)}{\eta}\}$. Then, $\mathbf{1}_C \leq f \leq \mathbf{1}_{C^\eta}$ and, hence,

$$(p_2)_* \gamma(C) = \int_S \mathbf{1}_C d(p_2)_* \gamma \leq \int_S f d(p_2)_* \gamma = \lim_{n \to \infty} \int_S f d(p_2)_* \gamma_n$$

$$\leq \lim_{n \to \infty} \int_S f d\mu_n$$

$$= \int_S f d\mu$$

$$\leq \int_S \mathbf{1}_{C^\eta} d\mu = \mu(C^\eta).$$

Since this holds for every $\eta > 0$ and $C$ is closed, we get $(p_2)_*\gamma(C) \le \mu(C)$. Then, since $(p_2)_*\gamma$ is regular by Ulam's theorem, we get

$$(p_2)_*\gamma(A) = \sup\{(p_2)_*\gamma(K) \mid K \text{ compact}, K \subseteq A, K \in \mathcal{B}(S)\} \le \mu(A)$$

for every $A \in \mathcal{B}(S)$. All in all, we have $\gamma \in \mathcal{M}(\mu, \nu)$.

For every $\varepsilon > \mathsf{P}(\mu, \nu)$, we have $\varepsilon > \mathsf{P}(\mu_n, \nu_n)$ if $n$ is large enough since $\mathsf{P}(\mu_n, \nu_n)$ converges to $\mathsf{P}(\mu, \nu)$. Then,

$$\|\mu\| - \|\nu\| + \gamma\left(\overline{\Delta_\varepsilon}\right) \le \|\mu\| - \|\nu\| + \liminf_{n\to\infty} \gamma_n\left(\overline{\Delta_\varepsilon}\right) \le \liminf_{n\to\infty} \mathsf{P}(\mu_n, \nu_n) = \mathsf{P}(\mu, \nu),$$

where the first inequality follows from the Portmanteau theorem. Then, we get

$$\mathsf{W}(\mu, \nu) \le \|\mu\| - \|\nu\| + \varepsilon \cdot \|\gamma\| + \|d\|_\infty \cdot \gamma\left(\overline{\Delta_\varepsilon}\right) \le \varepsilon + \mathsf{P}(\mu, \nu),$$

similarly to the finite case. Since this holds for every $\varepsilon > \mathsf{P}(\mu, \nu)$, the claim follows.

For the last inequality, we show that $\mathsf{P}(\mu, \nu) \le 2\sqrt{\mathsf{BL}(\mu, \nu)}$ by following the proof of [35, Theorem 11.3.3] for probability measures. We have

$$\mathsf{P}(\mu, \nu) = \inf\{\varepsilon > 0 \mid \mu(A) \le \nu(A^\varepsilon) + \varepsilon \text{ for every } A \in \mathcal{B}(S)\}$$

and fix an $A \in \mathcal{B}(S)$ and an $\varepsilon > 0$. Let $f(x) := \max\{0, 1 - \frac{d(x,A)}{\varepsilon}\}$. Then, $\|f\|_{BL} \le 1 + \frac{1}{\varepsilon}$ and, hence,

$$\begin{aligned}
\mu(A) \le \int_S f \, d\mu &= \int_S f \, d\nu + \left(\int_S f \, d\mu - \int_S f \, d\nu\right) \\
&\le \int_S f \, d\nu + \|f\|_{BL} \cdot \mathsf{BL}(\mu, \nu) \\
&\le \int_S f \, d\nu + \left(1 + \frac{1}{\varepsilon}\right) \cdot \mathsf{BL}(\mu, \nu) \le \nu(A^\varepsilon) + \left(1 + \frac{1}{\varepsilon}\right) \cdot \mathsf{BL}(\mu, \nu),
\end{aligned}$$

which implies that $\mathsf{P}(\mu, \nu) \le \max\{\varepsilon, \left(1 + \frac{1}{\varepsilon}\right) \cdot \mathsf{BL}(\mu, \nu)\}$. For $\varepsilon := \sqrt{\mathsf{BL}(\mu, \nu)}$, this yields that $\mathsf{P}(\mu, \nu) \le \mathsf{BL}(\mu, \nu) + \sqrt{\mathsf{BL}(\mu, \nu)}$. Hence, if $\mathsf{BL}(\mu, \nu) \le 1$, then $\mathsf{P}(\mu, \nu) \le 2 \cdot \sqrt{\mathsf{BL}(\mu, \nu)}$. If $\mathsf{BL}(\mu, \nu) > 1$, then $\mathsf{P}(\mu, \nu) \le 2 \cdot \sqrt{\mathsf{BL}(\mu, \nu)}$ trivially holds since $\mathsf{P}(\mu, \nu) \le 1$ as $\mu$ and $\nu$ have total mass at most one. $\qquad\square$

**Lemma 24.** Let $0 \le h \le \infty$. The metric $d_{\mathsf{W},h}$ is well-defined and metrizes the topology of $\mathbb{M}_h$. The metric $\delta_{\mathsf{W},h}$ is well-defined and metrizes the topology of $\mathscr{P}(\mathbb{M}_h)$.

*Proof.* By Theorem 13 and Lemma 22, for a complete separable metric space $(S, d)$, we have that $(\mathscr{M}(S), W)$ is a complete separable metric space and convergence in $W$ is equivalent to weak convergence of measures. Then, the proof is analogous to Lemma 14. $\qquad\square$

### C.2.4 Computability of the unbalanced Wasserstein metric

It only remains to argue that $d_{\mathsf{W},h}$ and $\delta_{\mathsf{W},h}$ are polynomial-time computable. This can be done in the same way as in Theorem 16, i.e., one again constructs a matrix of distances that one updates while running the 1-$\mathsf{WL}$ in parallel on the two input graphs. Here, the metric $W$ can be computed in polynomial-time by using the same flow network [98] as Chen et al. [26] did in their proof for the polynomial-time computability of their Weisfeiler–Leman distance. Following their arguments, one then obtains a running time of $\mathcal{O}(h \cdot n^5 \cdot \log n)$ for $\delta_{\mathsf{W},h}$, where $h \in \mathbb{N}$ and $n$ is the maximum number of vertices in the two input graphs.

For $d_{\mathsf{W},\infty}$ and $\delta_{\mathsf{W},\infty}$, in contrast to Theorem 16, we cannot argue that all distances are of the form $i/(n \cdot m)$, where $n$ and $m$ are the numbers of vertices of the input graphs: with every iteration, we possibly get an additional factor of $1/(n \cdot m)$. However, we can start rounding the distances after $\log(1/\varepsilon)$ iterations to integer multiples of $1/(n \cdot m)^{\log_2(1/\varepsilon)}$, which results in an additive error of $\varepsilon$.

### C.2.5 Continuity of message-passing graph neural networks

It remains to prove Lemma 3. Recall that, for an $L$-layer MPNN model $\boldsymbol{\varphi} = (\varphi_i)_{i=0}^L$, we inductively defined the *Lipschitz constant* $C_{\boldsymbol{\varphi}} \geq 0$ of $\boldsymbol{\varphi}$ by $C_{\boldsymbol{\varphi}_0} := 0$ for $t = 0$ and

$$C_{\boldsymbol{\varphi}_t} := \|\varphi_t\|_L \cdot (\|\mathbf{h}_-^{(t-1)}\|_\infty + C_{\boldsymbol{\varphi}_{t-1}})$$

for $t > 0$. If additionally $\psi$ is Lipschitz, then

$$C_{(\boldsymbol{\varphi}, \psi)} := \|\psi\|_L \cdot (\|\mathbf{h}_-^{(L)}\|_\infty + C_{\boldsymbol{\varphi}}).$$

**Lemma 3.** Let $\boldsymbol{\varphi}$ be an $L$-layer MPNN model for $L \in \mathbb{N}$ and $\psi$ be Lipschitz. Then,

$$\|\mathbf{h}_\alpha^{(L)} - \mathbf{h}_\beta^{(L)}\|_2 \leq C_{\boldsymbol{\varphi}} \cdot d_{\mathsf{W},L}(\alpha, \beta) \qquad \text{and} \qquad \|\mathbf{h}_\mu - \mathbf{h}_\nu\|_2 \leq C_{(\boldsymbol{\varphi}, \psi)} \cdot \delta_{\mathsf{W},L}(\mu, \nu)$$

for all $\alpha, \beta \in \mathbb{M}_L$ and all $\mu, \nu \in \mathscr{P}(\mathbb{M}_L)$, respectively. These inequalities also hold for $d_{\mathsf{W},\infty}$ and $\delta_{\mathsf{W},\infty}$ with an additional factor of $L$ in the Lipschitz constant.

*Proof.* We first note that, by Claim 23, for a complete separable metric space $(S, d)$ and a Lipschitz function $f \colon S \to \mathbb{R}^n$, we have

$$\left\| \int_S f d\mu - \int_S f d\nu \right\|_2 \leq \|f\|_{BL} \cdot \mathsf{W}(\mu, \nu).$$

for all $\mu, \nu \in \mathscr{M}_{\leq 1}(S)$. Let us now prove the first inequality by induction on $L$. For $L = 0$, the statement trivially holds since $\mathbb{M}_0$ is the one-point space. For the inductive step, we have

$$
\begin{aligned}
\|\mathbf{h}_\alpha^{(L)} - \mathbf{h}_\beta^{(L)}\|_2 &= \left\| \varphi_L \left( \int_{\mathbb{M}_{L-1}} \mathbf{h}_-^{(L-1)} d\alpha \right) - \varphi_L \left( \int_{\mathbb{M}_{L-1}} \mathbf{h}_-^{(L-1)} d\beta \right) \right\|_2 \\
&\leq \|\varphi_L\|_L \cdot \left\| \int_{\mathbb{M}_{L-1}} \mathbf{h}_-^{(L-1)} d\alpha - \int_{\mathbb{M}_{L-1}} \mathbf{h}_-^{(L-1)} d\beta \right\|_2 \\
&\leq \|\varphi_L\|_L \cdot \|\mathbf{h}_-^{(L-1)}\|_{BL} \cdot \mathsf{W}(\alpha, \beta) \\
&= \|\varphi_L\|_L \cdot (\|\mathbf{h}_-^{(L-1)}\|_\infty + \|\mathbf{h}_-^{(L-1)}\|_L) \cdot d_{\mathsf{W},L}(\alpha, \beta) \\
&\leq \|\varphi_L\|_L \cdot (\|\mathbf{h}_-^{(L-1)}\|_\infty + C_{\boldsymbol{\varphi}_{L-1}}) \cdot d_{\mathsf{W},L}(\alpha, \beta) \\
&= C_{\boldsymbol{\varphi}} \cdot d_{\mathsf{W},L}(\alpha, \beta)
\end{aligned}
$$

for all $\alpha, \beta \in \mathbb{M}_h$ by the induction hypothesis. Hence, we get the first inequality. The second equality then follows from the first by the same reasoning as in the inductive step above.

It remains to prove the inequalities for $d_{\mathsf{W},\infty}$ and $\delta_{\mathsf{W},\infty}$. These follow from the two just-proven inequalities. First, for all $\alpha, \beta \in \mathscr{M}_{\leq 1}(\mathbb{M})$, we have

$$
\begin{aligned}
\|\mathbf{h}_\alpha^{(L)} - \mathbf{h}_\beta^{(L)}\|_2 = \|\mathbf{h}_{p_{\infty,L}(\alpha)}^{(L)} - \mathbf{h}_{p_{\infty,L}(\beta)}^{(L)}\|_2 &\leq C_{\boldsymbol{\varphi}} \cdot d_{\mathsf{W},L}(p_{\infty,L}(\alpha), p_{\infty,L}(\beta)) \\
&\leq C_{\boldsymbol{\varphi}} \cdot L \cdot d_{\mathsf{W},\infty}(\alpha, \beta).
\end{aligned}
$$

Second, for all $\mu, \nu \in \mathscr{P}(\mathbb{M})$, we have

$$
\begin{aligned}
\|\mathbf{h}_\mu - \mathbf{h}_\nu\|_2 = \|\mathbf{h}_{(p_{\infty,L})_*\mu} - \mathbf{h}_{(p_{\infty,L})_*\nu}\|_2 &\leq C_{(\boldsymbol{\varphi}, \psi)} \cdot \delta_{\mathsf{W},L}((p_{\infty,L})_*\mu, (p_{\infty,L})_*\nu) \\
&\leq C_{(\boldsymbol{\varphi}, \psi)} \cdot L \cdot \delta_{\mathsf{W},\infty}(\mu, \nu).
\end{aligned}
$$

$\square$

### C.3 Universality of message-passing graph neural networks

Here, we prove our universal approximation theorem for MPNNs on iterated degree measures, which states that the sets $\mathcal{N}_t^1$ and $\mathcal{NN}_t^1$ are dense in $C(\mathbb{M}_t, \mathbb{R})$ and $C(\mathscr{P}(\mathbb{M}_t), \mathbb{R})$, respectively. It follows, by simple inductive applications of the Stone–Weierstrass theorem, cf. Appendix A.4, to the set $\mathcal{N}_t^1$. Essentially, we show that $\mathcal{N}_t^1$ satisfies all requirements of the Stone–Weierstrass theorem. Then, an application of the Stone–Weierstrass theorem combined with the definition of iterated degree measures

yields that $\mathcal{N}_{t+1}^1$ separates points, which allows us to show that $\mathcal{N}_{t+1}^1$ again satisfies all requirements of the Stone–Weierstrass theorem. In the following, to stress the dependence of $\mathbf{h}_-^{(t)}$ and $\mathbf{h}_-$ on the MPNN model $\varphi$ and on the MPNN model $\varphi$ and the Lipschitz function $\psi$, we sometimes write $\varphi_-^{(t)}$ and $(\varphi, \psi)_-$ instead, respectively.

**Lemma 25.** Let $0 \le t \le \infty$. The set $\mathcal{N}_t^1$ is closed under multiplication and linear combinations, contains $\mathbf{1}_{\mathbb{M}_t}$, and separates points of $\mathbb{M}_t$.

*Proof. Base of the induction.* For $t = 0$, the claim trivially holds as $\mathcal{N}_0^1$ contains precisely the constant functions and $\mathbb{M}_0 = \{1\}$ is the one-point space.

*Induction step.* Let $t > 0$. Clearly, $\mathcal{N}_t^1$ contains the all-one function $\mathbf{1}_{\mathbb{M}_t}$ since we can always choose $\varphi_t$ in an MPNN model to be the all-one function. Next, consider two functions $\varphi_-^{(t)}$ and $\varphi'_-{}^{(t)}$ from $\mathcal{N}_t^1$ for $t$-layer MPNN models $\varphi$ and $\varphi'$. Define $\psi_{\mathsf{mul}}((x,y)^T) := x \cdot y$ and $\psi_{\mathsf{add}}((x,y)^T) := x + c \cdot y$ for all $x, y \in \mathbb{R}$, where $c \in \mathbb{R}$ is fixed. Then, define the MPNN model

$$\varphi_{\mathsf{mul}} := (\varphi_0 \times \varphi'_0, \ldots, \varphi_{t-1} \times \varphi'_{t-1}, \psi_{\mathsf{mul}} \circ (\varphi_t \times \varphi'_t))$$

and define $\varphi_{\mathsf{add}}$ analogously via $\psi_{\mathsf{add}}$. Note that $\varphi_{\mathsf{mul}}$ is in fact an MPNN model since multiplication on a compact, and hence, a bounded subset of $\mathbb{R}^2$ is Lipschitz continuous. Then,

$$\varphi_-^{(t)} \cdot \varphi'_-{}^{(t)} = \varphi_{\mathsf{mul}-}^{(t)} \qquad \text{and} \qquad \varphi_-^{(t)} + c \cdot \varphi'_-{}^{(t)} = \varphi_{\mathsf{add}-}^{(t)},$$

which implies that $\mathcal{N}_t^1$ is closed under multiplication and linear combinations.

Finally, let $\alpha, \beta \in \mathbb{M}_t = \mathscr{M}_{\le 1}(\mathbb{M}_{t-1})$ with $\alpha \ne \beta$. By the induction hypothesis, the set $\mathcal{N}_{t-1}^1$ is closed under multiplication and linear combinations, contains $\mathbf{1}_{\mathbb{M}_{t-1}}$, and separates points of $\mathbb{M}_{t-1}$. Hence, it is a subalgebra of $C(\mathbb{M}_{t-1})$ that separates points and contains the constants. By the Stone–Weierstrass theorem, $\mathcal{N}_{t-1}^1$ is dense in $C(\mathbb{M}_{t-1})$, which means that there is a $(t-1)$-layer MPNN model $\varphi$ with output dimension one such that

$$\int_{\mathbb{M}_{t-1}} \varphi_-^{(t-1)} d\alpha \ne \int_{\mathbb{M}_{t-1}} \varphi_-^{(t-1)} d\beta.$$

Define the $t$-layer MPNN model $\varphi' := (\varphi_0, \ldots, \varphi_{t-1}, \psi_{\mathsf{id}})$, where $\psi_{\mathsf{id}}(x) := x$ for every $x \in \mathbb{R}$. Then, $\varphi'_-{}^{(t)} \in \mathcal{N}_t^1$, separates $\alpha$ and $\beta$ since

$$\varphi'_\alpha{}^{(t)} = \int_{\mathbb{M}_{t-1}} \varphi_-^{(t-1)} d\alpha \ne \int_{\mathbb{M}_{t-1}} \varphi_-^{(t-1)} d\beta = \varphi'_\beta{}^{(t)},$$

cf. Section 2.

The claim for $\mathcal{N}_\infty^1$ now easily follows: $\mathcal{N}_\infty^1$ contains the all-one function $\mathbf{1}_{\mathbb{M}} = \mathbf{1}_{\mathbb{M}_0} \circ p_{\infty,0} \in \mathcal{N}_\infty^1$ and separates points of $\mathbb{M}$ since, if we have $\alpha, \beta \in \mathbb{M}$ with $\alpha \ne \beta$, there is some $t \in \mathbb{N}$ such that $\alpha_t \ne \beta_t$. By the already proven claim for $\mathcal{N}_t^1$, there is a $t$-layer MPNN model $\varphi$ with output dimension one such that $\varphi_{\alpha_t}^{(t)} \ne \varphi_{\beta_t}^{(t)}$. Hence, $\varphi_-^{(t)} \circ p_{\infty,t}(\alpha) = \varphi_{\alpha_t}^{(t)} \ne \varphi_{\beta_t}^{(t)} = \varphi_-^{(t)} \circ p_{\infty,t}(\beta)$ for $\varphi_-^{(t)} \circ p_{\infty,t} \in \mathcal{N}_\infty^1$.

To see that $\mathcal{N}_\infty^1$ is closed under multiplication and linear combinations, let $\varphi_-^{(t)} \circ p_{\infty,t}$ and $\varphi'_-{}^{(t')} \circ p_{\infty,t'}$ be two functions from $\mathcal{N}_\infty^1$, where $\varphi$ and $\varphi'$ are $t$-layer and $t'$-layer MPNN models with output dimension one, respectively. We assume that $t \ge t'$; the other case is analogous. Then,

$$\varphi'_-{}^{(t')} \circ p_{\infty,t'} = \varphi'_-{}^{(t')} \circ p_{t,t'} \circ p_{\infty,t}$$

by the definition of $\mathbb{M}_\infty$. For now, assume that $\varphi'_-{}^{(t')} \circ p_{t,t'} \in \mathcal{N}_t^1$. Then, by the already proven claim for $\mathcal{N}_t^1$, we have $\varphi_-^{(t)} \cdot (\varphi'_-{}^{(t')} \circ p_{t,t'}) \in \mathcal{N}_t^1$, which means that

$$(\varphi_-^{(t)} \circ p_{\infty,t}) \cdot (\varphi'_-{}^{(t')} \circ p_{\infty,t'}) = (\varphi_-^{(t)} \cdot (\varphi'_-{}^{(t')} \circ p_{t,t'})) \circ p_{\infty,t} \in \mathcal{N}_\infty^1.$$

This implies that $\mathcal{N}_\infty^1$ is closed under multiplication, and by analogous reasoning, we get that it is closed under linear combinations. To finish the proof, it remains to show that $\varphi'_-{}^{(t')} \circ p_{t,t'} \in \mathcal{N}_t^1$, which follows from the following claim.

**Claim 26.** Let $n, t \geq 0$. Let $\varphi$ be a $t$-layer MPNN-model with $d_t = n$. Then, $\mathbf{h}_{-}^{(t)} \circ p_{t+1,t} \in \mathcal{N}_{t+1}^n$.

*Proof.* We prove the claim by induction on $t$.

*Base of the induction.* For $t = 0$, we have $\mathbf{h}_{\alpha}^{(0)} = \varphi_0$ for every $\alpha \in \mathbb{M}_0$, i.e., the only element of $\mathbb{M}_0$ is mapped to $\varphi_0 \in \mathbb{R}^n$. Let $\varphi_1 \colon \mathbb{R}^n \to \mathbb{R}^n$ be the constant function that maps all inputs to $\varphi_0$ and define the 1-layer MPNN model $\varphi' \coloneqq (\varphi_0, \varphi_1)$. Then,

$$\varphi_{-}^{(0)} \circ p_{1,0}(\alpha) = \varphi_0 = \varphi_1 \left( \int_{\mathbb{M}_0} \varphi_0 d\alpha \right) = \varphi_{\alpha}^{\,(1)}$$

for every $\alpha \in \mathbb{M}_1$, i.e., $\varphi_{-}^{(0)} \circ p_{1,0} = \varphi'_{-}^{(1)} \in \mathcal{N}_1^n$.

*Induction step.* Let $t > 0$ and $\varphi$ be a $t$-layer MPNN model. Then, for every $\alpha \in \mathbb{M}_{t+1}$, we have

$$(\varphi_{-}^{(t)} \circ p_{t+1,t})(\alpha) = \varphi_t \left( \int_{\mathbb{M}_{t-1}} \varphi_{-}^{(t-1)} dp_{t+1,t}(\alpha) \right) = \varphi_t \left( \int_{\mathbb{M}_{t-1}} \varphi_{-}^{(t-1)} d(p_{t,t-1})_*(\alpha) \right)$$

$$= \varphi_t \left( \int_{\mathbb{M}_t} \varphi_{-}^{(t-1)} \circ p_{t,t-1} \, d\alpha \right).$$

By the induction hypothesis, we have $\varphi_{-}^{(t-1)} \circ p_{t,t-1} \in \mathcal{N}_t^{d_t}$, i.e., $\varphi_{-}^{(t-1)} \circ p_{t,t-1} = \varphi'_{-}^{(t)}$ for a $t$-layer MPNN model $\varphi'$. Hence, we have

$$(\varphi_{-}^{(t)} \circ p_{t+1,t})(\alpha) = \varphi_t \left( \int_{\mathbb{M}_t} \varphi'_{-}^{(t)} \, d\alpha \right) = \varphi''_{\alpha}^{\,(t)}$$

for every $\alpha \in \mathbb{M}_{t+1}$, where $\varphi'' \coloneqq (\varphi'_0, \ldots, \varphi'_t, \varphi_t)$ is an MPNN model with $t+1$ layers. Hence, $\varphi_{-}^{(t)} \circ p_{t+1,t} \in \mathcal{N}_{t+1}^n$. ∎

□

With Lemma 25, we immediately obtain Theorem 4, which we restate here for better readability.

**Theorem 4.** Let $0 \leq L \leq \infty$. Then, $\mathcal{N}_L^1$ is dense in $C(\mathbb{M}_L, \mathbb{R})$ and $\mathcal{NN}_L^1$ is dense in $C(\mathcal{P}(\mathbb{M}_L), \mathbb{R})$.

*Proof.* By Lemma 25, the Stone–Weierstrass theorem is applicable to $\mathcal{N}_L^1$, and hence, $\mathcal{N}_L^1$ is dense in $C(\mathbb{M}_L, \mathbb{R})$. We can then use this to show that $\mathcal{NN}_L^1$ is dense in $C(\mathcal{P}(\mathbb{M}_L), \mathbb{R})$. By the same arguments as in the inductive step in the proof of Lemma 25, $\mathcal{NN}_L^1$ is closed under multiplication and linear combinations, contains the all-one function, and separates points of $\mathcal{P}(\mathbb{M}_L)$. Hence, an application of the Stone–Weierstrass theorem yields that $\mathcal{NN}_L^1$ is dense in $C(\mathcal{P}(\mathbb{M}_L), \mathbb{R})$. □

Theorem 4 then yields Corollary 5.

**Corollary 5.** Let $0 \leq L \leq \infty$ and $n > 0$. Let $\nu \in \mathcal{P}(\mathbb{M}_L)$ and $(\nu_i)_i$ be a sequence with $\nu_i \in \mathcal{P}(\mathbb{M}_L)$. Then, $\nu_i \to \nu$ if and only if $\mathbf{h}_{\nu_i} \to \mathbf{h}_\nu$ for all $L$-layer MPNN models $\varphi$ and Lipschitz $\psi \colon \mathbb{R}^{d_L} \to \mathbb{R}^n$.

*Proof.* First, let $n = 1$. When restricted to functions $\mathbf{h}_- \in \mathcal{NN}_L^n$ of the form $\mathbf{h}_\nu = \int_{\mathbb{M}_L} \mathbf{h}_-^{(L)} \, d\nu$, i.e., the function $\psi$ is the identity, the claim follows since $\mathcal{N}_L^1$ is dense in $C(\mathbb{M}_L, \mathbb{R})$ by Theorem 4 and the definition of the weak topology on $\mathcal{P}(\mathbb{M}_L)$, cf. Section 2. Since the function $\psi$ in the definition of $\mathbf{h}_- \in \mathcal{NN}_L^n$, which, in general, is of the form $\mathbf{h}_\nu = \psi \left( \int_{\mathbb{M}_L} \mathbf{h}_-^{(L)} \, d\nu \right)$, is continuous, the equivalence also holds when considering all functions in the set $\mathcal{NN}_L^n$. Finally, since one can always consider the projection to a single component and conversely map a single real number to a vector of these numbers, the equivalence also holds in the case $n > 1$. □

## C.4 Equivalence of distances

Here, we formally state and prove Theorem 1. Let us first recall the informal statement from the main body of the paper.

**Theorem 1** (informal). The following are equivalent for all graphons $U$ and $W$:

1. $U$ and $W$ are close in $\delta_P$ (or alternatively $\delta_W$).
2. $U$ and $W$ are close in $\delta_\square^{\mathcal{T}}$.
3. MPNN outputs on $U$ and $W$ are close for all MPNNs with Lipschitz constant $C$ and $L$ layers.
4. Homomorphism densities in $U$ and $W$ are close for all trees up to order $k$.

Let us give an overview of the proof. The implication $(1) \Rightarrow (3)$ is just Lemma 3, and its converse is Theorem 6. Convergence of measures in $\mathscr{P}(\mathbb{M})$ in the weak topology is equivalent to convergence in $\delta_P$ (or $\delta_W$) by Theorem 2, convergence of all MPNN outputs by Corollary 5, convergence of tree homomorphism densities by Grebík and Rocha [52], cf. Appendix D, and thus, convergence in the tree distance by Böker [18]. Therefore, the remaining equivalences all follow with a compactness argument similar to the one in Theorem 6; this was also used by Böker [18] to prove the equivalence of (2) and (4). Let us formally state the equivalence of all mentioned notions of convergence in the following theorem.

**Theorem 27.** Let $(W_i)_i$ be a sequence of graphons $W_i \colon [0,1]^2 \to [0,1]$, and let $W \colon [0,1]^2 \to [0,1]$ be a graphon. Then, the following are equivalent:

1. $\delta_P(W_i, W) \to 0$ (or equivalently $\delta_W(W_i, W) \to 0$).
2. $\delta_\square^{\mathcal{T}}(W_i, W) \to 0$.
3. $\mathbf{h}_{W_i} \to \mathbf{h}_W$ for every MPNN model $\varphi$ and Lipschitz $\psi \colon \mathbb{R}^{d_L} \to \mathbb{R}^n$, where $n > 0$.
4. $t(T, W_i) \to t(T, W)$ for every tree $T$.
5. $\nu_{W_i} \to \nu_W$.

*Proof.* (1) and (3) are equivalent to (5) by Theorem 2 and Corollary 5. (2) is equivalent to (4) by [18], which in turn is equivalent to (5) by the result of Grebík and Rocha [52], cf. Appendix D. $\square$

We note that we only state Theorem 27 for graphons and not elements of $\mathscr{P}(\mathbb{M})$ since the tree distance of [18] is only defined for graphons and not for probability measures. We further note that the following non-approximative variant of Theorem 27 also holds.

**Theorem 28.** Let $U$ and $W$ be graphons. Then, the following are equivalent:

1. $\delta_P(U, W) = 0$ (or equivalently $\delta_W(U, W) = 0$).
2. $\delta_\square^{\mathcal{T}}(U, W) = 0$.
3. $\mathbf{h}_U = \mathbf{h}_W$ for every MPNN model $\varphi$ and Lipschitz $\psi \colon \mathbb{R}^{d_L} \to \mathbb{R}^n$, where $n > 0$.
4. $t(T, U) = t(T, W)$ for every tree $T$.
5. $\nu_U = \nu_W$.

*Proof.* (2) is equivalent to (4) by [18]. The other equivalences follow as in Theorem 27 since $\mathscr{P}(\mathbb{M})$ is Hausdorff. $\square$

Let us now demonstrate how compactness can be used to deduce Theorem 1 from Theorem 27. For the direction $(1) \Rightarrow (3)$, we actually do not need compactness since Lemma 3 already is a slightly stronger statement. For the sake of completeness, we nevertheless state it as an epsilon-delta statement.

**Theorem 29.** Let $n > 0$ be fixed. For every $L \in \mathbb{N}$, $C > 0$, and $\varepsilon > 0$, there is a $\delta > 0$ such that, for all graphons $U$ and $W$, if $\delta_P(U, W) \leq \delta$, then $\|\mathbf{h}_U - \mathbf{h}_W\|_2 \leq \epsilon$ for every $L'$-layer MPNN model $\varphi$ and Lipschitz $\psi \colon \mathbb{R}^{d_{L'}} \to \mathbb{R}^n$ with $L' \leq L$ and $C_{(\varphi, \psi)} \leq C$.

*Proof.* Follows immediately from Lemma 3. $\square$

Let us formally prove the converse direction $(3) \Rightarrow (1)$.

**Theorem 6.** Let $n > 0$ be fixed. For every $\varepsilon > 0$, there are $L \in \mathbb{N}$, $C > 0$, and $\delta > 0$ such that, for all graphons $U$ and $W$, if $\|\mathbf{h}_U - \mathbf{h}_W\|_2 \leq \delta$ for every $L'$-layer MPNN model $\varphi$ and Lipschitz $\psi \colon \mathbb{R}^{d_{L'}} \to \mathbb{R}^n$ with $L' \leq L$ and $C_{(\varphi, \psi)} \leq C$, then $\delta_P(U, W) \leq \varepsilon$.

*Proof.* Assume that there is an $\varepsilon > 0$ such that such $L \in \mathbb{N}$, $C > 0$, and $\delta > 0$ do not exist. Then, for every $k \geq 0$, there are graphons $U_k$ and $W_k$ with $\|\mathbf{h}_{U_k} - \mathbf{h}_{W_k}\|_2 \leq 1/k$ for every MPNN model $\boldsymbol{\varphi}$ and Lipschitz $\psi$ with output dimension $n$, at most $k$ layers and $C_{(\boldsymbol{\varphi},\psi)} \leq k$ but also $\delta_{\mathsf{P}}(U_k, W_k) > \varepsilon$. By the compactness of the graphon space, there are subsequences $(U_{i_k})_k$ and $(W_{i_k})_k$ of $(U_k)_k$ and $(W_k)_k$ converging to graphons $\widetilde{U}$ and $\widetilde{W}$, respectively, in the cut distance and, hence, also in the tree distance. Let $\boldsymbol{\varphi}$ be an $L$-layer MPNN model and $\psi \colon \mathbb{R}^{d_L} \to \mathbb{R}^n$ be Lipschitz. Then, by Theorem 27, also $(\mathbf{h}_{U_{i_k}})_k$ and $(\mathbf{h}_{W_{i_k}})_k$ converge to $\mathbf{h}_{\widetilde{U}}$ and $\mathbf{h}_{\widetilde{W}}$, respectively. Hence,

$$\|\mathbf{h}_{\widetilde{U}} - \mathbf{h}_{\widetilde{W}}\|_2 \leq \|\mathbf{h}_{\widetilde{U}} - \mathbf{h}_{U_{i_k}}\|_2 + \|\mathbf{h}_{U_{i_k}} - \mathbf{h}_{W_{i_k}}\|_2 + \|\mathbf{h}_{W_{i_k}} - \mathbf{h}_{\widetilde{W}}\|_2 \xrightarrow{k \to \infty} 0$$

by the assumption, i.e., $\mathbf{h}_{\widetilde{U}} = \mathbf{h}_{\widetilde{W}}$. Since this holds for every MPNN model and Lipschitz $\psi$, we have $\delta_{\mathsf{P}}(\widetilde{U}, \widetilde{W}) = 0$ by Theorem 28. Then, however

$$\delta_{\mathsf{P}}(U_{i_k}, W_{i_k}) \leq \delta_{\mathsf{P}}(U_{i_k}, \widetilde{U}) + \delta_{\mathsf{P}}(\widetilde{U}, \widetilde{W}) + \delta_{\mathsf{P}}(\widetilde{W}, W_{i_k}) \xrightarrow{k \to \infty} 0$$

since $(U_{i_k})_k$ and $(W_{i_k})_k$ converge to $\widetilde{U}$ and $\widetilde{W}$, respectively, also in $\delta_{\mathsf{P}}$ by Theorem 27. This contradicts the assumption that $\delta_{\mathsf{P}}(U_k, W_k) > \varepsilon$ for every $k \geq 0$. $\square$

In the above theorem, we could simply replace $\delta_{\mathsf{P}}$ by $\delta_{\mathsf{W}}$ or $\delta_{\square}^{\mathcal{T}}$; the proof is analogous. We can also state the equivalence of these metrics explicitly, e.g., graphons that are close in $\delta_{\mathsf{P}}$ are close in $\delta_{\square}^{\mathcal{T}}$ and vice versa.

**Theorem 30.** For every $\varepsilon > 0$, there is a $\delta > 0$ such that, for all graphons $U$ and $W$, if $\delta_{\mathsf{P}}(U, W) \leq \delta$, then $\delta_{\square}^{\mathcal{T}}(U, W) \leq \varepsilon$.

*Proof.* Assume that there is a $\varepsilon > 0$ such that such a $\delta > 0$ does not exist. Then, for every $k \geq 0$, there are graphons $U_k$ and $W_k$ with $\delta_{\mathsf{P}}(U_k, W_k) \leq 1/k$ but $\delta_{\square}^{\mathcal{T}}(U_k, W_k) > \varepsilon$. By the compactness of the graphon space, there are subsequences $(U_{i_k})_k$ and $(W_{i_k})_k$ of $(U_k)_k$ and $(W_k)_k$ weakly converging to graphons $\widetilde{U}$ and $\widetilde{W}$, respectively, in the cut distance and, hence, also in the tree distance, which in turn by Theorem 27 means that they also converge in $\delta_{\mathsf{P}}$. Combining this with the assumption yields that $\delta_{\mathsf{P}}(\widetilde{U}, \widetilde{W}) = 0$, which implies $\delta_{\square}^{\mathcal{T}}(\widetilde{U}, \widetilde{W}) = 0$ by Theorem 28. This, however, is a contradiction to $\delta_{\square}^{\mathcal{T}}(U_k, W_k) > \varepsilon$ since $(U_{i_k})_k$ and $(W_{i_k})_k$ converge to $\widetilde{U}$ and $\widetilde{W}$, respectively, in $\delta_{\square}^{\mathcal{T}}$. $\square$

**Theorem 31.** For every $\varepsilon > 0$, there is an $\delta > 0$ such that, for all graphons $U$ and $W$, if $\delta_{\square}^{\mathcal{T}}(U, W) \leq \delta$, then $\delta_{\mathsf{P}}(U, W) \leq \varepsilon$.

*Proof.* Analogous to Theorem 30. $\square$

The equivalence between the tree distance and tree homomorphism densities was proven in [18]. Let us state them here for the sake of completeness. Note that the statements are essentially the same as Theorem 29 and Theorem 6, where the constant $C$ and number of layers $L$ is replaced by the order $k$ of the trees.

**Theorem 32** ([18, Corollary 20]). For every tree $T$ and every $\varepsilon > 0$, there is a $\delta > 0$ such that, for all graphons $U$ and $W$, if $\delta_{\square}^{\mathcal{T}}(U, W) \leq \delta$, then $|t(T, U) - t(T, W)| \leq \varepsilon$.

**Theorem 33** ([18, Corollary 21]). For every $\varepsilon > 0$, there are $k > 0$ and $\delta > 0$ such that, for all graphons $U$ and $W$, if $|t(T, U) - t(T, W)|$ for every tree $T$ on $k$ vertices, then $\delta_{\square}^{\mathcal{T}}(U, W) \leq \varepsilon$.

# D  Tree homomorphism densities

We briefly repeat the proof of Grebík and Rocha [52] for a universal approximation theorem for (linear combinations) of tree homomorphism density functions. The purpose of this is twofold. First, it serves as a reference since we use this result for a part of the proof of Theorem 1. Secondly, however, it illustrates the striking similarities between tree homomorphism densities and MPNNs. Not only do both tree homomorphism densities and MPNNs satisfy a universal approximation theorem, but the theorem statements and proofs in this section for tree homomorphism densities and in Appendix C.3 for MPNNs are nearly identical. Hence, tree homomorphism densities and MPNNs are two different ways of arriving at the same end goal, a set of functions that has certain properties and, in particular, describes the similarity of graphons modulo the 1-WL test.

The following definition is implicit in Grebík and Rocha [52]. For a rooted tree $T$ of height at most $h \in \mathbb{N}$, we inductively define the *homomorphism density function* $t_h(T, \cdot) \colon \mathbb{M}_h \to [0, 1]$ by letting $t_h(T, \cdot) := \mathbf{1}_{\mathbb{M}_h}$ if the height of $T$ is zero and

$$t_h(T, \alpha) := \left( \int_{\mathbb{M}_{h-1}} t_{h-1}(T_1, \cdot) \, d\alpha \right) \cdot \ldots \cdot \left( \int_{\mathbb{M}_{h-1}} t_{h-1}(T_k, \cdot) \, d\alpha \right)$$

for every $\alpha \in \mathbb{M}_h$ if $h > 0$, where $T_1, \ldots, T_k$ are the trees rooted in the children of the root of $T$. Intuitively, $t_h(T, \alpha)$ is the homomorphism density of $T$ when the root of $T$ is mapped to a point of color $\alpha$. Additionally, we define $t_\infty(T, \cdot) \colon \mathbb{M} \to [0, 1]$ by $t_\infty(T, \cdot) := t_h(T, \cdot) \circ p_{\infty, h}$. One can show that this is well defined, i.e., that the definition is independent of $h$, by a simple induction.

**Lemma 34.** Let $T$ be a rooted tree of height at most $h$. Then, $t_{h+1}(T, \cdot) = t_h(T, \cdot) \circ p_{h+1, h}$.

Let $0 \leq h \leq \infty$. For a rooted tree $T$ of height at most $h$ and $\nu \in \mathscr{P}(\mathbb{M}_h)$, define $t(T, \nu) := \int_{\mathbb{M}_h} t_h(T, \cdot) \, d\nu$. For the DIDM $\nu_{W,h}$ associated to a graphon $W$, this then simply equals the homomorphism density of $T$ in $W$ when viewing $T$ as unrooted.

**Lemma 35** ([52, Proposition 7.3]). Let $0 \leq h \leq \infty$, let $T$ be a rooted tree of height at most $h$, and let $W \in \mathcal{W}$ be a graphon. Then,

$$t(T, W) = t(T, \nu_{W,h}) = t(T, \nu_W),$$

where $T$ in the expression $t(T, W)$ is treated as unrooted.

For $0 \leq h \leq \infty$, let

$$\mathcal{T}_h := \{ t_h(T, \cdot) \mid T \text{ rooted tree of height at most } h \} \subseteq C(\mathbb{M}_h, \mathbb{R}),$$

and let $\mathcal{T}_h'$ be the set of linear combinations of functions in $\mathcal{T}_h$. The set $\mathcal{T}_h$ is not closed under linear combinations, which is why we have to consider the set $\mathcal{T}_h'$ to obtain a sub-algebra of $C(\mathbb{M}_h, \mathbb{R})$ to which the Stone–Weierstrass theorem is then applicable. In fact, $\mathcal{T}_h$ is clearly not dense in $C(\mathbb{M}_h, \mathbb{R})$ since the codomain of every function in $\mathcal{T}_h$ is $[0, 1]$. Other than that, the proof of the following lemma is identical to the proof of Lemma 25.

**Lemma 36** ([52, Proposition 7.5]). Let $0 \leq h \leq \infty$. The set $\mathcal{T}_h$ is closed under multiplication, contains $\mathbf{1}_{\mathbb{M}_h}$, and separates points of $\mathbb{M}_h$.

*Proof.* Essentially, the proof is analogous to Lemma 25. When showing that $\mathcal{T}_h$ separates points of $\mathbb{M}_h$, one cannot apply the Stone–Weierstrass theorem directly since the set of $\mathcal{T}_h$ is not closed under linear combinations. However, one can apply it to $\mathcal{T}_h'$ instead and obtain that there always is a separating linear combination. Then, if a linear combination separates two points, a function in the linear combination has to separate these points, too. □

For $0 \leq h \leq \infty$, let

$$\mathcal{TT}_h := \{ \nu \mapsto t(T, \nu) \mid T \text{ rooted tree of height at most } h \} \subseteq C(\mathscr{P}(\mathbb{M}_h), \mathbb{R}),$$

and let $\mathcal{TT}_h'$ be the set of linear combinations of functions in $\mathcal{TT}_h$. We then obtain the following universal approximation theorem for linear combinations of tree homomorphism density functions.

**Theorem 37.** Let $0 \leq h \leq \infty$. Then, $\mathcal{T}_h'$ is dense in $C(\mathbb{M}_h, \mathbb{R})$ and $\mathcal{TT}_h'$ is dense in $C(\mathscr{P}(\mathbb{M}_h), \mathbb{R})$.

*Proof.* $\mathcal{T}'_h$ has all the properties as listed in Lemma 36 and is additionally closed under linear combinations, which means that the Stone–Weierstrass Theorem is applicable, which immediately yields the first claim. The first claim together with another application of the Stone–Weierstrass Theorem then yields the second claim, where we as in the proof of Lemma 36 use that if a function in $\mathcal{T}\mathcal{T}'_h$ separates two probability measures $\mu, \nu \in \mathscr{P}(\mathbb{M}_h)$, then so does a function in $\mathcal{T}\mathcal{T}_h$. $\qquad\square$

**Lemma 38.** Let $0 \leq h \leq \infty$. Let $(\nu_i)_i$ be a sequence of measures from $\mathscr{P}(\mathbb{M}_h)$ and $\nu \in \mathscr{P}(\mathbb{M}_h)$. Then, $\nu_i \to \nu$ if and only if $t(T, \nu_i) \to t(T, \nu)$ for every rooted tree $T$ of height at most $h$.

*Proof.* Follows from Theorem 37 and the definition of the weak topology. Here, we use that the integral and the limit are linear, which means that we can replace the set $\mathcal{T}'_h$ by the set $\mathcal{T}_h$ in order to obtain the exact statement of this lemma. $\qquad\square$

Let us conclude this section with what is known as the "counting lemma" in the world of graph homomorphisms, or in other words, a tree analogue to Lemma 3. Formally, we prove that tree homomorphism density functions are Lipschitz in the metrics we defined in Section 3, where the Lipschitz constant only depends on the order of the tree. This implies that if two graphons are close in one of our metrics, then all tree homomorphism densities for trees up to a certain order are close. Hence, in comparison to Lemma 3, the order of the tree takes the place of the constant and number of layers of an MPNN model.

**Lemma 39.** Let $h \in \mathbb{N}$ and $T$ be a rooted tree of height at most $h$. Then,

$$|t_h(T, \alpha) - t_h(T, \beta)| \leq |E(T)| \cdot d_{\mathsf{W},h}(\alpha, \beta) \quad \text{and} \quad |t(T, \mu) - t(T, \nu)| \leq |V(T)| \cdot \delta_{\mathsf{W},h}(\mu, \nu)$$

for all $\alpha, \beta \in \mathbb{M}_h$ and all $\mu, \nu \in \mathscr{P}(\mathbb{M}_h)$, respectively. These inequalities also hold for $d_{\mathsf{W},\infty}$ and $\delta_{\mathsf{W},\infty}$ with an additional factor of $h$ in the Lipschitz constant.

*Proof.* We prove the first inequality by induction on the height of $T$. If it is zero, the statement trivially holds. For the inductive step, let $T_1, \ldots, T_k$ denote the trees rooted in the children of $T$. Then,

$$
\begin{aligned}
|t_h(T, \alpha) - t_h(T, \beta)| &= \left| \prod_{i \in [k]} \int_{\mathbb{M}_{h-1}} t_{h-1}(T_i, \cdot) \, d\alpha - \prod_{i \in [k]} \int_{\mathbb{M}_{h-1}} t_{h-1}(T_i, \cdot) \, d\beta \right| \\
&\leq \sum_{i \in [k]} \left| \int_{\mathbb{M}_{h-1}} t_{h-1}(T_i, \cdot) \, d\alpha - \int_{\mathbb{M}_{h-1}} t_{h-1}(T_i, \cdot) \, d\beta \right| \\
&\leq \sum_{i \in [k]} \|t_{h-1}(T_i, \cdot)\|_{BL} \cdot \mathsf{BL}(\alpha, \beta) \\
&\leq \sum_{i \in [k]} (1 + \|t_{h-1}(T_i, \cdot)\|_L) \cdot \mathsf{W}(\alpha, \beta) \\
&\leq \sum_{i \in [k]} (1 + |E(T_i)|) \cdot \mathsf{W}(\alpha, \beta) \\
&= |E(T)| \cdot \mathsf{W}(\alpha, \beta) = |E(T)| \cdot d_{\mathsf{W},h}(\alpha, \beta),
\end{aligned}
$$

which proves the claim. The second inequality then follows from the first. We have

$$
\begin{aligned}
|t(T, \mu) - t(T, \nu)| &= \left| \int_{\mathbb{M}_h} t_h(T, \cdot) \, d\mu - \int_{\mathbb{M}_h} t_h(T, \cdot) \, d\nu \right| \\
&\leq \|t_h(T, \cdot)\|_{BL} \cdot \mathsf{BL}(\mu, \nu) \\
&\leq (1 + |E(T)|) \cdot \mathsf{W}(\mu, \nu) = |V(T)| \cdot \delta_{\mathsf{W},h}(\mu, \nu).
\end{aligned}
$$

Let $h(T)$ denote the height of $T$. Then,

$$
\begin{aligned}
|t(T, \alpha) - t(T, \beta)| &= |t(T, p_{\infty,h(T)}(\alpha)) - t(T, p_{\infty,h(T)}(\beta))| \\
&\leq |E(T)| \cdot d_{\mathsf{W},h(T)}(p_{\infty,h(T)}(\alpha), p_{\infty,h(T)}(\beta)) \\
&\leq |E(T)| \cdot h(T) \cdot d_{\mathsf{W},\infty}(\alpha, \beta).
\end{aligned}
$$

for all $\alpha, \beta \in \mathbb{M}_\infty$ and

$$\begin{aligned} |t(T, \mu) - t(T, \nu)| &= |t(T, (p_{\infty, h(T)})_* \mu) - t(T, (p_{\infty, h(T)})_* \nu)| \\ &\leq |V(T)| \cdot \delta_{\mathsf{W}, h(T)}((p_{\infty, h(T)})_* \mu, (p_{\infty, h(T)})_* \nu) \\ &\leq |V(T)| \cdot h(T) \cdot \delta_{\mathsf{W}, \infty}(\mu, \nu), \end{aligned}$$

for all $\mu, \nu \in \mathscr{P}(\mathbb{M}_\infty)$, where the last inequality follows from the definition of $d_{\mathsf{W}, \infty}$. $\qquad\square$

# E   Additional experimental results

Here, we provide additional experimental results.

## E.1   Graph distance preservation

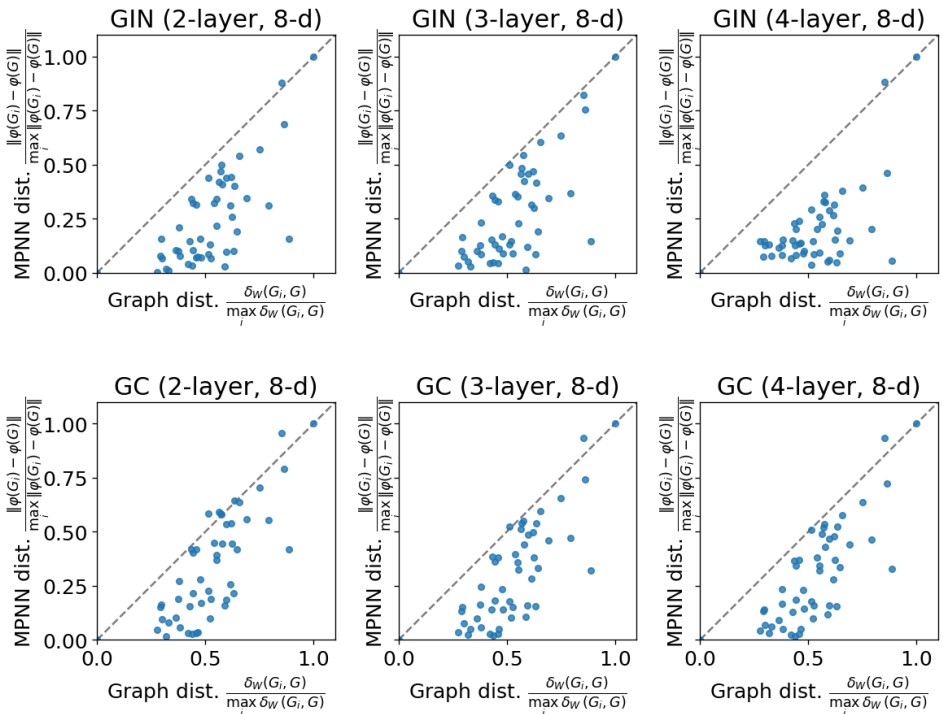

Figure 3: Increasing the number of layers first improves and then degrades the ability of MPNNs to preserve graph distance. Comparatively, untrained GIN embeddings are more sensitive than untrained GraphConv to changes in the number of layers.

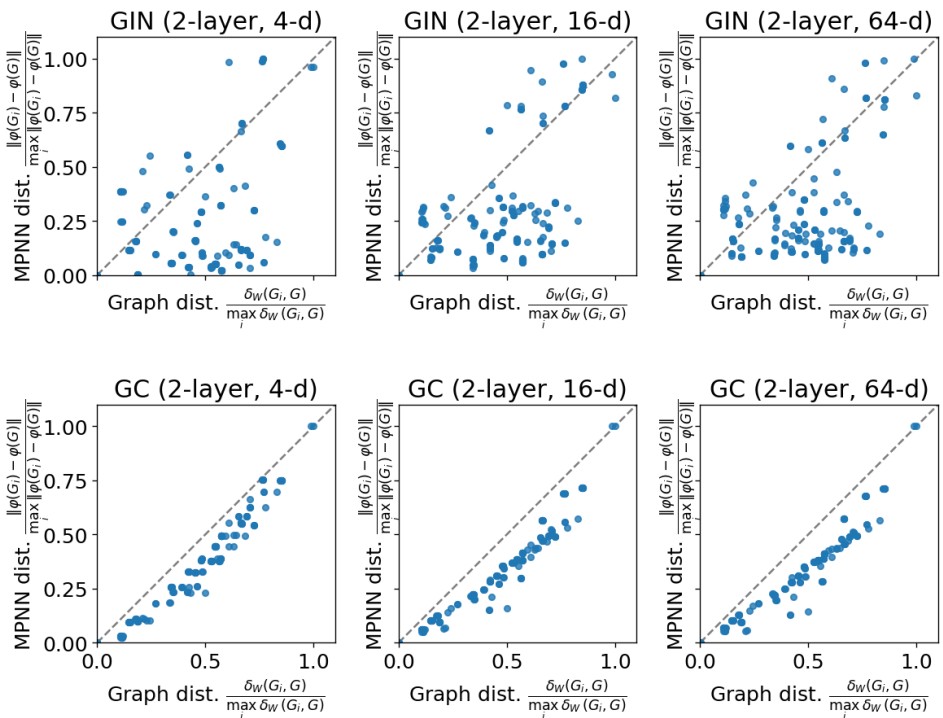

Figure 4: For MUTAG, MPNNs preserve graph distance better when increasing the number of hidden dimensions.

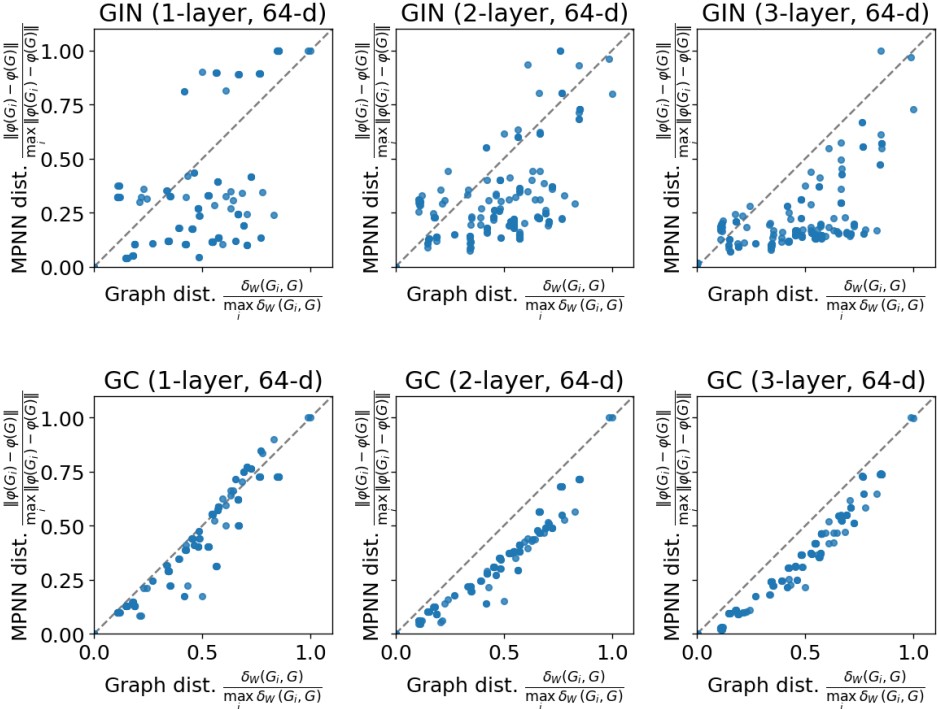

Figure 5: For MUTAG, MPNNs preserve graph distance worse when depth increases beyond a certain threshold.

## E.2 Untrained MPNNs

Table 2: Untrained MPNNs are significantly faster than trained MPNNs. We report the mean training time (in milliseconds) per epoch $\pm$ std over 200 epochs.

| Time $\downarrow$ | MUTAG | IMDB-BINARY | IMDB-MULTI | NCI1 | PROTEINS | REDDIT-BINARY |
|---|---|---|---|---|---|---|
| GIN (trained) | $17.09 \pm 0.04$ | $101.23 \pm 0.07$ | $148.09 \pm 0.03$ | $321.52 \pm 0.07$ | $90.31 \pm 0.06$ | $291.41 \pm 0.18$ |
| GIN (untrained) | $\mathbf{9.17 \pm 0.00}$ | $\mathbf{65.09 \pm 0.01}$ | $\mathbf{98.49 \pm 0.17}$ | $\mathbf{180.54 \pm 0.02}$ | $\mathbf{48.52 \pm 0.00}$ | $\mathbf{210.29 \pm 0.03}$ |
| GraphConv (trained) | $12.02 \pm 0.01$ | $80.59 \pm 0.07$ | $118.28 \pm 0.03$ | $226.31 \pm 0.07$ | $65.01 \pm 0.06$ | $228.08 \pm 0.18$ |
| GraphConv (untrained) | $\mathbf{8.08 \pm 0.00}$ | $\mathbf{64.44 \pm 0.01}$ | $\mathbf{95.41 \pm 0.17}$ | $\mathbf{163.08 \pm 0.02}$ | $\mathbf{45.95 \pm 0.00}$ | $\mathbf{185.76 \pm 0.03}$ |

Table 3: When the hidden dimensionality is smaller (3-layer, 128-hidden-dimension), untrained GIN-m is still competitive as trained GIN-m, whereas untrained GraphConv-m is less competitive as trained GraphConv-m. We report the mean accuracy $\pm$ std over 10 data splits.

| Accuracy $\uparrow$ | MUTAG | IMDB-BINARY | IMDB-MULTI | NCI1 | PROTEINS | REDDIT-BINARY |
|---|---|---|---|---|---|---|
| GIN-m (trained) | $\mathbf{78.8 \pm 2.05}$ | $69.76 \pm 1.44$ | $45.63 \pm 1.07$ | $\mathbf{78.8 \pm 0.58}$ | $73.11 \pm 0.7$ | $\mathbf{84.15 \pm 0.71}$ |
| GIN-m (untrained) | $78.34 \pm 2.0$ | $\mathbf{70.31 \pm 0.74}$ | $\mathbf{46.75 \pm 1.17}$ | $71.86 \pm 0.29$ | $\mathbf{73.17 \pm 0.63}$ | $78.8 \pm 0.5$ |
| GraphConv-m (trained) | $\mathbf{81.96 \pm 1.78}$ | $\mathbf{67.69 \pm 1.56}$ | $\mathbf{44.71 \pm 0.98}$ | $\mathbf{64.06 \pm 0.45}$ | $71.37 \pm 0.61$ | $\mathbf{81.9 \pm 0.35}$ |
| GraphConv-m (untrained) | $66.75 \pm 0.42$ | $62.33 \pm 0.87$ | $42.54 \pm 1.47$ | $62.31 \pm 0.35$ | $71.51 \pm 0.58$ | $74.09 \pm 0.34$ |

Table 4: For standard MPNNs using sum aggregation, sum pooling, and without $1/V(G)$ normalization (denoted as "MPNN-s"), the untrained ones also show competitive performance as trained ones (3-layer, 128-hidden-dimension). We report the mean accuracy $\pm$ std over 10 data splits.

| Accuracy $\uparrow$ | MUTAG | IMDB-BINARY | IMDB-MULTI | NCI1 | PROTEINS | REDDIT-BINARY |
|---|---|---|---|---|---|---|
| GIN-s (trained) | $80.99 \pm 3.57$ | $69.06 \pm 1.31$ | $44.59 \pm 0.91$ | $76.91 \pm 0.44$ | $\mathbf{72.72 \pm 0.93}$ | $80.33 \pm 1.62$ |
| GIN-s (untrained) | $\mathbf{81.78 \pm 3.29}$ | $69.44 \pm 1.37$ | $\mathbf{47.53 \pm 0.86}$ | $\mathbf{77.59 \pm 0.45}$ | $70.23 \pm 0.80$ | $\mathbf{83.22 \pm 0.75}$ |
| GraphConv-s (trained) | $70.74 \pm 2.47$ | $\mathbf{57.39 \pm 2.20}$ | $\mathbf{35.89 \pm 2.00}$ | $56.89 \pm 1.14$ | $63.6 \pm 0.97$ | $\mathbf{54.82 \pm 2.35}$ |
| GraphConv-s (untrained) | $\mathbf{71.35 \pm 2.28}$ | $52.06 \pm 1.62$ | $33.31 \pm 1.09$ | $\mathbf{57.12 \pm 1.88}$ | $\mathbf{68.19 \pm 1.26}$ | $52.45 \pm 1.63$ |

Table 5: Trained and Untrained MPNNs (3-layer, 512-hidden-dimension) with *mean* aggregation and mean (graph-level) pooling, denoted by "MPNN-mean". We report the mean accuracy $\pm$ std over ten data splits. Although our theoretical results do not apply to mean aggregation, we still see that untrained MPNNs are competitive compared to their trained counterparts.

| Accuracy $\uparrow$ | MUTAG | IMDB-BINARY | IMDB-MULTI | NCI1 | PROTEINS | REDDIT-BINARY |
|---|---|---|---|---|---|---|
| GIN-mean (trained) | $74.63 \pm 2.93$ | $49.48 \pm 1.56$ | $33.70 \pm 1.35$ | $73.74 \pm 0.45$ | $71.53 \pm 0.93$ | $50.04 \pm 0.70$ |
| GIN-mean (untrained) | $72.46 \pm 2.56$ | $49.18 \pm 1.83$ | $33.03 \pm 1.12$ | $77.16 \pm 0.39$ | $70.33 \pm 0.95$ | $49.90 \pm 0.83$ |
| GraphConv-mean (trained) | $65.87 \pm 3.24$ | $49.32 \pm 1.35$ | $33.15 \pm 1.19$ | $54.39 \pm 1.25$ | $66.76 \pm 0.96$ | $49.68 \pm 0.82$ |
| GraphConv-mean (untrained) | $63.30 \pm 3.55$ | $48.80 \pm 1.91$ | $32.51 \pm 0.90$ | $55.84 \pm 0.53$ | $70.73 \pm 0.69$ | $49.39 \pm 0.48$ |

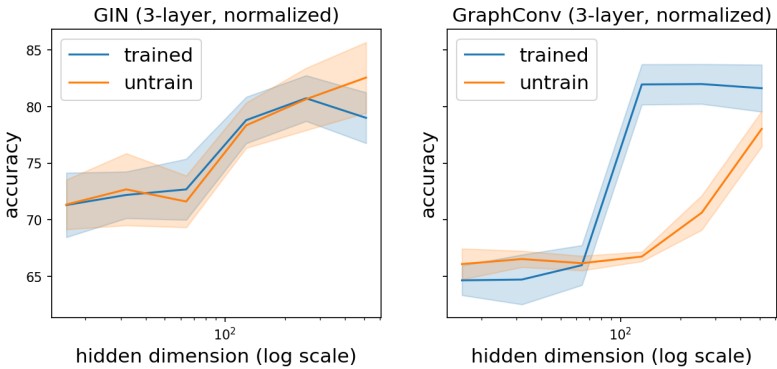

Figure 6: Increasing hidden dimension (number of functions) improves untrained MPNN performance, supporting Theorem 6.

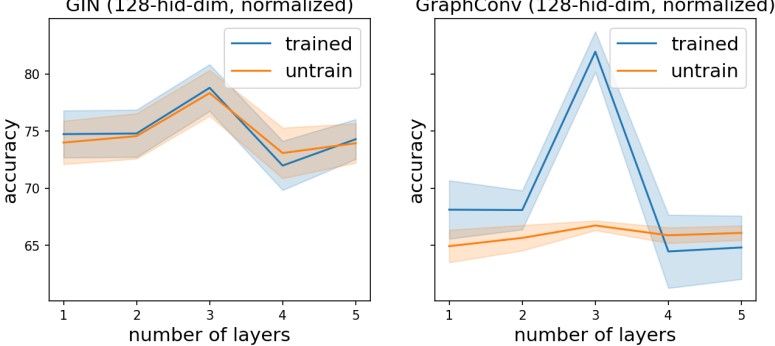

Figure 7: Increasing the number of message-passing layers (number of 1-WL iterations) first improves and then degrades the performance of untrained MPNNs.

Table 6: Classification accuracy of 1-NN on our graph distances. Results of $d_{\text{WL}}$ are taken from [26] using node degrees as initial labels.

| Accuracy ↑ | MUTAG | IMDB-BINARY |
|---|---|---|
| $d_{\text{WL}}^{(3)}$ [26] | $91.1 \pm 4.3$ | $69.4 \pm 3.9$ |
| $d_{\text{WLLB}}^{(3)}$ [26] | $85.2 \pm 3.5$ | $69.8 \pm 3.3$ |
| $\delta_{\text{W},3}$ | $87.89 \pm 4.11$ | $66.50 \pm 4.15$ |
| $\delta_{\text{W},\geq 3}$[1] | $86.32 \pm 4.21$ | $66.60 \pm 3.93$ |

## E.3 Evaluating graph distances in graph classification task

Our main theoretical results and empirical validation focus on the relationship between MPNNs and their induced graph distances, suggesting that the Euclidean distance between MPNN embeddings is an efficient lower bound of these graph distances. More precisely, the time complexity of computing $\delta_{\text{P},h}, \delta_{\text{W},h}$ is $\mathcal{O}(h \cdot n^5 \cdot \log n)$ (the same order as the WL distance in [26]). In contrast, the time complexity of computing $h$-layer, $d$-dimensional MPNN embedding distance (with $\psi_t, \psi$ chosen as the composition of linear maps and pointwise nonlinearities) is $\mathcal{O}(h \cdot n \cdot |E(G)| + n \cdot d)$, which is massively cheaper, especially for large, sparse graphs that are ubiquitous in real-world applications.

For ablation purposes, we evaluate our graph distances $\delta_{\text{P},h}, \delta_{\text{W},h}$ in graph classification tasks to examine how well they can separate graphs. For comparison, we follow the same set-up from Chen et al. [26]. Specifically, for each selected graph dataset in the TUDataset [90], we compute the pairwise distances for all graphs in the dataset. We then perform graph classification via 1-nearest-neighbor classifier (1-NN), which classifies the test graph based on the label of the closest training graph (measured by our graph distance). We use 90/10 training/test random split and repeat ten times, following the same random data split in [26].

Table 6 shows the mean classification accuracy of 1-NN using our graph distances and demonstrates that our graph distances achieve competitive classification performance as the WL distance in [26].

---

[1] $\delta_{\text{W}}$ computed up to at most three iterations after color stabilizes.

