# OpenReview forum: "Fine-grained Expressivity of Graph Neural Networks"
_NeurIPS.cc/2023/Conference — NeurIPS 2023 poster_

### Official Review · Reviewer_7umZ · 2023-07-04

**Soundness:** 3 good
**Presentation:** 3 good
**Contribution:** 2 fair
**Rating:** 7
**Confidence:** 2

**Summary:**

With graphon theory, this work quantifies which distance MPNNs induce and thus provides a deeper understanding of MPNNs’ capacity to capture graph structure, precisely determining when they can and when they cannot assign similar and dissimilar vectorial representations to graphs.

**Strengths:**

1. Strong universality results on graphon.
2. The iterated degree measure is intuitive and insightful.

**Weaknesses:**

1. Graphon focus on graph limit. While in real world settings like graph classfication, the graph is small and MPNN still lacks expressivity.

2. Only the sum aggregation is analyzed. While in application, max and mean aggregation are also important.

**Questions:**

Can you generalize the theory to max and mean aggregation?

**Limitations:**

Yes

---

> ### Author Rebuttal · Authors · 2023-08-09
>
> We thank the reviewer for their fair and constructive review.
>
> > Graphon focus on graph limit. While in real world settings like graph classfication, the graph is small and MPNN still lacks expressivity.
>
> You are correct that graphons are used in the GNN literature via large graphs. These works use graphons to formulate a generative model of the data. That is, graphs in the dataset are seen as sampled from some (usually small) set of graphons. In that approach, you need large graphs for the asymptotics to work. However, our work does not fall under this category, and does not define any generative model of graphs. In general, graphons are not limits of only large graphs. Any graph is a graphon, so graphons are also “limits” of small graphs. The space of graphs is a dense subset of the space of all graphons. The reason to consider graphon analysis is that we want to work with a compact metric space for results like the Stone–Weierstrass theorem to work. The space of graphons is the completion of the space of graphs to a compact metric space. To conclude, any probability distribution of graphs, including distributions restricted to small graphs, is a valid probability distribution of graphons. In the camera-ready version of the paper, we will clarify this point and the different philosophical approach from other GNN papers that use graphon analysis.
>
> > Only the sum aggregation is analyzed. While in application, max and mean aggregation are also important.
>
> > Can you generalize the theory to max and mean aggregation?
>
> Good question. Currently, the 1-WL paradigm of summing over the neighbors is crucial in our proofs, and extending our theory to max/mean aggregation functions is not straightforward. We are aiming to explore this in future work.

---

> > ### Comment · Reviewer_7umZ · 2023-08-10
> >
> > Thank you for the detailed reply. I keep my score to 7.

---

### Official Review · Reviewer_xsds · 2023-07-04

**Soundness:** 3 good
**Presentation:** 2 fair
**Contribution:** 3 good
**Rating:** 6
**Confidence:** 2

**Summary:**

This work aims at a fine-gined metric to characterize representation differences from MPNNs. Specifically, they first generalize MPNN to iterative degree measures and prove that Prokhorov metric and unbalanced Wasserstein metric can be used to bound the node/graph representation difference. This relation is validated in their experiments on random graphs.

**Strengths:**

 - [significance] This work builds a connection between the GNN representations and the graph similarity,  and makes a solid theoretical contribution to understand GNN expressivity.
 - Empirical results well support the theory.

**Weaknesses:**

- The motivation and the conclusion of untrained GNNs experiments are not very clear to me. How is "an untrained GNN can outperform trained GNN" related to the previous conclusions and what message does this part try to convey?
- Besides, TUDataset could be too small to faithfully evaluate the performance. Results on other datasets such as ZINC or OGB may further improve the evaluation.

**Questions:**

- Could you provide an intuitive explanation for Theorem 4, the universal theorem? Does it essentially mean that MPNNs have the same separation power as 1-WL on graphons?

---

> ### Author Rebuttal · Authors · 2023-08-09
>
> We thank the reviewer for their fair and constructive review.
>
> > The motivation and the conclusion of untrained GNNs experiments are not very clear to me. How is "an untrained GNN can outperform trained GNN" related to the previous conclusions and what message does this part try to convey?
>
> Our theory states: two graphs are close in our graph distance if and only if all L-layer C-Lipschitz MPNN embeddings are close. We want to apply it for the context of graph classification (Q2), which translates to: two graphs are in the same class if and only if all MPNN embeddings are close --- this includes both trained and untrained embeddings. This thus motivates us to examine the effectiveness of untrained MPNN embeddings.  Moreover, in practice, we cannot compute *all* MPNN embeddings, so we seek to test how well using a subset of them as an approximation (to graph distance) for graph classification. Our experiments demonstrate that using a subset of untrained MPNN embeddings does achieve competitive performance, illustrating the utility of our theoretical results.
>
> We will clarify better in the camera ready version the motivation behind the experiments. The theory says that we need to run over all possible MPNNs, and take the largest distance in the output space. In our experiments, we instead run over a finite set of random MPNNs. We indeed did not provide any “Monte Carlo” theory that relates the full MPNN space to the finite sample, and the theory in previous sections should be seen as heuristically motivating the experiments, not rigorously justifying them. We will clarify this in the camera ready version.
>
> > Besides, TUDataset could be too small to faithfully evaluate the performance. Results on other datasets such as ZINC or OGB may further improve the evaluation.
>
> Good point, we will try to add them in the camera ready.
>
> > Could you provide an intuitive explanation for Theorem 4, the universal theorem? Does it essentially mean that MPNNs have the same separation power as 1-WL on graphons?
>
> Intuitively, it says that every continuous function on graphons that is invariant under 1-WL colors can be approximated by an MPNN. This is in some sense a “continuous” or “metric” (and hence stronger) variant of the statement that MPNNs and 1-WL have the same separation power on graphons.

---

### Official Review · Reviewer_oM2o · 2023-07-06

**Soundness:** 3 good
**Presentation:** 2 fair
**Contribution:** 3 good
**Rating:** 6
**Confidence:** 3

**Summary:**

The paper considers the continuous variant of the 1-WL test and leverages it to characterize the expressive power of MPNNs on graphons.

The authors show that if two graphons have similar MPNN outputs then they are close in their metric, extending the existing result proving the opposite implication (graphons similar in their metric have similar MPNN outputs).
They establish a connection between the continuous variant of the 1-WL test and MPNNs to tree distance and tree homomorphism densities, where equivalences were only known to hold on a discrete level. Empirically, they show that untrained MPNNs (paired with a trained final classifier) obtain competitive performances on graph-level tasks. Finally, they experimentally compare different models in their ability to preserve graph distances.

**Strengths:**

1. The paper refines existing results on metrics and MPNNs outputs.
2. The paper aims at closing the gap between results that were known to hold on a discrete level (where graphs are exactly isomorphic), but that were under explored at the continuous level.

**Weaknesses:**

1. The paper is hard to read, as it introduces a lot of existing concepts and builds on them. Maybe it is inevitable due to intrinsic theoretical nature of the paper. However, I believe some extra effort is needed to remind the reader that a concept or theorem that is going to be introduced is, in simpler words, what has been presented in the introduction as a contribution.
2. I am unsure of the implications of this work. The theory is beautiful, but I am left unsure of what we can do with this gained knowledge. I don't see how this can help experimentally, or in real world cases.

**Questions:**

1. Can you discuss the implications of this work? While I understand that the theoretical claims hold, what can we do knowing that they hold?
2. Similarly to the above question, what is the take-home message of the experimental evaluation? I am unsure of the relevance of the questions you set to answer.

---

> ### Author Rebuttal · Authors · 2023-08-09
>
> We thank the reviewer for their fair and constructive review.
>
> > The paper is hard to read, as it introduces a lot of existing concepts and builds on them. Maybe it is inevitable due to intrinsic theoretical nature of the paper. However, I believe some extra effort is needed to remind the reader that a concept or theorem that is going to be introduced is, in simpler words, what has been presented in the introduction as a contribution.
>
> Thank you, we agree, we will use the additional page for the camera ready, to add some intuitions and make it more accessible to the wider GNN community.
>
> 1. We will define and discuss tree distance in the main paper instead of the appendix.
> 2. In *Message-passing graph neural networks* we will add a more extensive discussion about the relation between the formulation of MPNNs on IDMs and standard MPNNs on graphons/graphs. We agree that seeing the connection between the two formulation is critical for the understanding of our paper.
> 3. We will extend some of the formulas, opting for clearer longer explicit formulas rather than shorter implicit ones. For example, the formulas in line 253 can be replaced by explicit formulations.
>
> We kindly ask you to provide additional suggestions on specific parts that were hard to understand and should be improved.
>
> > I am unsure of the implications of this work. The theory is beautiful, but I am left unsure of what we can do with this gained knowledge. I don't see how this can help experimentally, or in real world cases.
>
> > Can you discuss the implications of this work? While I understand that the theoretical claims hold, what can we do knowing that they hold?
>
> We mainly view our work as a theoretical contribution, leading to a better understanding of what kind of functions GNNs express, potentially leading to a better understanding of their predictive performance. We think that our work is a necessary first step of defining the desired notion of graph similarity for GNNs, which then enables future work that is focused on better understanding these notions and the expressivity of GNNs.
> One main practical implication that the theory hints at is that we can use untrained MPNNs to solve various graph machine learning problems. In the camera-ready version, we will add a section discussing this.
>
> > Similarly to the above question, what is the take-home message of the experimental evaluation? I am unsure of the relevance of the questions you set to answer.
>
> Our experimental evaluation intends to show (Q1) how MPNN embedding Euclidean distance correlates with our proposed graph metrics, which illustrates Corollary 5; (Q2) how useful are untrained MPNN embedding to discriminate graphs, since our main results state that two graphs are close in our graph metrics if and only if all $L$-layer $C$-Lipschitz MPNN embeddings are close, which includes both trained and untrained MPNNs. Our experiments demonstrate that using a subset of untrained MPNN embeddings does achieve competitive performance, illustrating the utility of our theoretical results.
>
> An important implication for practitioners is to use MPNN embedding (distance) as an efficient lower bound to compute distances of large graphs. More precisely, the time complexity of computing $\delta_{\mathsf{W}, h}$ is $\mathcal{O}(h \cdot n^5 \cdot \log n)$ (the same order as the WL distance in [26], even worse for $\delta_{\mathsf{P}, h}$). In contrast, the time complexity of computing $h$-layer, $d$-dimensional MPNN embedding distance (with $\varphi_t$, $\psi$ chosen as the composition of linear maps and pointwise nonlinearities) is $\mathcal{O}(h · n · |E(G)| + n · d)$, which is massively cheaper, especially for large, sparse graphs that are ubiquitous in real-world applications.

---

> > ### Comment · Reviewer_oM2o · 2023-08-18
> >
> > I thank the authors for their reply. However, I would still argue that the implications of this work are hard to see, and consequently the experimental section is hardly relevant. However, I understand the authors' claim that this represents a first step, so potentially future work will have broader impact. I will keep my score.

---

### Official Review · Reviewer_Svzd · 2023-07-07

**Soundness:** 4 excellent
**Presentation:** 4 excellent
**Contribution:** 2 fair
**Rating:** 6
**Confidence:** 3

**Summary:**

The paper studies the classes of MPNNs on graphons, ultimately showing that the MPNN representations are sufficiently close (up to constants depending on Lipschitz regularity and layers) _if and only if_ the graphons are close according to several metric distances, mainly the Prokhorov metric and the unbalanced Wasserstein metric.

**Strengths:**

- The paper addresses a direction of research which is quite relevant, namely studying the response of GNNs and their dependence on properties of the input. In fact, the widely studied and adopted WL test is often a blunt yardstick for measuring the sensitivity of GNNs and investigating their dependence on metric functions defined on the space of graphs is worth looking into.
- The exposition of the paper and overview of the related works is well executed. The setup is clear.
- On a mathematical level, the works extends the results of [26] in several non-trivial ways and expands the efforts to understand GNNs through graphons in meaningful directions.

**Weaknesses:**

- The fact that MPNNs exhibit a Lipschitz property compared to distances on graphons does not seem surprising to me (i.e. Lemma 3) given their regularity. The converse statement (Theorem 6) is perhaps less obvious, although I fail to see how can this be used in practice? In fact, it is not the qualitative $\epsilon-\delta$ statements that are that interesting in my opinion, but rather the quantitative bounds which seem though to be lacking any significant insight? For example in Lemma 3 the constant depends on the fact that the features "have to" be bounded on a compact metric space, which is ultimately not enlightening. More generally, it would be interesting not to prove some Lipschitz property of MPNNs, but perhaps how some more transparent properties of the graph structure translate _quantitatively_ into the Lipschitz bounds.
- The analysis focuses on the case where graph(ons) are taken without features, which can be limiting although this setup is often conventional in the literature on expressivity of GNNs.
- There is an impractical cost for computing the distances which should be explicitly reported in the main text -- as it stands, mentioning the polynomial running time in Theorem 2 is a little too vague. While this is probably minor compared to the other points, it also contributes to reducing the impact of the submission.
- The experimental part is a little confusing. First, experiments on untrained MPNNs should be much better motivated, as it stands one can mainly guess what are you are trying to accomplish here. Second, validating that MPNNs can separate better with increased hidden dimension, is not surprising; besides, where is the role of the hidden dimension been discussed before? The fact that you can be competitive on some TUdataset "without" training over the layers is again not that surprising. These datasets are quite sensitive to hyperparameter tuning and I suspect that even removing the weights from the layer altogether and only train for the encoder and or decoder would be quite competitive. Maybe using different datasets -- more structural ones -- such as ZINC could be a little more indicative for a comparison.



**Questions:**

- As far as I understand, extending these type of results to GNNs that do not follow the 1-WL paradigm of summing over the neighbours is possibly non-trivial right? I think that it would be interesting in general to see how quantitative $\delta$-bounds depend on different paradigms for using the graph information (from structural encoding, to Graph-Transformers).

- I find that restricting to the feature-less case is a little detrimental to the overall message and I wonder in fact how challenging is extending this formalism to graphons with signals? I am not expecting a revision but am curious to hear comments here to gauge if this direction is actually pursuable or not.

- In the experiments you claim that your results support Corollary 4? What is Corollary 4 and where in the manuscript you have mentioned the role of the hidden dimension? Besides, is it in general really surprising that MPNNs with more hidden dimension may be better at separating graphs?

I think the paper has technical merits despite some mild concerns on the overall impact to the broader GNN community, and the score reflects this. The results are, on a qualitative level, intuitive, while on a quantitative level it is hard to extract some insight. The experimental section needs some revision as per my previous comments ---evaluating over a more structural-oriented task such as ZINC is perhaps a plus but not a must.

Disclaimer: I have not checked the mathematical proofs in the appendix.

**Limitations:**

Yes, limitations have been discussed and no societal impact can be foreseen.

---

> ### Author Rebuttal · Authors · 2023-08-09
>
> We thank the reviewer for their fair and constructive review.
>
> > The fact that MPNNs exhibit a Lipschitz property compared to distances on graphons does not seem surprising to me (i.e. Lemma 3) given their regularity. The converse statement (Theorem 6) is perhaps less obvious, although I fail to see how can this be used in practice? [...]
>
> Good point. Proving a quantitative version of Theorem 6 is, in fact highly non-trivial, cf. the very involved proof of the "Inverse Counting Lemma" in the book of Lovász [78]. We mainly view our work as a theoretical contribution establishing the first step of defining the appropriate notions. Lemma 3 and Theorem 6 show that our definitions are applicable in this sense. Then, proving quantitate bounds is the logical next step in this line of work. Additionally, we think that stating that the constant in Lemma 3 depends on the fact that the features "have to" be bounded on a compact metric space undermines the actual statement of Lemma 3: this is rather a technical detail that stems from the fact that graphons are normalized, but our MPNN functions are not. The vital part of the constants are the Lipschitz constants of the individual functions.
>
> > The analysis focuses on the case where graph(ons) are taken without features, which can be limiting although this setup is often conventional in the literature on expressivity of GNNs.
>
> > I find that restricting to the feature-less case is a little detrimental to the overall message and I wonder in fact how challenging is extending this formalism to graphons with signals? [...]
>
> That is an interesting question that fell victim to the page limit. In the camera-ready version, we will clarify better that the analysis is restricted to graphons without signals and add a remark on how this can be extended to graphons with signals.
>
> Here is what one has to adapt: Fix some compact metric space $K$ and consider graphons $W$ with a measurable signal function $\\ell \\colon [0,1] \\to K$. Replace the one-point space $\mathbb{M}_0$ by $K$ and adapt the definition of IDMs to include the old color in the new color (see the original definition of Grebík and Rocha [52]; including the old color in the new color becomes necessary also for 1-WL when the initial coloring is not constant since, otherwise, the second coloring would not include the initial coloring and so on).
>
> Then, the IDM spaces $\\mathbb{M}_{h}$ are still compact metrizable.
>
> Modify the 1-WL for graphons by setting $\\mathsf{i}_{(W,\\ell),0} \coloneqq \\ell$. In an MPNN model, $\\varphi_0$ is now a Lipschitz function $K \\to \\mathbb{R}^{d_0}$. Then, the universality result still holds since Lipschitz functions on $K$ are dense in $C(K)$. Finally, adapt the metrics by using the metric of the compact metric space $K$.
>
> Since the definitions of IDMs and our distances become more complicated and less intuitive, we did not directly include this in the main body. Moreover, homomorphisms and the tree distance do not have meaningful definitions for graphs with signals (as far as we are aware). Hence, we stated our main result for graphons without signals.
>
> > There is an impractical cost for computing the distances which should be explicitly reported in the main text -- as it stands, mentioning the polynomial running time in Theorem 2 is a little too vague. [...]
>
> Good point; we will make sure to point out the upper bounds, which we derived in the appendix, in the main body of the camera-ready version ($\mathcal{O}(h \cdot n^5 \cdot \log n) \text{ for } \delta_{\mathsf{W},h},\, \mathcal{O}(h \cdot n^7) \text{ for } \delta_{\mathsf{P},h}$). Moreover, we will stress that we do not propose computing the metrics in practice as a computational tool. The paper aims to hint that MPNNs can replace these metrics. The metrics are meaningful but hard to compute. On the other hand, MPNNs are mainly seen as black-box models but are easy to compute. The paper shows that *practical* MPNNs have the separation power of the *theoretical* metrics.
>
> > The experimental part is a little confusing. [...]
>
> We will clarify better in the camera-ready version the motivation behind the experiments. The theory says that we must run over all possible MPNNs, and take the largest distance in the output space. In our experiments, we instead run over a finite set of random MPNNs. We did not provide any “Monte Carlo” theory that relates the full MPNN space to the finite sample, and the theory in previous sections should be seen as heuristically motivating the experiments, not rigorously justifying them. We will clarify this in the camera-ready version.
>
> > As far as I understand, extending these type of results to GNNs that do not follow the 1-WL paradigm of summing over the neighbours is possibly non-trivial right? [...]
>
> These are interesting directions that we aim at exploring in future work. Currently, the 1-WL paradigm of summing over the neighbors is crucial in our proofs.
>
> > In the experiments you claim that your results support Corollary 4? What is Corollary 4 and where in the manuscript you have mentioned the role of the hidden dimension? Besides, is it in general really surprising that MPNNs with more hidden dimension may be better at separating graphs?
>
> We apologize for the typo and clarify that “Corollary 4” refers to Corollary 5, which states that two graph(on)s are close in our distance if and only if all their MPNN embeddings (of $L$-layer, $\psi$-Lipschitz) are close in the Euclidean metric. The wording “all” can be interpreted as using infinitely-many MPNNs, or collecting them into one MPNN with infinite hidden dimension size. In the experiment (Q2), we investigate the performance of increasing hidden size and observe that the performance of untrained MPNNs increases with hidden size, supporting our theory.

---

> > ### Comment · Reviewer_Svzd · 2023-08-13
> >
> > Thanks for the rebuttal, for clarifying how to extend the analysis to account for features and for commenting on the intended goals in the experimental section.
> >
> > Once again, I think this is a nice work -- and considered the other reviewers' scores I am confident it will be up for acceptance -- however my reservations about the broader implications remain. Selling it as a continuous version of 1-WL test is definitely appealing, but I think we cannot relate to the impact that the original work had in the discrete case.
> >
> > Some of the theoretical results that might be more appealing (quantitative statements / going beyond architectures bounded by the 1-WL test which nowadays are more and more only belonging to the academic world with the exception of node classification tasks over a single very large graph) are left for future work. In general, from a practitioner's side, I really can't see how this work can help settle some problems in the community (generalization, how properties of the training set affect the dynamics, how to design GNNs meaningfully more powerful) although I recognize that, potentially, this could be a first step towards achieving some of these goals.
> >
> > I still feel that the experimental section is a little weak, but this to me is super marginal considered the theoretical nature of the work.
> >
> > I also agree with another reviewer's observation about the choice of the layer-wise normalization not being conventional, this should be further emphasized in the paper.
> >
> > Having said that, I will maintain my score; thank you again for the rebuttal.

---

### Official Review · Reviewer_yRbN · 2023-07-09

**Soundness:** 3 good
**Presentation:** 4 excellent
**Contribution:** 4 excellent
**Rating:** 8
**Confidence:** 3

**Summary:**

The authors propose a novel way to generalize the expressivity of graph neural networks (and other general message passing algorithms) to the graphon case.
Furthermore, they identify metrics on graphons (and, consequently graphs) which allow to bound the distance of any MPNN representation of the graphons.


**Strengths:**

The paper provides an important step towards a more realistic analysis of MPNN expressivity.
This paper goes beyond both common approaches to measure expressivity of MPNNs, presenting an epsilon delta result for two metrics on graph(on)s and MPNNs with Lipschitz constant and fixed number of layers.
This properly generalizes the discrete setting and has potentially high impact sparking further research.


**Weaknesses:**

It seems to me that the definition of GNNs in Line 212 1/2 is a rather severe deviation from most MPNN formulations, as it requires activations to grow linearly with the graph size to offset the normalization $\frac{1}{|V(G)|}$. This seems to be generally only possible for dense graphs when e.g. a fixed set of (one hot encoded) categorical labels is present.

**Questions:**

I wonder if you can give some insights regarding the issue mentioned above?

Also, it is not clear to me how the discussion of Q2 in the experiments is connected to the theory in the previous sections. I might have missed the connection, but to me it is unintuitive why the Lipschitz-arguments should imply that larger $L$ should result in more similar outputs.

Minor Issues:
l 299: 'in an MPNNs'
p7 and p8 mention in relatively close distance two 'elegant' proofs. That struck me as a bit odd.


**Limitations:**

The authors discuss technical limitations such as the current restriction to discretely labeled graphs and 1-WL.
I don't see immediate negative societal impact of this work.

---

> ### Author Rebuttal · Authors · 2023-08-09
>
> We thank the reviewer for their fair and constructive review.
>
> > It seems to me that the definition of GNNs in Line 212 1/2 is a rather severe deviation from most MPNN formulations, as it requires activations to grow linearly with the graph size to offset the normalization. This seems to be generally only possible for dense graphs when e.g. a fixed set of (one hot encoded) categorical labels is present.
>
> > I wonder if you can give some insights regarding the issue mentioned above?
>
> The experiments show that normalized sum aggregation performs comparably to sum aggregation, and better than average aggregation. We have included the average aggregation experiments in the new Table 5 in the global response PDF:
> * Table 1 (main text) studies MPNNs with sum aggregation and $1/|V(G)|$ normalization, corresponding to our theory
> * Table 4 (appendix) studies MPNNs with sum aggregation without $1/|V(G)|$ normalization
> * new Table 5 (global response PDF) studies MPNNs with mean aggregation without $1/|V(G)|$ normalization
>
> All of above use the same MPNN backbone ($3$-layer, $512$-hidden-size) and experimental setup
>
> However, if you have a dataset of graphs with sizes that range from very small to very large, and the large graphs are sparse, then normalized sum aggregation is not appropriate. Luckily, most datasets consist of graphs of sizes that do not vary too much. Normalized sum aggregation is appropriate for such datasets, even if the graphs are sparse, as the normalization can be compensated by larger weights. We will add a comment about this in the camera ready version, and include experiments with average aggregation for comparison.
>
> On the theory side, the normalization by the number of vertices stems from viewing graphs as graphons. Requiring activations to grow linearly with the graph size makes sense in this context. For a very large graph, a small individual part of the graph does not play a significant role. More generally, if the size of our graphs grows towards infinity, then the importance of a set of fixed size goes towards zero. The activations have to grow with the graph size to identify these differences between graphs still. Hence, the Lipschitz constant we defined for an MPNN model has to grow to identify these differences. This is also what our theoretical result states: to guarantee $\varepsilon$-similarity of graph(on)s in our distances, we need similarity of MPNN models up to some Lipschitz constant $C$, where $C$ goes towards infinity as $\varepsilon$ goes to zero.
> Moreover, when taking an activation function from our setting to the usual, non-normalized setting, the normalization would have to become part of the function (to yield the same output), which would offset the linear growth.
>
> > Also, it is not clear to me how the discussion of Q2 in the experiments is connected to the theory in the previous sections. I might have missed the connection, but to me it is unintuitive why the Lipschitz-arguments should imply that larger $L$ should result in more similar outputs.
>
> Intuitively, you should equate the depth $L$ with the number of steps in 1-WL. With more layers you can separate graphs that can only be separated with more steps of 1-WL. Hence, larger $L$ can separate more graphs.
> Our theory states: two graphs are close in our graph distance if and only if all $L$-layer $C$-Lipschitz MPNN embeddings are close. We want to apply it for the context of graph classification (Q2), which translates to: two graphs are in the same class if and only if all MPNN embeddings are close—this includes both trained and untrained embeddings. This thus motivates us to examine the effectiveness of untrained MPNN embeddings.  Moreover, in practice, we cannot compute **all** MPNN embeddings, so we seek to test how well using a subset of them as an approximation (to graph distance) for graph classification. Our experiments demonstrate that using a subset of untrained MPNN embeddings does achieve competitive performance, illustrating the utility of our theoretical results.

---

> > ### Comment · Reviewer_yRbN · 2023-08-18
> >
> > Thank you for your clarifications. My questions have been answered. I agree that mentioning that typical graph datasets have graphs of similar size, for which the normalization indeed should not be an issue and cautioning the readers regarding the case of varying graph sizes.

---

### Author Rebuttal · Authors · 2023-08-09

We thank the reviewers for their fair and constructive reviews and appreciate that they recognize that we present a beautiful theory for graph similarity. Combined with a novel generalization of message-passing graph neural networks (MPNNs) to graphons, our theory allows us to prove that graph(on)s are close in a continuous variant of the 1-WL if and only if their MPNN outputs are close, an equivalence that was only known to hold in the discrete case (where graphs are either distinguished or not).

We mainly view our work as a theoretical contribution, leading to a better understanding of what kind of functions MPNNs express, potentially leading to a better understanding of their predictive performance. We further think that our work is a necessary first step of defining the desired notion of graph similarity for MPNNs, which then enables future work that is focused on better understanding these notions and the expressivity of MPNNs.

To illustrate our theory, we empirically investigate the relationship between MPNN embedding Euclidean distances and our proposed graph distances. Our experiments show that untrained MPNNs can be surprisingly effective, as our theory predicts, as long as we use **many** of them. Ideally, we would use **all** to preserve the graph distance (as our theoretical result states), but this is not possible of course. Our experiment shows that using enough of them (measured by the hidden size) suffices for downstream classification. Moreover, one can view MPNN embedding distances as an efficient lower bound for our graph distances since the time complexity of computing $h$-layer, $d$-dimensional MPNN embedding distances is massively cheaper than the time complexity of computing our distances $\delta_{\mathsf{W},h}$ and $\delta_{\mathsf{P},h}$, which is still polynomial but with an impractical exponent.

---

### Decision · Program_Chairs · 2023-09-21

**Decision:**

Accept (poster)

**Comment:**

The reviewers unanimously recommended to accept the paper.